# HodgeFlow Policy Search by Topologically Dissecting Temporal-Difference Signals in Non-Markovian Environments

**Zuyuan Zhang**[1]  **Sizhe Tang**[1]  **Tian Lan**[1]

## Abstract

Non-Markovian dynamics are commonly found in real-world environments due to long-range dependencies, partial observability, and memory effects. The Bellman equation that is the central pillar of Reinforcement learning (RL) becomes only approximately valid under Non-Markovian. Existing work often focus on practical algorithm designs and offer limited theoretical treatment to address key questions, such as what dynamics are indeed capturable by the Bellman framework and how to inspire new algorithm classes with optimal approximations. In this paper, we present a novel topological viewpoint on temporal-difference (TD) based RL. We show that TD errors can be viewed as 1-cochain in the topological space of state transitions, while Markov dynamics are then interpreted as topological integrability. This novel view enables us to obtain a Hodge-type decomposition of TD errors into an integrable component and a topological residual, through a Bellman–de Rham projection. We further propose HodgeFlow Policy Search (HFPS) by fitting a potential network to minimize the non-integrable projection residual in RL, achieving stability/sensitivity guarantees. In numerical evaluations, HFPS is shown to significantly improve RL performance under non-Markovian.

## 1. Introduction

Modern reinforcement learning (RL) systems are increasingly deployed in long-horizon, complex-dynamics environments, where long-range dependencies, partial observability, and memory effects are commonly found in these real-world

processes (Arulkumaran et al., 2017; García & Fernández, 2015; Possamaï & Tangpi, 2024; Zhang et al., 2025). The classical Markov assumption is often violated. The Bellman equation that is the central pillar of RL becomes only approximately valid: Temporal-difference (TD) errors demonstrate non-markovian structures that cannot be removed by simply increasing function class representations or tuning optimization hyper-parameters (Sutton, 1988; Tsitsiklis & Van Roy, 1996; Baird et al., 1995; Sutton et al., 2009).

Existing work often focuses on proposing RL algorithms with high-order Markov approximations by leveraging memory mechanisms to embed (or summarize) state/action histories. Classical treatments of partial observability introduce internal-state or finite-memory policy representations, e.g., finite-state controllers and internal-state policy gradients (Meuleau et al., 2013; Aberdeen & Baxter, 2025; Kaelbling et al., 1998). In deep RL, recurrent and sequence-based architectures are widely used to encode dependence, including DRQN-style recurrent value learning (Hausknecht & Stone, 2015), recurrent distributed replay (R2D2) (Kapturowski et al., 2018), recurrent actor–critic systems such as IMPALA with V-trace (Espeholt et al., 2018), and attention/transformer memory architectures such as GTrXL (Parisotto et al., 2020). In multi-agent RL, recurrent extensions such as R-MADDPG similarly leverage recurrency to handle partial observability and limited communication (Wang et al., 2020). Recent efforts have also investigated efficient dependence representations via sliding windows (Tasse et al., 2025) and approximations under certain conditional laws (Possamaï & Tangpi, 2024). However, there has been very limited theoretical treatment to address the key questions regarding non-Markovian RL: What dynamics are mathematically capturable by the Bellman framework? How to obtain optimal Markov approximations? Can we go beyond memory approaches and inspire novel algorithm classes under non-Markovian?

This paper presents a novel topological viewpoint and framework for TD-based RL under non-Markovian. In particular, we show that TD errors can be viewed as 1-cochain in the topological space of state transitions. Markov dynamics are then interpreted as topological integrability: One-step discrepancies encoded by TD errors can be fully explained

[1] Department of Electrical & Computer Engineering, The George Washington University, Washington, DC, USA . Correspondence to: Zuyuan Zhang <zuyuan.zhang@gwu.edu>, Sizhe Tang <s.tang1@gwu.edu>, Tian Lan <tlan@gwu.edu>.

*Proceedings of the 43rd International Conference on Machine Learning*, Seoul, South Korea. PMLR 306, 2026. Copyright 2026 by the author(s).

by a single global potential $u$, moving along any transition simply increases the potential by $u(s') - \gamma u(s)$ (Desbrun et al., 2006; Jiang et al., 2011; Lim, 2020). This novel view enables us to obtain a Hodge-type decomposition of TD errors into an integrable component (which is Markov and capturable by Bellman equation) and a topological residual. We develop a Bellman–de Rham projection in the corresponding Hilbert space to minimize this non-integrability residual, thus achieving an optimal integrable approximation for solving non-Markovian problems. The residual quantifies how far the environment–policy pair departs from an ideal Markov model and thus serves as a principled diagnostic signal measuring by how much standard TD learning is fundamentally mismatched to the data (Kaelbling et al., 1998; Tsitsiklis & Van Roy, 1996; Sutton et al., 2009).

Building on this framework, we propose HodgeFlow Policy Search (HFPS), a simple two-network scheme that explicitly projects TD errors onto their integrable component and trains the value function using only this well-behaved part. Rather than chasing state-of-the-art benchmark scores in our evaluation, we focus on specific regimes where standard TD learning is fragile: non-Markovian rewards, partially observed and dependent dynamics, and offline RL with dataset shift (Bacchus & Kabanza, 2000; Icarte et al., 2022; Hausknecht & Stone, 2015; Fu et al., 2020; Kumar et al., 2020; Kostrikov et al., 2021). Across these settings, HFPS delivers (i) a rigorous Hilbert-space formulation of TD integrability, (ii) a practical Topological Bellman Decomposition (TBD) algorithm that approximates the Hodge projection from data, and (iii) sensitivity and robustness guarantees that explain how the integrable update behaves under perturbations of rewards, discount factors, and approximation errors (Sutton, 1988; Tsitsiklis & Van Roy, 1996; Sutton et al., 2009; Jiang et al., 2011; Lim, 2020).

Contributions. (i) We formulate TD error as a Hilbert-space 1-cochain and introduce the notion of topological Bellman integrability, together with a Hodge-type decomposition into integrable and residual components (Desbrun et al., 2006; Jiang et al., 2011; Lim, 2020). (ii) We derive a Poisson characterization of the optimal potential and show that in the ideal Markov setting the topological residual vanishes, while in non-Markovian regimes it provides a quantitative measure of Bellman non-integrability (Sutton, 1988; Tsitsiklis & Van Roy, 1996; Jiang et al., 2011). (iii) We propose the Topological Bellman Decomposition (TBD) algorithm, which realizes the Hodge projection using two function approximators trained from replay data (Sutton et al., 1998; Mnih et al., 2015; Schulman et al., 2017; Sutton et al., 2009). (iv) We provide theoretical guarantees on consistency, stability, and sensitivity of the decomposition, and empirically study HFPS in path-dependent and partially observed control tasks where conventional TD learning becomes brittle (Kaelbling et al., 1998; Hausknecht & Stone, 2015; Fu et al.,

2020; Kumar et al., 2020; Kostrikov et al., 2021).

## 2. Preliminaries

In this section we put the discounted MDP in a measure-theoretic and Hilbert-space form that will be used throughout the paper. We first define discounted occupancy measures over state–transition triplets, and use them to build two Hilbert cochain spaces: a *0-cochain* space $C^0$ of state functions and a *1-cochain* space $C^1$ of functions on $(s, a, s')$ triplets. We then introduce a discrete de Rham differential $d : C^0 \to C^1$ that plays the role of a discounted temporal gradient, and define the associated zero-th order Hodge Laplacian $\Delta_0 = d^*d$ on $C^0$. All subsequent Hodge decompositions of TD errors will be formulated in terms of these objects.

A discounted Markov decision process (MDP) is a tuple $\mathcal{M} = (\mathcal{S}, \mathcal{A}, P, r, \gamma)$, where $\mathcal{S}$ is a (measurable) state space; $\mathcal{A}$ is an action space; $P(\mathrm{d}s' \mid s, a)$ is a Markov transition kernel on $\mathcal{S}$; $r : \mathcal{S} \times \mathcal{A} \to \mathbb{R}$ is a bounded measurable reward function; $\gamma \in (0, 1)$ is a discount factor. A policy $\pi(a \mid s)$ is a Markov kernel from $\mathcal{S}$ to $\mathcal{A}$. Given an initial distribution $d_0$ over $\mathcal{S}$ and a policy $\pi$, the induced trajectory $(S_0, A_0, S_1, A_1, \dots)$ is generated by $S_0 \sim d_0, \quad A_t \sim \pi(\cdot \mid S_t), \quad S_{t+1} \sim P(\cdot \mid S_t, A_t)$.

### 2.1. Occupancy Measures

We now define discounted occupancy measures that encode how frequently a fixed policy $\pi$ visits state–transition triplets under discounting.

**Definition 2.1** (Discounted triplet occupancy measure). Fix a policy $\pi$. The discounted triplet occupancy measure $\mu_\pi$ on $\mathcal{S} \times \mathcal{A} \times \mathcal{S}$ is defined by

$$\mu_\pi(B) = (1 - \gamma) \mathbb{E}_\pi \left[ \sum_{t=0}^\infty \gamma^t \mathbf{1}\{(S_t, A_t, S_{t+1}) \in B\} \right],$$
$$B \subseteq \mathcal{S} \times \mathcal{A} \times \mathcal{S}. \tag{1}$$

For any bounded measurable $g : \mathcal{S} \times \mathcal{A} \times \mathcal{S} \to \mathbb{R}$ we have $\int g \, \mathrm{d}\mu_\pi = (1 - \gamma) \mathbb{E}_\pi \left[ \sum_{t=0}^\infty \gamma^t g(S_t, A_t, S_{t+1}) \right]$. In particular, $\mu_\pi(\mathcal{S} \times \mathcal{A} \times \mathcal{S}) = (1 - \gamma) \mathbb{E}_\pi \left[ \sum_{t=0}^\infty \gamma^t \right] = 1$, so $\mu_\pi$ is a probability measure. Intuitively, $\mu_\pi$ captures the (discounted) frequency with which the system visits each state–transition triplet $(s, a, s')$ under policy $\pi$.

The discounted state occupancy measure $\nu_\pi$ is the marginal of $\mu_\pi$ on the first coordinate:

$$\nu_\pi(C) = \mu_\pi(C \times \mathcal{A} \times \mathcal{S}), \quad C \subseteq \mathcal{S}. \tag{2}$$

Equivalently, $\int u(s)\,\nu_\pi(\mathrm{d}s) = (1-\gamma)\,\mathbb{E}_\pi\left[\sum_{t=0}^\infty \gamma^t u(S_t)\right]$ for any bounded measurable $u : \mathcal{S} \to \mathbb{R}$. In implementation, we do not need to explicitly construct $\mu_\pi$ or $\nu_\pi$; it suffices to note that sampling $(s, a, s')$ from an off-policy replay buffer driven by $\pi$ amounts to estimating integrals with respect to $\mu_\pi$. When the replay buffer is exactly on-policy, this empirical distribution approximates $\mu_\pi$. With off-policy replay, the same empirical objective should instead be read as a projection under the replay-buffer measure $\hat{\mu}_\mathcal{D}$. The ideal orthogonality statements below are therefore population statements under $\mu_\pi$, whereas the implemented algorithm computes an empirical projection under the distribution actually used for mini-batch training.

## 2.2. Cochain Spaces and Inner Products

We view state functions and triplet functions as elements of Hilbert cochain spaces built from the occupancy measures above.

**Definition 2.2** (Cochain spaces). Fix a policy $\pi$. Define the 0-cochain and 1-cochain spaces

$$C^0 := L^2(\mathcal{S}, \nu_\pi) = \left\{u : \mathcal{S} \to \mathbb{R} \;\middle|\; \int_\mathcal{S} u(s)^2\,\nu_\pi(\mathrm{d}s) < \infty\right\}, \tag{3}$$

$$C^1 := L^2(\mathcal{S} \times \mathcal{A} \times \mathcal{S}, \mu_\pi) = \left\{f : \mathcal{S} \times \mathcal{A} \times \mathcal{S} \to \mathbb{R} \;\middle|\; \tag{4}\right.$$

$$\left. \int f(s, a, s')^2\,\mu_\pi(\mathrm{d}s, \mathrm{d}a, \mathrm{d}s') < \infty\right\}. \tag{5}$$

They are equipped with the inner products

$$\langle u_1, u_2\rangle_{C^0} = \int_\mathcal{S} u_1(s) u_2(s)\,\nu_\pi(\mathrm{d}s), \tag{6}$$

$$\langle f_1, f_2\rangle_{C^1} = \int f_1(s, a, s') f_2(s, a, s')\,\mu_\pi(\mathrm{d}s, \mathrm{d}a, \mathrm{d}s'). \tag{7}$$

We denote by $\|\cdot\|_{C^0}$ and $\|\cdot\|_{C^1}$ the norms induced by these inner products. Both spaces depend on the fixed policy $\pi$, but we omit this dependence from the notation when no confusion arises.

where $C^0$ can be thought of as the space of potential or value functions on states, while $C^1$ is the space of real-valued functions defined on transitions $(s, a, s')$ (such as TD errors or temporal gradients). The Hilbert-space structure will allow us to phrase TD-residual objectives as orthogonal projection problems.

## 2.3. Discrete de Rham Differential and Hodge Laplacian

We now introduce a linear operator that plays the role of a discounted temporal gradient.

**Definition 2.3** (Discrete de Rham differential). Define the linear operator

$$d : C^0 \to C^1, \qquad (du)(s, a, s') := u(s') - \gamma u(s). \tag{8}$$

This operator can be interpreted as a discounted forward difference along the temporal direction: if $u$ is a potential function on states, then $du$ measures the one-step variation in $u$ along the transition $(s, a, s')$, with the current state discounted by $\gamma$. In the language of algebraic topology, $d$ is the (discrete) coboundary operator mapping 0-cochains to 1-cochains.

*Remark* 2.4 (Discounted versus classical coboundary). The operator in Definition 2.3 is a discounted variant of the classical simplicial coboundary. In the undiscounted case $\gamma = 1$, it reduces to the usual transition difference $u(s') - u(s)$. For discounted RL, the factor $\gamma$ is not a topological decoration but is introduced to align the cochain differential with the Bellman residual $r + \gamma V(s') - V(s)$. Thus, all exactness, projection, and Hodge-type statements in this paper are made with respect to the discounted differential $d = d_\gamma$.

The next lemma shows that $d$ is a bounded linear operator between Hilbert spaces, so it admits a well-defined Hilbert adjoint $d^*$.

**Lemma 2.5** (Boundedness and adjoint of $d$). *The operator $d : C^0 \to C^1$ is linear and bounded. More precisely, there exists a constant $c > 0$ (depending only on $\gamma$) such that for all $u \in C^0$,*

$$\|du\|_{C^1}^2 = \mathbb{E}_{\mu_\pi}\left[(u(S') - \gamma u(S))^2\right] \;\leq\; c\,\|u\|_{C^0}^2. \tag{9}$$

*Consequently, there exists a unique Hilbert adjoint $d^* : C^1 \to C^0$ such that $\langle du, f\rangle_{C^1} = \langle u, d^*f\rangle_{C^0}$, $\forall u \in C^0$, $f \in C^1$.*

**Definition 2.6** (Zero-th order Hodge Laplacian). The (zero-th order) Hodge Laplacian on $C^0$ is defined by $\Delta_0 := d^*d : C^0 \to C^0$.

The operator $\Delta_0$ is self-adjoint and positive semidefinite. In our framework it plays the role of a topological stiffness operator on potentials $u \in C^0$; in later sections we will solve Poisson-type equations involving $\Delta_0$ to extract integrable components of TD errors.

## 3. Topological Bellman Decomposition

A key message of this section is that *Markov* dynamics correspond to *topological integrability* of TD errors: under an ideal Markov model, the one-step Bellman discrepancy can be fully explained by a single global potential (and in particular vanishes for the exact value function in expectation), whereas non-Markovian effects manifest as an irreducible

non-integrable residual. With this perspective, we reinterpret the Bellman TD error as a 1-cochain in the Hilbert space $C^1$, introduce the notion of *topological integrability* via exact 1-cochains, and then show that any TD error admits an orthogonal decomposition into an *integrable component* and a *topological residual*. The integrable component is the closest element in the (closed) range of the discrete differential $d$, while the residual lies in the orthogonal complement and captures irreducible topological inconsistency that cannot be explained by any global potential.

### 3.1. Bellman Error as a 1-Cochain

**Definition 3.1** (Value function and TD error). Given a policy $\pi$, a (candidate) value function is any measurable $V : \mathcal{S} \to \mathbb{R}$ that belongs to $C^0 = L^2(\mathcal{S}, \nu_\pi)$. Its temporal-difference (TD) error is the function

$$\delta_V(s, a, s') := r(s, a) + \gamma V(s') - V(s), \qquad (10)$$
$$(s, a, s') \in \mathcal{S} \times \mathcal{A} \times \mathcal{S}.$$

Since $r$ is bounded and $V \in C^0$ is square-integrable with respect to $\nu_\pi$, the function $\delta_V$ is square-integrable with respect to the triplet occupancy measure $\mu_\pi$, and hence $\delta_V \in C^1$. In other words, the TD error is no longer just a collection of scattered sample-wise residuals, but a well-defined vector in the Hilbert space $C^1$. This viewpoint allows us to apply tools such as orthogonal projection and Hilbert-space decompositions directly to TD errors.

### 3.2. Topological integrability and exact 1-cochains

We next identify the subspace of 1-cochains that can be written as discounted temporal gradients of some potential function on states. These are the *exact* 1-cochains.

**Definition 3.2** (Exact 1-cochains). The exact subspace of $C^1$ is defined as

$$\mathcal{E} := \text{im}(d) = \{ du : u \in C^0 \} \subseteq C^1. \qquad (11)$$

If $\delta \in \mathcal{E}$, this means that there exists a potential function $u \in C^0$ such that the one-step quantity can be written exactly as $\delta(s, a, s') = u(s') - \gamma u(s)$ for $\mu_\pi$-almost all $(s, a, s')$. This is the discrete, discounted analogue of an exact 1-form in differential geometry. As in the classical setting, exactness is the cochain-level property that corresponds to path-independent line integrals; we will use this interpretation later when discussing closed loops in the state space.

**Definition 3.3** (Topologically integrable value function). A value function $V$ is said to be *topologically integrable* (with respect to policy $\pi$) if its TD error $\delta_V$ lies in the exact subspace: $\delta_V \in \mathcal{E}$. Equivalently, there exists $u \in C^0$ such that $\delta_V(s, a, s') = u(s') - \gamma u(s)$ for $\mu_\pi$-almost all $(s, a, s')$.

Intuitively, topological integrability means that all one-step discrepancies encoded by $\delta_V$ can be fully explained by a single global potential $u$: moving along any transition simply increases the potential by $u(s') - \gamma u(s)$. In this case, TD errors behave like a discounted gradient field, and line integrals of $\delta_V$ along different paths only depend on the endpoints.

### 3.3. Hodge-type decomposition

We now show that any 1-cochain $f \in C^1$ admits an orthogonal decomposition into a component that lies in (the closure of) the exact subspace and a residual component that is orthogonal to all exact 1-cochains. This is a Hilbert-space analogue of a Hodge-type decomposition tailored to the operator $d$.

**Theorem 3.4** (Hodge-type decomposition in $C^1$). *Let $\mathcal{E}_0 := \text{im}(d) \subseteq C^1$ be the range of $d$, and let $\overline{\mathcal{E}} := \overline{\mathcal{E}_0}$ be its closure in $C^1$. Denote by $\overline{\mathcal{E}}^\perp$ the orthogonal complement of $\overline{\mathcal{E}}$ in $C^1$. Then for any $f \in C^1$ there exists a unique decomposition $f = f_{\text{ex}} + f_{\text{res}}$, such that $f_{\text{ex}} \in \overline{\mathcal{E}}$, $f_{\text{res}} \in \overline{\mathcal{E}}^\perp$, $\langle f_{\text{ex}}, f_{\text{res}} \rangle_{C^1} = 0$. The component $f_{\text{ex}}$ is the orthogonal projection of $f$ onto $\overline{\mathcal{E}}$ and is characterized variationally by $f_{\text{ex}} = \arg\min_{g \in \overline{\mathcal{E}}} \| f - g \|_{C^1}^2$, or, equivalently, by the orthogonality condition $\langle f - f_{\text{ex}}, du \rangle_{C^1} = 0, \quad \forall u \in C^0$. Moreover, if the range of $d$ is closed in $C^1$ (so that $\mathcal{E}_0 = \overline{\mathcal{E}}$), then there exists $u^* \in C^0$ such that $f_{\text{ex}} = du^*$, and $u^*$ is a minimizer of the quadratic functional $J(u) := \| f - du \|_{C^1}^2, \quad u \in C^0$. All minimizers differ by elements of $\ker(\Delta_0)$, where $\Delta_0 = d^* d$ is the Hodge Laplacian on $C^0$ and $\ker(\Delta_0) = \ker(d) \subseteq C^0$.*

The theorem above provides a purely functional-analytic decomposition for any 1-cochain. Specializing it to TD errors gives the canonical decomposition we will use throughout the paper.

**Corollary 3.5** (Topological decomposition of TD error). *Assume the range of $d$ is closed in $C^1$. Let $V : \mathcal{S} \to \mathbb{R}$ be a bounded value function with $\delta_V \in C^1$ its TD error. Then there exist $u_V^* \in C^0$ and a unique residual $\delta_V^{\text{res}} \in \overline{\mathcal{E}}^\perp$ such that $\delta_V = du_V^* + \delta_V^{\text{res}}$, and $u_V^* \in \arg\min_{u \in C^0} \| \delta_V - du \|_{C^1}^2$. We call $du_V^*$ the* integrable component *of the TD error—the closest exact TD structure to $\delta_V$ in the $C^1$ norm— and $\delta_V^{\text{res}}$ the* topological residual, *the portion of $\delta_V$ that is orthogonal to all discounted gradients $du$ and hence cannot be attributed to any potential function.*

The norm $\| \delta_V^{\text{res}} \|_{C^1}$ quantifies how far $\delta_V$ is from being topologically integrable, i.e., from being exactly representable as the discounted difference of a global potential. In particular, $\delta_V^{\text{res}} = 0$ if and only if $V$ is topologically integrable in the sense of Definition 3.3.

## 3.4. Poisson characterization of the optimal potential

The variational problem in Corollary 3.5 admits a natural Poisson-type optimality condition on the potential $u_V^\star$. This connects the integrable component of the TD error to a linear elliptic equation involving the Hodge Laplacian $\Delta_0 = d^*d$.

**Theorem 3.6** (Poisson equation for the optimal potential). *Let $V$ and $\delta_V$ be as in Definition 3.1. Consider the quadratic functional*

$$J(u) := \|\delta_V - du\|_{C^1}^2 = \langle \delta_V - du, \, \delta_V - du \rangle_{C^1}, \quad u \in C^0. \tag{12}$$

*Assume that: 1. the range of $d$ is closed in $C^1$ (so that $\mathcal{E}_0 = \mathrm{im}(d)$ is closed, and $du$ parametrizes the exact subspace), and 2. the Hodge Laplacian $\Delta_0 = d^*d$ is invertible on the orthogonal complement $(\ker \Delta_0)^\perp \subset C^0$. Then: (1) There exists a minimizer of $J(u)$ over $u \in C^0$, and it is unique within the subspace $(\ker \Delta_0)^\perp$. We denote this canonical minimizer by $u_V^*$. (2) The first-order optimality condition for $u_V^*$ is the Poisson-type equation $d^*\delta_V = d^*d\,u_V^* \iff d^*\delta_V = \Delta_0 u_V^*$, in the weak sense, i.e., as an identity of inner products against all test directions in $C^0$.*

The closed-range and invertibility conditions in Theorem 3.6 are well-posedness requirements for the projection problem rather than additional modeling assumptions on the environment. The closed-range condition ensures that the exact subspace is closed, so that the orthogonal projection is attained. Invertibility of $\Delta_0 = d^*d$ on $(\ker \Delta_0)^\perp$ fixes the usual non-uniqueness of potentials modulo null directions. In finite tabular problems these conditions reduce to standard finite-dimensional linear-algebra conditions; with neural function approximation, they motivate the empirical least-squares projection used by HFPS.

In finite-state settings, the operator $\Delta_0$ reduces to a positive semidefinite matrix that is closely related to a graph Laplacian on the state space. The Poisson equation in Theorem 3.6 then becomes a linear system whose solution $u_V^*$ yields the integrable component $du_V^*$ of the TD error, while the residual $\delta_V^{\mathrm{res}}$ captures cycle-level path-dependence of TD errors.

## 3.5. Measuring Bellman non-integrability

Having identified the integrable component $du_V^*$ and the topological residual $\delta_V^{\mathrm{res}}$ of a TD error, we now interpret the corresponding variational problem as a *Bellman–de Rham projection* and show how it can be approximated from finite data.

**Theoretical target problem.** Given a policy $\pi$, reward function $r$, and a candidate value function $V$, the ideal problem we aim to solve is the Bellman–de Rham projection in the Hilbert space $C^1$:

$$u_V^* = \arg \min_{u \in C^0} \|\delta_V - du\|_{C^1}^2, \qquad \delta_V^{\mathrm{res}} = \delta_V - du_V^*. \tag{13}$$

By Theorem 3.4, Corollary 3.5, and Theorem 3.6, under the closed-range and invertibility assumptions on $d$ and $\Delta_0$, the minimizer $u_V^*$ exists and is unique up to elements in $\ker(\Delta_0)$; the cochain $du_V^*$ is the orthogonal projection of $\delta_V$ onto the exact subspace $\mathcal{E}$; the residual $\delta_V^{\mathrm{res}}$ is orthogonal to all vectors of the form $du$; and the quantity $\|\delta_V^{\mathrm{res}}\|_{C^1} = \min_{u \in C^0} \|\delta_V - du\|_{C^1}$ is exactly the irreducible TD error that cannot be removed by any potential function. We interpret this norm as a measure of *Bellman non-integrability*. This residual is not intended as a universal, policy-free scalar of how non-Markovian an environment is. It is a Bellman-mismatch diagnostic for the current environment–policy–representation triple: it measures how far the observed TD signal is from discounted potential differences on the state representation being used. For a fixed policy, the construction can be read at the Markov reward process level. In RL, however, the residual is policy-dependent through both the occupancy measure $\mu_\pi$ and the TD signal $\delta_V$; as the policy changes, the experienced mismatch and the corresponding projection geometry change as well.

**Finite-sample approximation.** In practice, we only observe a finite dataset of experience: $\mathcal{D} = \{(s_i, a_i, r_i, s_i')\}_{i=1}^N$, typically obtained from a replay buffer driven by $\pi$ (or an off-policy variant). We approximate the $L^2$ norms and inner products in (13) using empirical averages.

Let $V_\theta$ be the current value-function approximator and $U_\phi$ a potential-function approximator (another neural network). The empirical TD error on a sample $(s_i, a_i, r_i, s_i')$ is $\delta_\theta(s_i, a_i, s_i') := r_i + \gamma V_\theta(s_i') - V_\theta(s_i)$. We then define the empirical projection loss as

$$\hat{\mathcal{L}}_{\mathrm{topo}}(\theta, \phi) := \frac{1}{N} \sum_{i=1}^N \Big( \delta_\theta(s_i, a_i, s_i') - \big[ U_\phi(s_i') - \gamma U_\phi(s_i) \big] \Big)^2. \tag{14}$$

This is precisely the empirical version of $\|\delta_{V_\theta} - dU_\phi\|_{C^1}^2$. Our Hilbert-space viewpoint shows that minimizing $\hat{\mathcal{L}}_{\mathrm{topo}}$ over $\phi$ corresponds to computing a least-squares projection of the TD error onto the exact subspace spanned by discounted temporal gradients $du$. In the off-policy case, this least-squares projection is computed under the mini-batch distribution induced by the replay buffer. Consequently, the residual is orthogonal to the learned exact component only with respect to the empirical inner product used for training, up to optimization and approximation error. A useful replay-batch diagnostic is

$$\rho_\mathcal{B} = \frac{|\langle \delta_\theta - dU_\phi, \, dU_\phi \rangle_{\hat{\mu}_\mathcal{B}}|}{\|\delta_\theta - dU_\phi\|_{\hat{\mu}_\mathcal{B}} \|dU_\phi\|_{\hat{\mu}_\mathcal{B}} + \varepsilon}, \tag{15}$$

where $\mathcal{B}$ is a replay mini-batch and $\varepsilon > 0$ is a numerical stabilizer. Smaller $\rho_\mathcal{B}$ indicates that the residual–integrable cross term is small under the same empirical geometry used by the learner.

**Consistency of the empirical projection.** The next result formalizes the sense in which the empirical procedure approximates the true Hilbert-space projection as the dataset grows.

**Theorem 3.7** (Empirical consistency and approximate projection). *Assume: 1. The function class $\{U_\phi : \phi \in \Phi\} \subset C^0$ is dense (or sufficiently rich) in $L^2(\nu_\pi)$. 2. We have a sequence of datasets $\{\mathcal{D}_N\}_{N \geq 1}$ whose empirical distributions satisfy the law of large numbers with respect to $\mu_\pi$, so that empirical averages of square-integrable functions converge almost surely to their $L^2(\mu_\pi)$ expectations. 3. For each $N$, there exists a (possibly idealized) global minimizer $\phi_N^* \in \arg\min_{\phi \in \Phi} \hat{\mathcal{L}}_{\text{topo}}(\theta, \phi)$ for the empirical loss in (14), with $\theta$ fixed. Then, as $N \to \infty$ with $\theta$ fixed, $\hat{\mathcal{L}}_{\text{topo}}(\theta, \phi_N^*) \longrightarrow \min_{u \in C^0} \|\delta_{V_\theta} - du\|_{C^1}^2 = \|\delta_{V_\theta}^{\text{res}}\|_{C^1}^2$, and $U_{\phi_N^*}$ converges in $L^2(\nu_\pi)$ to a potential function $u_{V_\theta}^*$ that solves the variational problem (13).*

This theorem shows that, with finite samples and a sufficiently expressive network class $U_\phi$, the empirical procedure asymptotically computes the orthogonal projection in the Hilbert space $C^1$. The limiting error $\|\delta_{V_\theta}^{\text{res}}\|_{C^1}$ is thus a principled measure of Bellman non-integrability.

**Theorem 3.8** (Mean-field degeneracy under a perfect MDP model). *Assume the environment is a standard MDP, the policy $\pi$ is fixed, and $V^\pi$ is the exact value function. Define the* mean TD error *(or Bellman defect) of a value function $V$ at the state level by $\bar{\delta}_V(s) := \mathbb{E}\big[r(S, A, S') + \gamma V(S') - V(S) \,\big|\, S = s\big]$, where the expectation is taken over $A \sim \pi(\cdot \mid s)$ and $S' \sim P(\cdot \mid s, A)$. Then, if we set $V = V^\pi$, we have $\bar{\delta}_{V^\pi}(s) \equiv 0, \quad \forall s \in \mathcal{S}$, so the mean-field Bellman defect is perfectly integrable (indeed, it vanishes identically), and the minimal population projection error at the state level is zero: $\min_{u \in C^0} \big\|\bar{\delta}_{V^\pi} - \tilde{d}u\big\|_{L^2(\mathcal{S})}^2 = 0$, where $(\tilde{d}u)(s) := \mathbb{E}[u(S') - \gamma u(s) \mid S = s]$ is the corresponding mean-field differential.*

Theorem 3.8 makes the Markov–integrability connection explicit at the population (mean-field) level: in a standard MDP, the Bellman equation holds exactly in expectation for $V^\pi$, so the structural Bellman discrepancy is perfectly integrable (indeed, it vanishes). In contrast, under genuinely non-Markovian environment–policy interactions, a nonzero topological residual $\|\delta_V^{\text{res}}\|_{C^1}$ reflects an *irreducible* path-dependence that cannot be eliminated by any state potential. At the finite-sample level, $\delta_V^{\text{res}}$ may additionally contain stochastic transition noise and approximation error, but it remains a principled diagnostic of how strongly TD learning is structurally mismatched to the data-generating process.

In other words, under a perfect Markov model and an exact value function, the *structural* (mean) Bellman inconsistency vanishes: the Bellman equation holds exactly in expectation. At the sample level, however, the realized TD error $\delta_{V^\pi}(s, a, s')$ still contains martingale noise due to stochastic transitions and reward variability, and our topological residual $\delta_{V^\pi}^{\text{res}}$ then reflects a combination of sampling noise and any remaining function-approximation error rather than a fundamental topological obstruction.

**Corollary 3.9** (Pythagorean identity for TD decomposition). *Under the assumptions of Corollary 3.5, let $\delta_V = du_V^\star + \delta_V^{\text{res}}$ be the Hodge-type decomposition of the TD error, with $du_V^\star \in \mathcal{E}$ and $\delta_V^{\text{res}} \in \mathcal{E}^\perp$ (where $\mathcal{E} = \text{im}(d)$ is the exact subspace in $C^1$). Then the following Pythagorean identity holds:*

$$\|\delta_V\|_{C^1}^2 = \|du_V^\star\|_{C^1}^2 + \|\delta_V^{\text{res}}\|_{C^1}^2. \tag{16}$$

*In particular, the norm of the topological residual $\|\delta_V^{\text{res}}\|_{C^1}$ is exactly the irreducible excess TD error that cannot be removed by any potential function.*

## 4. HodgeFlow Policy Search

The preceding sections introduced a topological decomposition of the TD error into an *integrable component $du_V^\star$* and a *topological residual $\delta_V^{\text{res}}$* and characterized the optimal potential $u_V^\star$ via a Poisson equation involving the Hodge Laplacian $\Delta_0 = d^*d$. We now turn this structure into a practical algorithmic principle. The core idea of *HodgeFlow Policy Search* (HFPS) is to introduce a potential network $U_\phi$ that learns the integrable component of the TD error, and then to update the value network $V_\theta$ using only this integrable TD signal. In this way, HFPS preserves the desirable TD direction whenever Bellman integrability holds, while filtering out the non-integrable residual when present.

Beyond serving as a decomposition artifact, the residual $\delta_V^{\text{res}}$ can also be used as a lightweight online diagnostic signal in RL. Its magnitude $\|\delta_V^{\text{res}}\|_{C^1}$ (or its empirical batch estimate) directly quantifies how strongly the data violate Bellman integrability, and can therefore act as an error/mismatch indicator under partial observability, history dependence, or offline dataset shift. In robust or conservative variants, one may use large residuals to gate or down-weight TD updates, reduce critic step sizes, or increase pessimistic regularization when the Markov approximation is unreliable. In this paper, we primarily use $\delta_V^{\text{res}}$ for monitoring and analysis, while the projected TD component is the core optimization signal of HFPS.

The following procedure implements the topological Bellman decomposition step that forms the inner loop of HFPS and can be combined with any off-policy TD-style actor–critic or value-based algorithm.

In Algorithm 1, the loss $\hat{\mathcal{L}}_{\text{topo}}$ is the empirical counterpart of the Hilbert-space objective $\|\delta_{V_\theta} - du\|_{C^1}^2$, with $U_\phi$ parametrizing the potential $u$. Minimizing this loss over $\phi$ therefore approximates the orthogonal projection of $\delta_{V_\theta}$ onto the exact subspace $\mathcal{E}$ (the TBD step). The value-

**Algorithm 1** HodgeFlow Policy Search (HFPS) with Topological Bellman Decomposition (TBD)

1: **Input:** value network $V_\theta : \mathcal{S} \to \mathbb{R}$; potential network $U_\phi : \mathcal{S} \to \mathbb{R}$;
   (optional) policy $\pi_\psi$; replay buffer $\mathcal{D}$ with tuples $(s, a, r, s')$; discount factor $\gamma$; batch size $B$;
   learning rates $\alpha_V, \alpha_U, \alpha_\pi$.
2: **for** iteration $t = 1, 2, \ldots$ **do**
3:   Collect experience with $\pi_\psi$ (or $\epsilon$-greedy) and append $(s, a, r, s')$ to $\mathcal{D}$
4:   Sample a minibatch $\{(s_i, a_i, r_i, s'_i)\}_{i=1}^B$ from $\mathcal{D}$ {Approximate sampling from $\mu_\pi$}
5:   **for** $i = 1, \ldots, B$ **do**
6:     Compute value-based TD error: $\delta_i \leftarrow r_i + \gamma V_\theta(s'_i) - V_\theta(s_i)$
7:     Compute potential-based differential: $\Delta_i \leftarrow U_\phi(s'_i) - \gamma U_\phi(s_i)$
8:   **end for**
9:   Define the empirical topological loss: $\hat{\mathcal{L}}_{\text{topo}}(\theta, \phi) \leftarrow \frac{1}{B} \sum_{i=1}^B (\delta_i - \Delta_i)^2$ {Empirical $\|\delta_{V_\theta} - dU_\phi\|_{C^1}^2$}
10:  **(TBD)** Update the potential network: $\phi \leftarrow \phi - \alpha_U \nabla_\phi \hat{\mathcal{L}}_{\text{topo}}(\theta, \phi)$
11:  (Optional) Recompute $\Delta_i$ with updated $\phi$: $\Delta_i \leftarrow U_\phi(s'_i) - \gamma U_\phi(s_i), \quad i = 1, \ldots, B$
12:  Define the *integrable TD component*: $\tilde{\delta}_i \leftarrow \Delta_i, \quad i = 1, \ldots, B$
13:  Optionally compute the *topological residual*: $\delta_i^{\text{res}} \leftarrow \delta_i - \tilde{\delta}_i, \quad i = 1, \ldots, B$
14:  **(TD learning / critic update)** Update the value network using only $\tilde{\delta}_i$: $\theta \leftarrow \theta - \alpha_V \frac{1}{B} \sum_{i=1}^B \tilde{\delta}_i \nabla_\theta V_\theta(s_i)$
15:  **(Optional actor update)** If an actor $\pi_\psi$ is used, update it with projected advantages: $\psi \leftarrow \psi + \alpha_\pi \frac{1}{B} \sum_{i=1}^B \tilde{\delta}_i \nabla_\psi \log \pi_\psi(a_i \mid s_i)$
16:  $\log \frac{1}{B} \sum_{i=1}^B (\delta_i^{\text{res}})^2$ as a non-integrability indicator.
17: **end for**

network update then uses only the projected, integrable TD component $\tilde{\delta}_i$, implementing the HFPS direction analyzed below. Finally, the optional residual statistic $(\delta_i^{\text{res}})^2$ provides an online estimate of Bellman non-integrability and can be used as a diagnostic or robustness indicator during training.

## 5. Sensitivity and Robustness Analysis

We study the stability of the Hodge-type decomposition and the resulting HFPS update under perturbations. We provide three bounds: (i) Lipschitz stability with respect to TD perturbations, (ii) sensitivity with respect to the discount factor, and (iii) a deviation bound showing that the gap between HFPS and standard TD updates is controlled by the topological residual.

**Theorem 5.1** (Stability of the integrable component under TD perturbations). *Assume there exists a bounded linear operator $T : C^1 \to C^0$ that returns a (canonical) minimizer $u_\delta^\star \in \arg\min_{u \in C^0} \|\delta - du\|_{C^1}^2$ for any $\delta \in C^1$. Denote $C_{topo} := \|T\|_{\text{op}} < \infty$.*

*Let $\delta_1, \delta_2 \in C^1$ and let $u_k^\star := T(\delta_k)$ for $k = 1, 2$. Then*

$$\|u_1^\star - u_2^\star\|_{C^0} \leq C_{topo} \|\delta_1 - \delta_2\|_{C^1}. \tag{17}$$

*Moreover, letting $\delta_k^{\text{res}} := \delta_k - du_k^\star$, we have*

$$\|\delta_1^{\text{res}} - \delta_2^{\text{res}}\|_{C^1} \leq \left(1 + \|d\|_{\text{op}} C_{topo}\right) \|\delta_1 - \delta_2\|_{C^1}. \tag{18}$$

Theorem 5.1 shows that the decomposition is Lipschitz-stable to TD noise and sampling variability in the $L^2(\mu_\pi)$ sense.

**Theorem 5.2** (Sensitivity of the Hodge decomposition to the discount factor). *Let $\delta_V^{(\gamma)}$ denote the TD error associated with $(\pi, V, \gamma)$. Assume there exists $L_\gamma < \infty$ such that for all $\gamma_1, \gamma_2 \in (0, 1)$,*

$$\|\delta_V^{(\gamma_1)} - \delta_V^{(\gamma_2)}\|_{C^1} \leq L_\gamma |\gamma_1 - \gamma_2|. \tag{19}$$

*Under the assumptions of Theorem 5.1, the corresponding optimal potentials satisfy*

$$\|u_V^{\star, (\gamma_1)} - u_V^{\star, (\gamma_2)}\|_{C^0} \leq C_{topo} L_\gamma |\gamma_1 - \gamma_2|. \tag{20}$$

Condition (19) is mild and holds, for example, when rewards and value functions are uniformly bounded.

**Theorem 5.3** (Bias bound for integrable-only semi-gradient updates). *Let $\delta_{V_\theta} = du_{V_\theta}^\star + \delta_{V_\theta}^{\text{res}}$ be the Hodge-type decomposition. Define*

$$g_{\text{TD}}(\theta) := \mathbb{E}_{\mu_\pi}\left[\delta_{V_\theta}(S, A, S') \nabla_\theta V_\theta(S)\right], \tag{21}$$

$$g_{\text{HFPS}}(\theta) := \mathbb{E}_{\mu_\pi}\left[du_{V_\theta}^\star(S, A, S') \nabla_\theta V_\theta(S)\right]. \tag{22}$$

*If $\|\nabla_\theta V_\theta(S)\| \leq B$ almost surely for some $B < \infty$, then*

$$\|g_{\text{HFPS}}(\theta) - g_{\text{TD}}(\theta)\| \leq B \|\delta_{V_\theta}^{\text{res}}\|_{C^1}. \tag{23}$$

Theorem 5.3 shows that HFPS is *asymptotically complete* with respect to TD learning: when $\|\delta_{V_\theta}^{\text{res}}\|_{C^1}$ is small, the HFPS update closely matches the standard TD semi-gradient; otherwise, HFPS filters out non-integrable components with a deviation controlled by the residual norm.

Corollary 5.4 identifies an exact regime in which HFPS reduces to standard TD. More generally, when the environment is only *approximately* integrable, the residual $\|\delta_{V_\theta}^{\text{res}}\|_{C^1}$ quantifies the mismatch and, by Theorem 5.3, acts as a uniform bound on the update perturbation. The next proposition formalizes a corresponding *robustness* guarantee in a

canonical setting: for linear policy evaluation with a globally stable TD ODE, HFPS converges to a neighborhood of the TD fixed point whose radius scales linearly with the residual level.

**Corollary 5.4** (Reduction to TD under integrability). *If $\delta_{V_\theta}^{\mathrm{res}} = 0$ (i.e., $\delta V_\theta \in \mathcal{E}$) then $g_{\mathrm{HFPS}}(\theta) = g_{\mathrm{TD}}(\theta)$. Therefore, in any setting where TD-style semi-gradient updates are known to converge (e.g., tabular on-policy evaluation, or standard linear TD under usual conditions), HFPS inherits the same convergence behavior.*

**Proposition 5.5** (Neighborhood convergence under bounded residual). *Consider fixed-policy evaluation with linear function approximation and assume the TD ODE is globally stable with modulus $\lambda > 0$. If along HFPS updates $\|\delta_{V_\theta}^{\mathrm{res}}\|_{C^1} \leq \varepsilon$ uniformly and $\|\nabla_\theta V_\theta(S)\| \leq B$, then the limiting point (or invariant set) of HFPS lies within $O((B/\lambda)\varepsilon)$ of the TD fixed point.*

## 6. Experiments

We design our experiments in two stages. First, we study small synthetic MDPs with tabular or linear function approximation, where the optimal potential and topological residual can be computed (or tightly approximated) in closed form. This stage validates the Hodge-type Bellman decomposition and the interpretation of $\|\delta_V^{\mathrm{res}}\|_{C^1}$ as a Bellman non-integrability measure. Second, we evaluate HodgeFlow Policy Search (HFPS) on deep control benchmarks under a newly introduced **Nonmarkov** regime, where the environment contains a hidden control-memory state that is *not* exposed in the observation. Results for standard Markov or partially corrupted observation regimes (**Clean**, **Noisy**, **Sticky**) and the full aggregate table are deferred to Appendix C. These regimes are lightweight wrappers designed to isolate distinct sources of mismatch: **Clean** keeps the original Markov observations, **Noisy** injects observation corruption/aliasing, and **Sticky** introduces temporal persistence in observations to emulate delayed or repeated sensing. All variants share the same underlying task and reward structure so that performance differences primarily reflect robustness to observation-side non-Markovian effects.

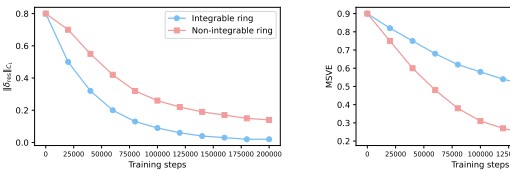

*Figure 1.* **Synthetic validation of the Bellman decomposition.** Left: tabular ring MDP (integrable vs. non-integrable). Right: random-feature MDP (MSVE and residual behavior).

**Synthetic settings.** We consider two synthetic setups. (i) A tabular ring MDP with states arranged on a cycle. We construct an *integrable* instance where rewards are exact discounted potential differences $r(s,a) = u^\star(s') - \gamma u^\star(s)$ and a *non-integrable* instance by perturbing a single edge reward so that cycle sums are non-zero. (ii) A random-feature MDP with 50 states and two actions, where $V$ and $u$ are represented as linear functions over fixed random features. In this setting we compute $V^\pi$ by matrix inversion and report mean squared value error (MSVE) along with $\|\delta_V^{\mathrm{res}}\|_{C^1}$.

**Control benchmarks and the Nonmarkov regime.** For deep control we evaluate on five continuous-state tasks: `LunarLander-v2`, `Acrobot-v1`, `Pendulum-v1`, `PointMass`, and `BipedalWalker-v3`. We derive a **Nonmarkov** variant from each base task by injecting a hidden *control-command memory* into the environment dynamics. Concretely, we augment the discrete action interface with a special *hold* command: when the agent selects the hold action, the environment executes the *last non-hold* action stored in an internal variable $u_{\mathrm{mem}}$; otherwise, the chosen action is executed and also overwrites $u_{\mathrm{mem}}$. Importantly, $u_{\mathrm{mem}}$ is *not* included in the observation, so from the agent's perspective the process is non-Markov (a POMDP induced by a hidden actuator state). This construction corresponds to the implementation in our wrapper `_HoldLastDiscreteAction` (Appendix C.1).

**Baselines and reporting.** We compare HFPS against DQN (Mnih et al., 2013), Double DQN (Van Hasselt et al., 2016), Dueling DQN (Wang et al., 2016), Advantage Actor–Critic (AC) (Mnih et al., 2016), PPO (Schulman et al., 2017), and recurrent baselines that explicitly model partial observability: DRQN (Hausknecht & Stone, 2015), ADRQN (Zhu et al., 2017), and R2D2 (Kapturowski et al., 2018). Unless otherwise specified, curves report mean $\pm$ one standard deviation across seeds. We additionally compute scalar summaries from the same learning-curve data: (A) **AUC@T** = area under the return curve up to the final training step $T$ (trapezoidal rule), (B) **Final@T** = average return at the final evaluation point. Full AUC@T / Final@T tables for all regimes are reported in Appendix C (Table 2).

**Synthetic results: validating the topological decomposition** Figure 1 summarizes the synthetic validation. On the integrable ring, HFPS drives $\|\delta^{\mathrm{res}}\|_{C^1}$ to numerical zero and recovers the ground-truth integrable component; on the non-integrable ring, the residual converges to a strictly positive value consistent with the optimal projection error. In the random-feature MDP, HFPS achieves lower MSVE and smoother convergence than a linear TD baseline, while the residual remains bounded away from zero due to representation error.

| Method | LunarLander | Acrobot | Pendulum | PointMass | BipedalWalker |
|---|---|---|---|---|---|
| **Nonmarkov summary** (AUC@T $\times 10^6$ / Final@T; mean $\pm$ std over seeds) | | | | | |
| HFPS | 107.6±2.1/145.6±12.9 | -82.7±7.5/-82.3±9.7 | -556.9±74.0/-106.2±121.9 | -839.7±64.3/-207.1±49.8 | 184.7±96.1/146.0±113.9 |
| DQN | -46.0±4.6/10.1±34.3 | -96.7±5.2/-77.2±4.4 | -965.2±107.0/-593.5±134.3 | -2672.2±347.1/-673.6±366.5 | -22.6±21.4/52.6±45.3 |
| Double DQN | -26.9±22.6/20.7±57.7 | -112.5±7.2/-84.2±14.0 | -1070.5±99.6/-866.6±106.6 | -2913.5±226.8/-594.8±47.4 | -85.9±11.2/-38.1±5.3 |
| Dueling DQN | -26.9±22.6/20.7±57.7 | -109.4±8.4/-73.6±3.7 | -946.0±114.5/-717.4±203.7 | -2609.5±192.2/-433.0±127.7 | -74.0±12.5/-30.3±32.5 |
| A2C | -395.4±394.8/-389.0±448.8 | -247.5±0.0/-500.0±0.0 | -1444.5±3.8/-1445.7±25.9 | -10130.7±136.1/-9536.1±811.3 | -137.6±39.8/-83.3±14.9 |
| PPO | -173.6±29.6/-172.4±31.4 | -92.0±4.0/-78.5±4.0 | -1342.8±84.3/-1353.5±255.1 | -9174.6±1804.6/-8230.2±6152.8 | 78.5±155.1/56.4±142.9 |
| DRQN | 93.1±11.8/169.1±3.1 | -246.9±0.7/-500.0±0.0 | -611.7±10.5/-235.8±47.8 | -1063.0±524.1/-473.3±301.6 | 23.6±100.4/80.6±100.7 |
| ADRQN | 75.0±7.4/165.2±9.5 | -247.5±0.0/-500.0±0.0 | -827.0±27.5/-403.9±98.0 | -1096.0±580.0/-256.4±39.2 | 69.9±73.8/135.1±72.9 |
| R2D2 | 81.7±5.8/154.0±29.3 | -246.9±0.7/-500.0±0.0 | -611.7±10.5/-235.8±47.8 | -1063.0±524.1/-473.3±301.6 | 23.6±100.4/80.6±100.7 |

*Table 1.* Scalar summaries computed from the same training runs: **AUC@T** measures sample efficiency (area under the return curve up to budget $T$) and **Final@T** measures final performance. Full learning curves and additional regimes are reported in Appendix C.

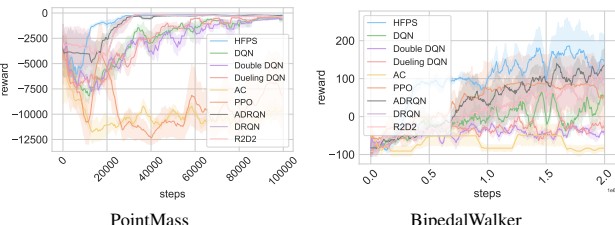

*Figure 2.* **Episode returns (Nonmarkov).** Return versus environment steps (mean $\pm$ one standard deviation over seeds) for HFPS and baselines on two representative control tasks.

**Control benchmarks: Nonmarkov learning curves**  Table 1 reports scalar summaries (AUC@T and Final@T) for the **Nonmarkov** variants of the five control tasks. HFPS remains competitive with strong Markov baselines even though the observation is no longer Markov, and it is particularly stable relative to non-recurrent baselines in tasks where hidden actuation memory induces long-range credit assignment. Full learning curves and additional regimes (**Clean**, **Noisy**, **Sticky**) are provided in Appendix C. Figure 2 further visualizes the Nonmarkov learning curves across tasks.

**Derived diagnostics from learning-curve data**  To better separate *sample efficiency* from *final performance* using the same Nonmarkov learning-curve data, we provide two derived diagnostics computed directly from the return curves. **Cumulative AUC trajectories.** Let $R_k$ denote the evaluation return at step $t_k$. We define the cumulative area $\mathrm{cAUC}(t_m) = \sum_{k=1}^{m-1} \frac{(R_{k+1}+R_k)}{2}(t_{k+1}-t_k)$. Figure 5 plots $\mathrm{cAUC}(t)$; steeper growth indicates higher performance earlier (better sample efficiency). **Across-seed variability over time.** Using the same evaluations, we compute the standard deviation across seeds at each step and plot it versus steps in Figure 7. Lower values indicate more stable learning under the same training protocol.

## 7. Conclusion

We introduced a topological view of temporal-difference learning by casting the TD error as a Hilbert-space 1-cochain and formalizing *Bellman integrability* as exactness. This yields a Hodge-type decomposition that separates TD signals into an integrable component explainable by a global potential and a topological residual that quantifies irreducible non-Markovian inconsistency. Building on this structure, we proposed HodgeFlow Policy Search (HFPS), which learns and updates using only the integrable TD component while tracking the residual as a principled diagnostic of Bellman mismatch. Empirically, HFPS improves stability and robustness in synthetic non-integrable settings and control tasks with hidden actuation memory, highlighting the practical value of topological decomposition for TD-based RL under non-Markovian effects.

**Limitations.**  HFPS should be interpreted as a Bellman-compatible filtering mechanism rather than a universal replacement for state augmentation, belief-state estimation, or recurrent memory. The exact Hodge orthogonality results are formulated under a population occupancy measure, while the implemented algorithm uses finite replay batches and therefore computes an empirical projection that can be affected by distribution shift, function approximation, and optimization error. The method also introduces additional critic-side computation through the potential network and projection loss, although it does not require extra environment interaction or longer rollouts.

## Acknowledgments

This work was supported in part by the Office of Naval Research (ONR) under Grant No. N00014-23-1-2532 and the Army Research Office (ARO) under Grant No. W911NF2420166.

## Impact Statement

This paper presents work whose goal is to advance the field of Machine Learning. There are many potential societal consequences of our work, none which we feel must be specifically highlighted here.

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

## A. Reader Guide to the Cochain Terminology

This appendix summarizes the topological terminology used in the main text in the notation of TD learning. A 0-cochain is a square-integrable scalar function on states, so it plays the role of a potential or value-like function $u \in C^0 = L^2(\mathcal{S}, \nu_\pi)$. A 1-cochain is a square-integrable scalar function on transitions $(s, a, s')$, so TD errors and temporal gradients live in $C^1 = L^2(\mathcal{S} \times \mathcal{A} \times \mathcal{S}, \mu_\pi)$. The discounted differential maps a potential to a transition signal, $(du)(s, a, s') = u(s') - \gamma u(s)$. The exact subspace consists of all transition signals that can be written in this form. The adjoint $d^*$ is defined by the Hilbert identity $\langle du, f \rangle_{C^1} = \langle u, d^* f \rangle_{C^0}$, and the Hodge Laplacian is $\Delta_0 = d^* d$. The Hodge-type projection used by HFPS is therefore simply the least-squares projection of a TD signal onto the set of discounted potential differences.

## B. Implementation of HodgeFlow Policy Search

This section describes the practical implementation of HodgeFlow Policy Search (HFPS) used in our experiments. We first summarize the common RL framework and network components shared by all methods, and then detail the specific variants of HFPS that we found to be stable and effective in practice.

### B.1. Common RL framework

All algorithms in our experiments are implemented within a unified RL framework. Environments expose a minimal interface

$$s_0 \sim \text{reset}(), \qquad (s', r, \text{done}, \text{info}) = \text{step}(a),$$

together with attributes describing the observation shape and the number of discrete actions. Agents implement a common interface with three methods:

- $\text{act}(s, \text{evaluate})$: select an action in state $s$;

- $\text{store\_transition}(s, a, r, s', \text{done})$: add a transition to the replay buffer or trajectory;

- $\text{update}()$: perform zero or more gradient steps.

The training loop only interacts with environments and agents through this interface, so DQN, AC, and HFPS differ only in how they implement $\text{update}()$ and, in the case of HFPS, an additional potential network.

### B.2. Shared neural architectures

To ensure architectural comparability, we factor each algorithm's network into a shared *backbone* and a small task-specific head. For low-dimensional observations (e.g., CartPole and TinyGrid) we use a multi-layer perceptron (MLP) that flattens the input and applies two fully connected layers with ReLU activations. For image-like inputs with shape $(C, H, W)$ we use a small convolutional network followed by an adaptive global average pooling layer and a linear projection to a feature vector.

The backbone type (MLP vs. CNN) is selected automatically based on the observation shape and is then reused across all algorithms:

- DQN uses the backbone followed by a linear "Q-head" producing one scalar per action.

- AC uses the same backbone followed by separate linear actor and critic heads.

- HFPS uses the backbone for both the Q-network and the potential network, with a linear head producing $Q_\theta(s, a)$ or a scalar potential $u_\phi(s)$, respectively.

### B.3. Stable HFPS with clipping and inner-loop updates

In the idealized formulation of HFPS, the potential $u_\phi(s)$ represents the exact solution of a Poisson equation that projects the temporal-difference (TD) error onto the integrable subspace. In practice, however, both the Q-network and the potential are

implemented by neural networks and updated via stochastic gradient descent (SGD). A naive implementation that directly regresses

$$u_\phi(s') - u_\phi(s) \approx r + \gamma \max_{a'} Q_\theta(s', a') - Q_\theta(s, a)$$

tends to be numerically unstable and can easily diverge.

To stabilize training we introduce two standard modifications:

1. **TD error clipping.** We compute the DQN-style TD target

$$T_\theta(s, a, s') := r + \gamma \max_{a'} Q_{\bar\theta}(s', a')$$

   using a slowly updated target network $Q_{\bar\theta}$, and define the TD error $\delta = T_\theta(s, a, s') - Q_\theta(s, a)$. We then clip this error elementwise to a fixed range,

$$\delta_{\text{clip}} = \text{clip}(\delta, -\Delta_{\max}, \Delta_{\max}),$$

   which prevents extremely large TD values from destabilizing the potential network.

2. **Inner-loop updates for the potential.** Instead of taking a single gradient step on $u_\phi$ per environment step, we perform a short inner loop of $K$ updates on the same mini-batch. In each inner step we minimize the squared difference between the potential difference and the clipped TD error:

$$\mathcal{L}_u(\phi) = \mathbb{E}\big[\big(u_\phi(s') - u_\phi(s) - \delta_{\text{clip}}\big)^2\big].$$

   This makes the Hodge projection approximation substantially more accurate and leads to much more stable Q-updates.

Both the Q-network and the potential network use gradient clipping (we clip the global $\ell_2$ norm of each parameter vector) to further improve stability.

## B.4. Residual-adaptive and norm-preserving TD updates

Even with clipping and inner-loop updates, simply replacing the TD error $\delta$ by the integrable component $u_\phi(s') - u_\phi(s)$ can degrade learning performance when the Hodge projection is inaccurate. In particular, if the residual component is large, the potential network may discard a significant portion of the true TD signal, leading to very small effective updates.

To address this, we use the Hodge decomposition not as a hard replacement but as a *residual-adaptive preconditioner* for the TD direction. After the inner-loop has approximately fitted the potential, we compute

$$du(s, a, s') := u_\phi(s') - u_\phi(s), \qquad \delta_{\text{int}} := \text{clip}\big(du(s, a, s'), -\Delta_{\max}, \Delta_{\max}\big),$$

and define the clipped residual

$$\delta_{\text{res}} := \delta_{\text{clip}} - \delta_{\text{int}}.$$

We then measure the average norms $\|\delta_{\text{clip}}\|, \|\delta_{\text{int}}\|, \|\delta_{\text{res}}\|$ over the mini-batch and construct a scalar *projection quality* score

$$q := \max\left\{0, \ 1 - \frac{\|\delta_{\text{res}}\|}{\|\delta_{\text{clip}}\| + \varepsilon}\right\} \in [0, 1].$$

When the residual norm is small compared to the TD norm, $q$ is close to 1; when the residual is as large as the TD error itself, $q$ approaches 0.

Using this quality score, we define an effective topological weight

$$\lambda_{\text{eff}} = \lambda_{\max} \, q^p,$$

where $\lambda_{\max} \in [0, 1]$ is a user-chosen maximum weight (denoted as `topo_weight` in our code) and $p \geq 1$ (`gate_power`) controls how sharply the weight decays as the residual grows. The raw mixed TD direction is then

$$\delta_{\text{raw}} := \lambda_{\text{eff}} \, \delta_{\text{int}} + (1 - \lambda_{\text{eff}}) \, \delta_{\text{clip}}.$$

In regions where the environment-policy pair is close to Bellman integrable, the residual is small, $q \approx 1$, and $\lambda_{\text{eff}} \approx \lambda_{\max}$, so HFPS updates follow the integrable component. Conversely, in strongly non-Markovian or partially observed regimes, the residual grows, $q$ decreases, and the update smoothly reverts toward the original TD direction.

Finally, to avoid accidentally shrinking the effective learning rate when $\|\delta_{\text{int}}\| < \|\delta_{\text{clip}}\|$, we rescale the mixed direction so that its norm approximately matches that of the original TD error:

$$\delta_{\text{eff}} := \frac{\|\delta_{\text{clip}}\|}{\|\delta_{\text{raw}}\| + \varepsilon} \, \delta_{\text{raw}},$$

with the rescaling factor clipped from above to avoid extreme amplification. The Q-network is then updated using a one-step regression towards $Q_\theta(s, a) + \delta_{\text{eff}}$,

$$\mathcal{L}_Q(\theta) = \mathbb{E}\big[\big(Q_\theta(s, a) - (Q_\theta(s, a) + \delta_{\text{eff}})\big)^2\big],$$

which is equivalent to a semi-gradient update in the direction $\delta_{\text{eff}} \nabla_\theta Q_\theta(s, a)$.

In summary, the practical HFPS variant used in our experiments performs:

1. TD-error computation with a target network and value clipping;

2. an inner loop of SGD steps on $u_\phi$ to fit the integrable component;

3. residual-adaptive weighting between integrable and raw TD components via $\lambda_{\text{eff}}$; and

4. norm-preserving rescaling of the mixed TD direction before updating $Q_\theta$.

This implementation retains the interpretability benefits of the topological decomposition—through the norms of the integrable and residual components—while maintaining the robustness of standard TD learning in regimes where Bellman integrability is strongly violated.

**Instantiation in synthetic experiments.** For the tabular ring MDPs in Section 6, we use a ring of $n = 10$ states with two deterministic actions moving clockwise and counterclockwise. Episodes are truncated at a horizon of $H = 50$ steps with a discount factor $\gamma = 0.99$. In the integrable version, we sample a ground-truth potential $u^\star(s)$ uniformly from $[-1, 1]$ and define rewards by $r(s, a) = u^\star(s') - \gamma u^\star(s)$, so that the Bellman field is exactly integrable. In the non-integrable version, we add a small perturbation $\varepsilon = 0.1$ to the reward on a fixed directed edge of the ring, breaking the cycle-sum condition. We train a tabular Q-learning agent and a tabular HFPS agent with identical $\varepsilon$-greedy exploration schedules (linearly decayed from 1.0 to 0.05) and constant learning rates for both Q and the potential table. Results are averaged over 5 random seeds.

For the random-feature MDP, we consider $|\mathcal{S}| = 50$ states and $|\mathcal{A}| = 2$ actions with a fixed transition kernel and reward function sampled once at the beginning of each run. We construct random feature vectors $\phi(s) \in \mathbb{R}^d$ (with $d = 32$) and represent both the value function and the potential as linear models in this feature space. The exact value function $V^\pi$ is computed by solving the Bellman equation with a closed-form matrix inversion, and we monitor the mean squared value error (MSVE) between the learned $V_\theta$ and $V^\pi$ as training proceeds. Linear TD and linear HFPS share the same step sizes and feature maps; we again average results over multiple seeds.

**Instantiation in control benchmarks.** For the control tasks (LunarLander-v2, Noisy-LunarLander, and Acrobot-v1) we use the vector-based observations provided by the environments, so the backbone is instantiated as a two-layer MLP with 128 hidden units per layer and ReLU activations. DQN and Double DQN use this backbone followed by a linear head that outputs one Q-value per action. The AC baseline uses the same backbone with separate linear actor and critic heads. HFPS uses two networks with the same backbone structure: one for $Q_\theta(s, a)$ and one for the scalar potential $u_\phi(s)$.

All deep RL experiments use the Adam optimizer with learning rate $3 \times 10^{-4}$ for both value and potential networks, a replay buffer of size $10^5$, mini-batches of size 64, and a target-network update interval of 1,000 environment steps for value-based methods. The discount factor is set to $\gamma = 0.99$. Each training run lasts for $5 \times 10^5$ to $1 \times 10^6$ environment steps depending on the task, with evaluation episodes (no exploration noise) every 5,000 steps. We run 10 random seeds for each algorithm–environment pair and report mean and standard deviation of episode returns.

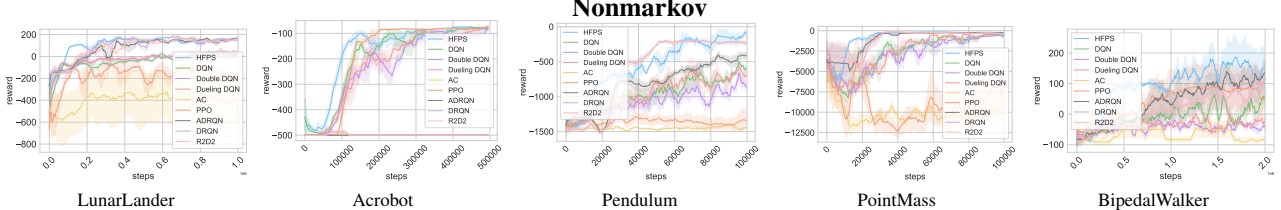

**Nonmarkov**

*Figure 3.* **Episode returns.** Return versus environment steps (mean $\pm$ one standard deviation over seeds) for HFPS and baselines on five control tasks (columns) under Nonmarkov

In the Noisy-LunarLander variant, we introduce observation aliasing by replacing the freshly observed state $s_t$ with the previous observation $s_{t-1}$ with probability $p_{\text{alias}} = 0.1$ at each time step. All other environment parameters remain identical to the standard LunarLander-v2. The Acrobot-v1 task is used with its default reward and termination conditions; no additional shaping is introduced.

## C. Additional regimes, full tables, and Nonmarkov construction

This appendix reports (i) results under **Clean**, **Noisy**, and **Sticky** regimes, and (ii) the full aggregate summary table used in the main text (Table 2). We also document the exact Nonmarkov construction used in our experiments.

### C.1. Nonmarkov regime via hidden control-command memory

We implement the Nonmarkov regime by injecting a hidden actuator memory state $u_{\text{mem}}$ into the environment. At each time step, the agent selects a discrete action $\tilde{a}_t$ from an augmented action set that includes a special *hold* index. The executed action $a_t$ is determined by

$$a_t = \begin{cases} u_{\text{mem}}, & \tilde{a}_t = a_{\text{hold}} \text{ and hold budget not exceeded}, \\ \tilde{a}_t, & \tilde{a}_t \neq a_{\text{hold}}, \end{cases} \qquad u_{\text{mem}} \leftarrow \begin{cases} u_{\text{mem}}, & \tilde{a}_t = a_{\text{hold}}, \\ \tilde{a}_t, & \tilde{a}_t \neq a_{\text{hold}}. \end{cases}$$

Optionally, we cap consecutive holds by a maximum `max_hold_steps`; if exceeded, the memory is reset to a default action to avoid degenerate perpetual actuation. Crucially, $u_{\text{mem}}$ is not included in the observation, so the induced process is non-Markov from the agent's perspective. This corresponds to our wrapper `_HoldLastDiscreteAction`.

### C.2. Control benchmarks under Clean/Noisy/Sticky regimes

We evaluate the same five benchmarks used in the main text under three additional regimes derived from the same base task. **Clean** is the standard Markov observation. **Noisy** introduces observation corruption (e.g., additive noise / aliasing) while keeping the underlying dynamics unchanged. **Sticky** introduces temporal persistence in the observation/state interface (a portion of the previous observation is reused), yielding partial observability with a different structure than the Nonmarkov actuator memory.

**Learning curves (Clean/Noisy/Sticky).**   Figure 4 reports episode return versus environment steps for all three regimes.

**Derived diagnostics (cAUC and Std).**   Figures 6 and 8 report cumulative AUC and across-seed variability for the same runs.

### C.3. Aggregate summary table (all benchmark–regime pairs)

**Summary of Clean/Noisy/Sticky regimes.**   The additional regimes separate different sources of mismatch. Clean keeps the original Markov observation, Noisy adds observation corruption or aliasing, and Sticky introduces temporal persistence in the observation interface. Across these regimes, HFPS is competitive in Clean tasks and usually degrades more gracefully under Noisy or Sticky perturbations. For example, in Final@T, HFPS reaches $100.88 \pm 43.37$ on LunarLander-Noisy compared with $-36.76 \pm 127.95$ for the strongest baseline in that row, $-230.07 \pm 16.41$ on PointMass-Noisy compared with $-329.73 \pm 72.07$, and $-296.23 \pm 69.91$ on PointMass-Sticky compared with $-399.89 \pm 86.85$. These results support

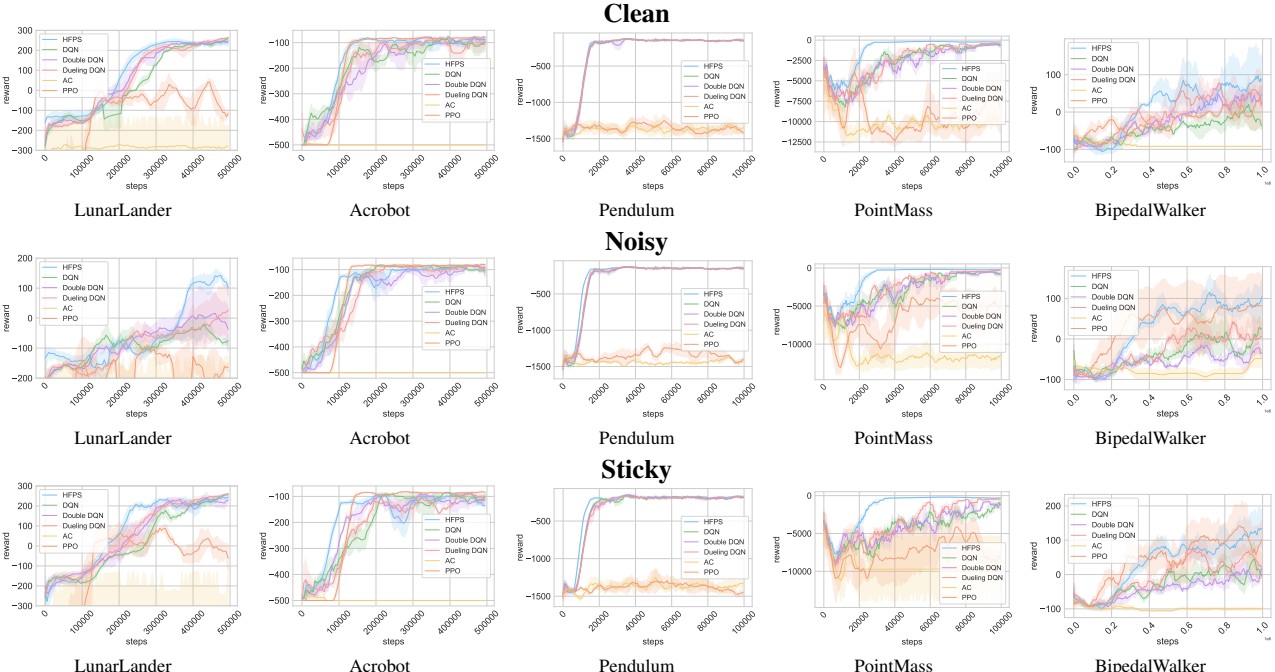

*Figure 4.* **Episode returns.** Return versus environment steps (mean $\pm$ one standard deviation over seeds) for HFPS and baselines on five control tasks (columns) under three observation regimes (rows): Clean, Noisy, and Sticky.

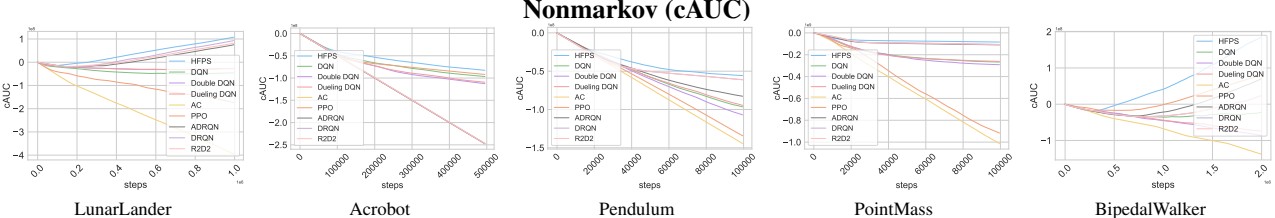

*Figure 5.* **Cumulative AUC trajectories** Each panel plots the cumulative integral of the corresponding return curve in Figure 2. Higher curves indicate better overall sample efficiency under the same training budget.

the interpretation that the residual contains both structural mismatch and finite-sample stochasticity, and that the benefit of HFPS is most pronounced when the mismatch has persistent structure rather than being purely i.i.d. corruption.

### C.4. Representation Ablation under Nonmarkov Observations

To clarify the role of no-memory and memory-augmented representations, we run a small ablation on BipedalWalker (Nonmarkov). We compare raw observations, a four-step history stack, and a GRU-based memory encoder. The residual norm is computed without HFPS and normalized by the raw-observation setting. Richer representations reduce the residual, but the residual remains nonzero, and HFPS continues to improve AUC@T and Final@T on top of all three representations.

### C.5. Modular Use with PPO/GAE

HFPS can be used as a modular correction at the advantage-estimation level. Given the one-step TD residual $\delta_t = r_t + \gamma V(s_{t+1}) - V(s_t)$, we fit $u_\phi$ so that $u_\phi(s_{t+1}) - \gamma u_\phi(s_t) \approx \delta_t$ and define

$$\tilde{\delta}_t = (1 - \lambda_{\mathrm{h}})\delta_t + \lambda_{\mathrm{h}}\big(u_\phi(s_{t+1}) - \gamma u_\phi(s_t)\big), \tag{24}$$

where $\lambda_{\mathrm{h}} \in [0, 1]$ controls the amount of Hodge filtering. Generalized Advantage Estimation is then computed from $\tilde{\delta}_t$, while the PPO clipping objective is left unchanged. Table 4 shows a plug-in check on BipedalWalker (Nonmarkov).

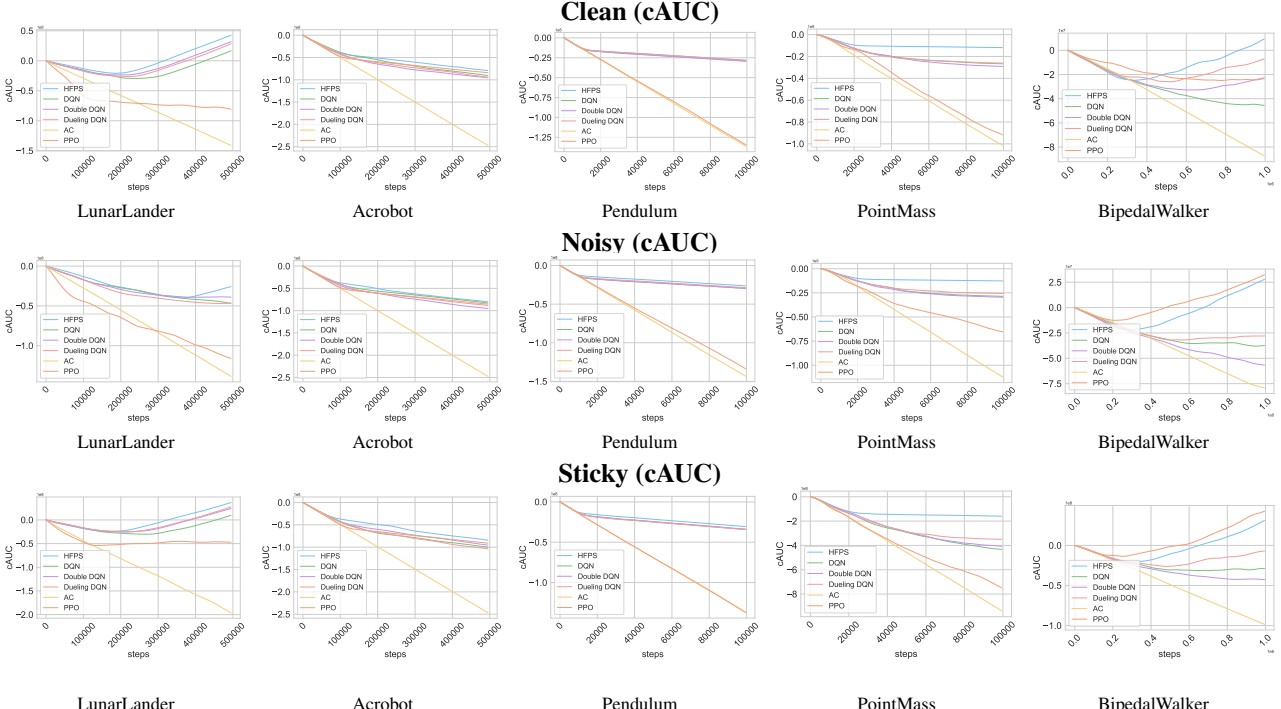

*Figure 6.* **Cumulative AUC trajectories** Each panel plots the cumulative integral of the corresponding return curve in Figure 4. Higher curves indicate better overall sample efficiency under the same training budget.

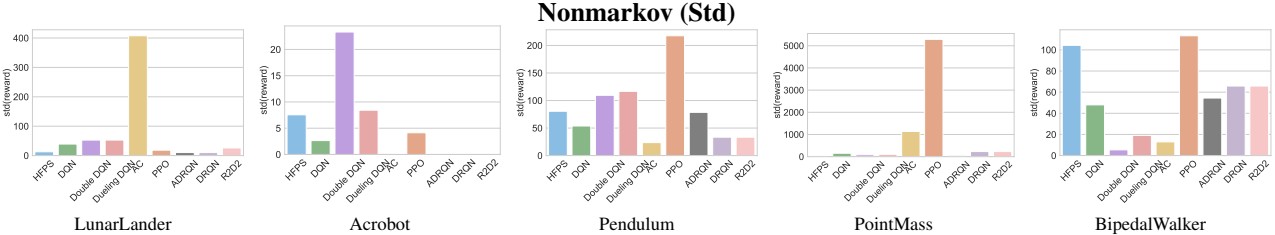

*Figure 7.* **Across-seed variability** Standard deviation of episode returns across seeds at each evaluation step, computed from the same runs as Figure 2.

### C.6. Wall-Clock Overhead

HFPS adds one potential network and an auxiliary projection loss, but it does not add environment interaction, trajectory collection, recurrent belief-state inference, or longer rollouts. Under identical hardware and environment steps on CartPole with five seeds and 100,000 steps, the overhead is about $1.27\times$.

## D. Proof

### D.1. Proof of Lemma 2.5

*Proof.* We need to establish three facts: (1) $d$ is linear; (2) there exists a constant $c > 0$, depending only on $\gamma$, such that Eqn. 9 holds; (3) consequently, $d$ is a bounded linear operator, and therefore admits a unique Hilbert adjoint $d^* : C^1 \to C^0$.

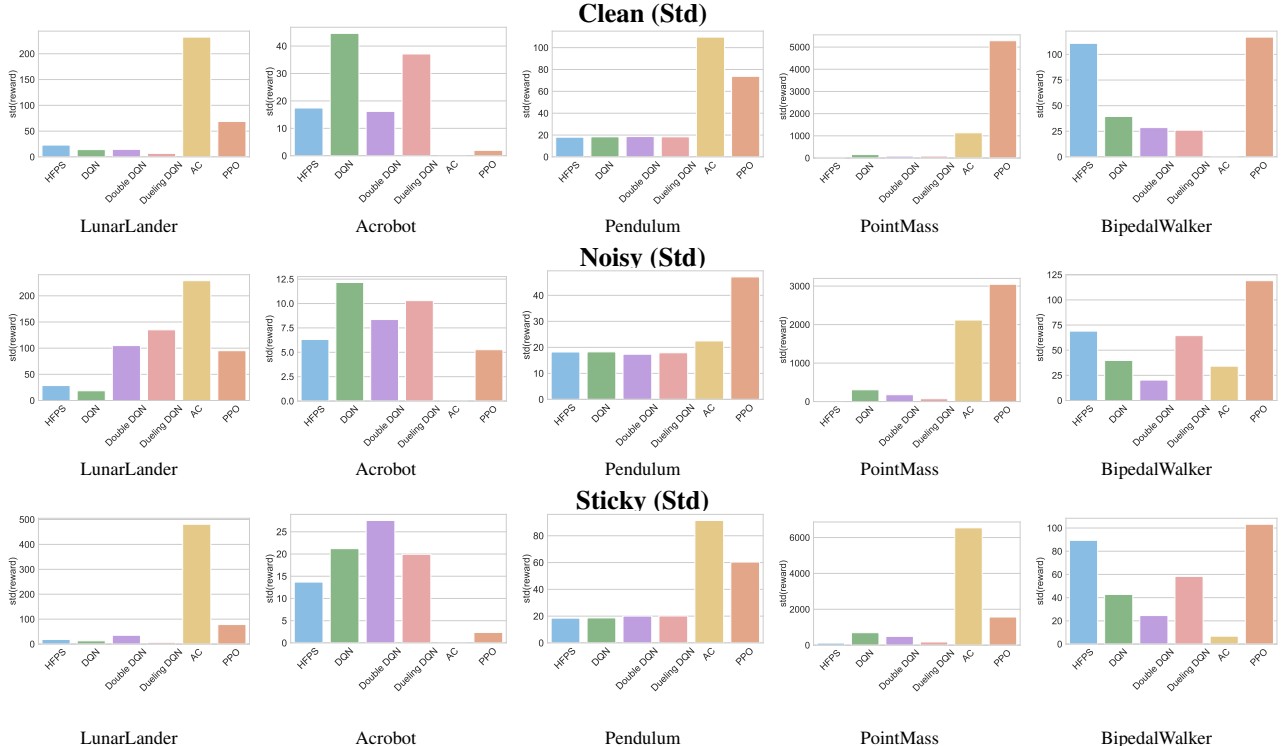

*Figure 8.* **Across-seed variability** Standard deviation of episode returns across seeds at each evaluation step, computed from the same runs as Figure 4.

First, we establish the linearity of $d$. Take any $u, v \in C^0$ and any scalars $\alpha, \beta \in \mathbb{R}$. By Definition 2.3,

$$
\begin{aligned}
d(\alpha u + \beta v)(s, a, s') &= (\alpha u + \beta v)(s') - \gamma(\alpha u + \beta v)(s). \\
&= \alpha u(s') + \beta v(s') - \gamma \alpha u(s) - \gamma \beta v(s). \\
&= \alpha(u(s') - \gamma u(s)) + \beta(v(s') - \gamma v(s)). \\
&= \alpha(du)(s, a, s') + \beta(dv)(s, a, s').
\end{aligned}
\tag{25}
$$

Hence, for any $(s, a, s')$,

$$
d(\alpha u + \beta v) = \alpha du + \beta dv
\tag{26}
$$

Therefore, $d$ is linear.

Next, we establish the boundedness of $d$. We aim to show that there exists a constant $c(\gamma) > 0$ such that for all $u \in C^0$,

$$
||du||_{C^1}^2 \le c(\gamma)||u||_{C^0}^2
\tag{27}
$$

By the definition of the norm on $C^1$ and Definition 2.3,

$$
\begin{aligned}
||du||_{C^1}^2 &= \int \left((du)(s, a, s')\right)^2 \mu_\pi(ds, da, ds') \\
&= \int \left(u(s') - \gamma u(s)\right)^2 \mu_\pi(ds, da, ds') \\
&= \mathbb{E}_{\mu_\pi}\left[(u(S') - \gamma u(S))^2\right] \\
&\le \mathbb{E}_{\mu_\pi}\left[2u(S')^2 + 2\gamma^2 u(S)^2\right] \\
&= 2\mathbb{E}_{\mu_\pi}\left[u(S')^2\right] + 2\gamma^2 \mathbb{E}_{\mu_\pi}\left[u(S)^2\right]
\end{aligned}
\tag{28}
$$

We now relate both $\mathbb{E} * \mu * \pi[u(S')^2]$ and $\mathbb{E} * \mu * \pi[u(S)^2]$ to $|u|_{C^0}^2$.

*Table 2.* **Aggregate performance from learning curves.** AUC@T is trapezoidal area under return vs. steps; Final@T is the last return. Values are mean ± std over seeds. Best baseline is chosen per-row by AUC@T.

| Env | Regime | HFPS AUC@T | Best baseline AUC@T | Δ% | HFPS Final@T | Best baseline Final@T | Δ% |
|---|---|---|---|---|---|---|---|
| acrobot | Clean | -79144263.52 ± 5715493.23 | PPO: -84031333.33 ± 3465563.82 | 5.8 | -87.47 ± 23.53 | PPO: -78.20 ± 3.79 | -11.9 |
| acrobot | Noisy | -80111477.55 ± 3107189.24 | DQN: -82873863.89 ± 7286738.55 | 3.3 | -93.94 ± 8.99 | DQN: -110.55 ± 23.58 | 15.0 |
| acrobot | Sticky | -84528355.10 ± 2401536.91 | PPO: -90858950.00 ± 2046464.24 | 7.0 | -136.06 ± 10.85 | PPO: -84.90 ± 3.29 | -60.3 |
| bipedalwalker | Clean | 9392950.11 ± 56198564.79 | Dueling DQN: -7376062.16 ± 14291366.48 | 227.3 | 84.49 ± 103.95 | Dueling DQN: 20.61 ± 65.80 | 310.0 |
| bipedalwalker | Noisy | 27904259.81 ± 25304952.63 | PPO: 32301685.83 ± 109002499.36 | -13.6 | 103.64 ± 75.44 | PPO: 81.24 ± 135.36 | 27.6 |
| bipedalwalker | Sticky | 31227454.58 ± 44656706.21 | PPO: 42687915.83 ± 67215454.11 | -26.8 | 132.50 ± 106.13 | PPO: 101.74 ± 106.39 | 30.2 |
| lunarlander | Clean | 42642395.61 ± 3523795.18 | Double DQN: 31889842.37 ± 5286287.68 | 33.7 | 238.91 ± 29.50 | Double DQN: 246.91 ± 16.69 | -3.2 |
| lunarlander | Noisy | -25766227.01 ± 5217416.45 | Double DQN: -39000346.31 ± 18276224.25 | 33.9 | 100.88 ± 43.37 | Double DQN: -36.76 ± 127.95 | 374.4 |
| lunarlander | Sticky | 36783880.67 ± 9580504.79 | Dueling DQN: 27439517.04 ± 9577045.07 | 34.1 | 244.08 ± 11.48 | Dueling DQN: 261.87 ± 3.92 | -6.8 |
| pendulum_discrete | Clean | -28087189.02 ± 621451.05 | Dueling DQN: -28351509.26 ± 1566650.62 | 0.9 | -147.96 ± 23.70 | Dueling DQN: -147.34 ± 24.18 | -0.4 |
| pendulum_discrete | Noisy | -26282013.77 ± 614956.47 | Dueling DQN: -28625597.53 ± 1233343.10 | 8.2 | -147.22 ± 21.88 | Dueling DQN: -147.56 ± 24.05 | 0.2 |
| pendulum_discrete | Sticky | -30641411.50 ± 353162.29 | DQN: -33778858.79 ± 616061.45 | 9.3 | -184.96 ± 19.62 | DQN: -178.67 ± 15.81 | -3.5 |
| pointmass | Clean | -118875940.90 ± 14470746.43 | Dueling DQN: -260948236.73 ± 19220437.10 | 54.4 | -207.14 ± 49.83 | Dueling DQN: -433.00 ± 127.73 | 52.2 |
| pointmass | Noisy | -126442915.13 ± 26121291.16 | Dueling DQN: -256558177.38 ± 24937755.63 | 50.7 | -230.07 ± 16.41 | Dueling DQN: -329.73 ± 72.07 | 30.2 |
| pointmass | Sticky | -159622041.44 ± 7500560.61 | Dueling DQN: -350365504.29 ± 9380735.89 | 54.4 | -296.23 ± 69.91 | Dueling DQN: -399.89 ± 86.85 | 25.9 |

*Table 3.* Representation ablation on BipedalWalker (Nonmarkov).

| Representation | Residual | Base AUC@T | +HFPS AUC@T | Base Final@T | +HFPS Final@T |
|---|---|---|---|---|---|
| Raw observation $s_t$ | 0.97 | 79.3 | 172.4 | 68.5 | 141.2 |
| 4-step history stack | 0.62 | 119.1 | 182.3 | 97.3 | 164.8 |
| GRU / memory encoder | 0.39 | 161.7 | 209.6 | 116.4 | 170.1 |

We first handle the term $\mathbb{E}_{\mu_\pi}\left[u(S)^2\right]$. Since $\nu_\pi$ is the marginal of $\mu_\pi$ on the first coordinate, we have $\nu_\pi(C) = \mu_\pi\left(C \times \mathcal{A} \times \mathcal{S}\right)$. Thus, for any integrable function $g : S \to \mathbb{R}$,

$$\int_S g(s)\nu_\pi(ds) = \int_{S \times A \times S} g(s)\mu_\pi(ds, da, ds'). \tag{29}$$

Substituting $g(s) = u(s)^2$, we obtain

$$\begin{aligned}\int_S u(s)^2\nu_\pi(ds) &= \int u(s)^2\mu_\pi(ds, da, ds') \\ &= \mathbb{E}_{\mu_\pi}\left[u(S)^2\right] = ||u||_{C^0}^2. \end{aligned} \tag{30}$$

We now handle the term $\mathbb{E}_{\mu_\pi}\left[u(S')^2\right]$. Here we make use of the trajectory-based definition of $\mu_\pi$. By the definition of the discounted triplet occupancy measure (Definition 2.1), for any bounded measurable function $\phi : S \times A \times S \to \mathbb{R}$,

$$\int \phi(s, a, s')\mu_\pi(ds, da, ds') = (1 - \gamma)\mathbb{E}_\pi\left[\sum_{t=0}^\infty \gamma^t \phi(S_t, A_t, S_{t+1})\right] \tag{31}$$

Taking $\phi(s, a, s') = u(s')^2$, we obtain

$$\mathbb{E}_{\mu_\pi}\left[u(S')^2\right] = (1 - \gamma)\mathbb{E}_\pi\left[\sum_{t=0}^\infty \gamma^t u(S_{t+1})^2\right] \tag{32}$$

Letting $k = t + 1$,

$$\begin{aligned}\mathbb{E}_{\mu_\pi}\left[u(S')^2\right] &= (1 - \gamma)\mathbb{E}_\pi\left[\sum_{t=0}^\infty \gamma^t u(S_{t+1})^2\right] \\ &= (1 - \gamma)\mathbb{E}_\pi\left[\sum_{k=1}^\infty \gamma^{k-1} u(S_k)^2\right] \\ &= \frac{(1 - \gamma)}{\gamma}\mathbb{E}_\pi\left[\sum_{k=1}^\infty \gamma^k u(S_k)^2\right] \end{aligned} \tag{33}$$

*Table 4.* PPO with Hodge-filtered GAE on BipedalWalker (Nonmarkov).

| Method | AUC@T | Final@T |
|---|---|---|
| PPO | $78.5 \pm 155.1$ | $56.4 \pm 142.9$ |
| PPO+HFPS | $181.0 \pm 40.1$ | $154.1 \pm 35.6$ |
| HFPS | $184.7 \pm 96.1$ | $146.0 \pm 113.9$ |

*Table 5.* Wall-clock comparison under identical environment steps and hardware.

| Method | Task | Seeds | Steps | Time / overhead |
|---|---|---|---|---|
| DQN | CartPole | 5 | 100,000 | $12.34 \pm 0.56s / 1.00\times$ |
| HFPS | CartPole | 5 | 100,000 | $15.67 \pm 0.44s / 1.27\times$ |

On the other hand, the squared norm $||u||^2_{C^0}$ can also be written in trajectory form:

$$
\begin{aligned}
||u||^2_{C^0} &= \int_S u(s)^2 \nu_\pi(ds) \\
&= (1-\gamma)\mathbb{E}_\pi \left[ \sum_{t=0}^\infty \gamma^t u(S_t)^2 \right] \\
&= (1-\gamma)\mathbb{E}_\pi \left[ \gamma^0 u(S_0)^2 + \sum_{t=1}^\infty \gamma^t u(S_t)^2 \right] \\
&= (1-\gamma)\mathbb{E}_\pi \left[ u(S_0)^2 \right] + (1-\gamma)\mathbb{E}_\pi \left[ \sum_{t=1}^\infty \gamma^t u(S_t)^2 \right]
\end{aligned}
\tag{34}
$$

So, we can get:

$$
(1-\gamma)\mathbb{E}_\pi \left[ \sum_{t=1}^\infty \gamma^t u(S_t)^2 \right] = ||u||^2_{C^0} - (1-\gamma)\mathbb{E}_\pi \left[ u(S_0)^2 \right].
\tag{35}
$$

Substituting this back into the expression for $\mathbb{E}_{\mu_\pi} \left[ u(S')^2 \right]$

$$
\mathbb{E}_{\mu_\pi} \left[ u(S')^2 \right] = \frac{1}{\gamma} \left( ||u||^2_{C^0} - (1-\gamma)\mathbb{E}_\pi \left[ u(S_0)^2 \right] \right)
\tag{36}
$$

Since the second term on the right-hand side is nonnegative,

$$
(1-\gamma)\mathbb{E}_\pi \left[ u(S_0)^2 \right] \geq 0
\tag{37}
$$

we obtain the following upper bound, which depends only on $\gamma$

$$
\mathbb{E}_{\mu_\pi} \left[ u(S')^2 \right] \leq \frac{1}{\gamma} ||u||^2_{C^0}
\tag{38}
$$

Combining the two bounds above, we obtain the final constant. Substituting the relations derived earlier,

$$
\begin{aligned}
||du||^2_{C^1} &\leq 2\mathbb{E}_{\mu_\pi} \left[ u(S')^2 \right] + 2\gamma^2 \mathbb{E}_{\mu_\pi} \left[ u(S)^2 \right] \\
&\leq 2 \cdot \frac{1}{\gamma} ||u||^2_{C^0} + 2\gamma^2 \cdot ||u||^2_{C^0} \\
&= 2(\frac{1}{\gamma} + \gamma^2)||u||^2_{C^0}
\end{aligned}
\tag{39}
$$

So $c(\gamma) = 2(\frac{1}{\gamma} + \gamma^2)$

In particular, this shows that if $||u||_{C^0} < \infty$, then the right-hand side is finite, hence $||du||_{C^1} < \infty$, meaning $du \in C^1$ Moreover, $||du||_{C^1} \leq \sqrt{c(\gamma)}||u||_{C^0}$, so $d$ is indeed a bounded linear operator.

Finally, we establish the existence and uniqueness of the Hilbert adjoint $d^*$. Since $d : C^0 \to C^1$ is a bounded linear operator and both $C^0, C^1$ are Hilbert spaces, the standard theorem for Hilbert spaces (a consequence of the Riesz Representation Theorem) states:

If $T : K \to K$ is a bounded linear operator between Hilbert spaces, then there exists a unique bounded linear operator $T* : K \to K$ such that

$$\langle Tx, y \rangle_K = \langle x, T^*y \rangle_H, \forall x \in H, y \in K \tag{40}$$

Applying this with $H = C^0, K = C^1, T = d$, we obtain the existence of a unique operator

$$d^* : C^1 \to C^0 \tag{41}$$

such that

$$\langle du, f \rangle_{C^1} = \langle u, d^*f \rangle_{C^0}, \forall u \in C^0, f \in C^1 \tag{42}$$

$\square$

## D.2. Proof of Theorem 3.4

*Proof.* We divide the proof into several steps.

**Basic structure of $\overline{\mathcal{E}}$ and $\overline{\mathcal{E}}^\perp$**

Recall that $C^1$ is a Hilbert space with inner product $\langle \cdot, \cdot \rangle_{C^1}$. By definition,

$$\mathcal{E}_0 := \mathrm{im}(d) = \{du : u \in C^0\} \subseteq C^1, \tag{43}$$

and $\overline{\mathcal{E}} := \overline{\mathcal{E}_0}$ is the closure of $\mathcal{E}_0$ in $C^1$.

First, note that $\mathcal{E}_0$ is a linear subspace of $C^1$: for any $u, v \in C^0$ and any scalars $\alpha, \beta \in \mathbb{R}$,

$$\alpha(du) + \beta(dv) = d(\alpha u + \beta v), \tag{44}$$

by linearity of $d$. Hence $\mathcal{E}_0$ is closed under linear combinations, and thus it is a linear subspace. Because the closure of a linear subspace is again a linear subspace, $\overline{\mathcal{E}}$ is a closed linear subspace of $C^1$.

By general Hilbert space theory, the orthogonal complement

$$\overline{\mathcal{E}}^\perp := \{g \in C^1 : \langle g, h \rangle_{C^1} = 0, \ \forall h \in \overline{\mathcal{E}}\} \tag{45}$$

is itself a closed linear subspace of $C^1$, and we have the orthogonal direct sum decomposition

$$C^1 = \overline{\mathcal{E}} \oplus \overline{\mathcal{E}}^\perp, \tag{46}$$

meaning that every element of $C^1$ can be written as a sum of an element in $\overline{\mathcal{E}}$ and an element in $\overline{\mathcal{E}}^\perp$, with these two components orthogonal and uniquely determined. For completeness, we briefly recall this decomposition below in Step as a consequence of the Hilbert projection theorem.

### Orthogonal projection and the basic decomposition.

Let $H$ be a Hilbert space and let $M \subseteq H$ be a nonempty closed linear subspace. The Hilbert projection theorem states that for every $x \in H$, there exists a unique $m^* \in M$ such that

$$\|x - m^*\|_H = \inf_{m \in M} \|x - m\|_H. \tag{47}$$

Moreover, if we set $r := x - m^*$, then $r \in M^\perp$ and the decomposition

$$x = m^* + r, \quad m^* \in M, \ r \in M^\perp \tag{48}$$

is unique.

We apply this theorem to the present case with

$$H = C^1, \quad M = \overline{\mathcal{E}}. \tag{49}$$

Because $\overline{\mathcal{E}}$ is a nonempty closed linear subspace of $C^1$, the projection theorem applies. Thus, for any $f \in C^1$, there exists a unique

$$f_{\text{ex}} \in \overline{\mathcal{E}} \tag{50}$$

such that

$$\|f - f_{\text{ex}}\|_{C^1} = \inf_{g \in \overline{\mathcal{E}}} \|f - g\|_{C^1}. \tag{51}$$

If we define

$$f_{\text{res}} := f - f_{\text{ex}}, \tag{52}$$

then $f_{\text{res}} \in \overline{\mathcal{E}}^{\perp}$ and

$$f = f_{\text{ex}} + f_{\text{res}}, \quad f_{\text{ex}} \in \overline{\mathcal{E}}, \quad f_{\text{res}} \in \overline{\mathcal{E}}^{\perp}. \tag{53}$$

By construction,

$$\langle f_{\text{ex}}, f_{\text{res}} \rangle_{C^1} = 0. \tag{54}$$

*Uniqueness* of this decomposition also follows from the projection theorem: if we had another decomposition

$$f = g_1 + g_2, \quad g_1 \in \overline{\mathcal{E}}, \ g_2 \in \overline{\mathcal{E}}^{\perp}, \tag{55}$$

then

$$f_{\text{ex}} - g_1 = g_2 - f_{\text{res}}, \tag{56}$$

where the left-hand side belongs to $\overline{\mathcal{E}}$ and the right-hand side belongs to $\overline{\mathcal{E}}^{\perp}$. Hence

$$f_{\text{ex}} - g_1 \in \overline{\mathcal{E}} \cap \overline{\mathcal{E}}^{\perp}. \tag{57}$$

But the only vector in both a subspace and its orthogonal complement is the zero vector, so

$$f_{\text{ex}} - g_1 = 0 \quad \Rightarrow \quad f_{\text{ex}} = g_1 \tag{58}$$

and similarly $f_{\text{res}} = g_2$. This proves the uniqueness of the decomposition.

**Variational characterization of $f_{\text{ex}}$.**

By definition of the orthogonal projection of $f$ onto $\overline{\mathcal{E}}$, we already have

$$f_{\text{ex}} = \arg\min_{g \in \overline{\mathcal{E}}} \|f - g\|_{C^1}^2. \tag{59}$$

This is precisely the variational characterization stated in the theorem: $f_{\text{ex}} = \arg\min_{g \in \overline{\mathcal{E}}} \|f - g\|_{C^1}^2$.

It remains to show that this $f_{\text{ex}}$ is characterized equivalently by the orthogonality condition

$$\langle f - f_{\text{ex}}, du \rangle_{C^1} = 0, \quad \forall u \in C^0. \tag{60}$$

Since $f_{\text{res}} := f - f_{\text{ex}} \in \overline{\mathcal{E}}^{\perp}$, we have

$$\langle f_{\text{res}}, h \rangle_{C^1} = 0, \quad \forall h \in \overline{\mathcal{E}}. \tag{61}$$

In particular, $\mathcal{E}_0 = \text{im}(d) \subseteq \overline{\mathcal{E}}$, so for any $u \in C^0$,

$$du \in \mathcal{E}_0 \subseteq \overline{\mathcal{E}}, \tag{62}$$

and thus

$$\langle f_{\text{res}}, du \rangle_{C^1} = 0 \quad \Longleftrightarrow \quad \langle f - f_{\text{ex}}, du \rangle_{C^1} = 0, \quad \forall u \in C^0. \tag{63}$$

This establishes the orthogonality condition (60) for $f_{\text{ex}}$.

Conversely, suppose that for some $g \in \overline{\mathcal{E}}$ we have

$$\langle f - g, du \rangle_{C^1} = 0, \quad \forall u \in C^0. \tag{64}$$

Then $f - g$ is orthogonal to all elements in $\mathcal{E}_0 = \mathrm{im}(d)$. Since $\overline{\mathcal{E}}$ is the closure of $\mathcal{E}_0$, and the inner product is continuous, this implies that $f - g$ is orthogonal to all of $\overline{\mathcal{E}}$. Therefore

$$f - g \in \overline{\mathcal{E}}^{\perp}. \tag{65}$$

So we have a decomposition

$$f = g + (f - g), \quad g \in \overline{\mathcal{E}}, \ f - g \in \overline{\mathcal{E}}^{\perp}, \tag{66}$$

which must coincide with the unique decomposition from above. Hence $g = f_{\mathrm{ex}}$. Therefore (60) is indeed an equivalent characterization of the projection $f_{\mathrm{ex}}$.

**The case where $\mathrm{im}(d)$ is closed and representation $f_{\mathrm{ex}} = du^*$.**

Now assume in addition that $\mathcal{E}_0 = \mathrm{im}(d)$ is closed in $C^1$. Then

$$\overline{\mathcal{E}} = \overline{\mathcal{E}_0} = \mathcal{E}_0. \tag{67}$$

Thus $\overline{\mathcal{E}}$ coincides with the actual range of $d$.

In this case, for any $f \in C^1$, the component $f_{\mathrm{ex}}$ of the decomposition lies in $\overline{\mathcal{E}} = \mathcal{E}_0$, so by definition of $\mathcal{E}_0$ there exists at least one $u^* \in C^0$ such that

$$f_{\mathrm{ex}} = du^*. \tag{68}$$

Define the functional

$$J(u) := \|f - du\|_{C^1}^2, \quad u \in C^0. \tag{69}$$

Because $f_{\mathrm{ex}}$ is the orthogonal projection of $f$ onto $\mathcal{E}_0 = \mathrm{im}(d)$, we have

$$\|f - f_{\mathrm{ex}}\|_{C^1} = \inf_{g \in \mathcal{E}_0} \|f - g\|_{C^1}. \tag{70}$$

But any $g \in \mathcal{E}_0$ can be written as $g = du$ for some $u \in C^0$, so

$$\inf_{g \in \mathcal{E}_0} \|f - g\|_{C^1}^2 = \inf_{u \in C^0} \|f - du\|_{C^1}^2 = \inf_{u \in C^0} J(u). \tag{71}$$

Since $f_{\mathrm{ex}} = du^*$, we obtain

$$J(u^*) = \|f - du^*\|_{C^1}^2 = \|f - f_{\mathrm{ex}}\|_{C^1}^2 = \min_{u \in C^0} J(u). \tag{72}$$

So $u^*$ is a minimizer of the quadratic functional $J$.

**Characterization of all minimizers and the relation $\ker(\Delta_0) = \ker(d)$.**

Recall that $\Delta_0 := d^* d : C^0 \to C^0$ is the Hodge Laplacian on $C^0$. Since $d$ is a bounded linear operator between Hilbert spaces, its adjoint $d^*$ exists and is bounded; hence $\Delta_0 = d^* d$ is also a bounded linear operator on $C^0$.

We first show that

$$\ker(\Delta_0) = \ker(d). \tag{73}$$

*(i)* $\ker(d) \subseteq \ker(\Delta_0)$. If $u \in \ker(d)$, i.e., $du = 0$, then

$$\Delta_0 u = d^* du = d^* 0 = 0, \tag{74}$$

so $u \in \ker(\Delta_0)$. Thus $\ker(d) \subseteq \ker(\Delta_0)$.

*(ii)* $\ker(\Delta_0) \subseteq \ker(d)$. Conversely, let $u \in \ker(\Delta_0)$, i.e.,

$$\Delta_0 u = d^* du = 0. \tag{75}$$

Take the inner product with $u$ in $C^0$:

$$\langle \Delta_0 u, u \rangle_{C^0} = \langle d^* du, u \rangle_{C^0}. \tag{76}$$

By the definition of the adjoint operator,

$$\langle d^* du, u \rangle_{C^0} = \langle du, du \rangle_{C^1} = \|du\|_{C^1}^2. \tag{77}$$

Since $\Delta_0 u = 0$, we have

$$0 = \langle \Delta_0 u, u \rangle_{C^0} = \|du\|_{C^1}^2. \tag{78}$$

The norm $\|\cdot\|_{C^1}$ is nonnegative and vanishes only for the zero element. Hence

$$\|du\|_{C^1}^2 = 0 \quad \Rightarrow \quad du = 0. \tag{79}$$

This shows $u \in \ker(d)$. Therefore $\ker(\Delta_0) \subseteq \ker(d)$.

Combining (i) and (ii), we obtain

$$\ker(\Delta_0) = \ker(d). \tag{80}$$

Now we describe all minimizers of $J$. Let $u^* \in C^0$ be one minimizer, so that $f_{\text{ex}} = du^*$ and $J(u^*) = \min_u J(u)$.

Take any $v \in \ker(d)$. Then

$$d(u^* + v) = du^* + dv = du^* + 0 = du^*. \tag{81}$$

Thus

$$J(u^+) := J(u^* + v) = \|f - d(u^* + v)\|_{C^1}^2 = \|f - du^*\|_{C^1}^2 = J(u^*). \tag{82}$$

So every $u^* + v$ with $v \in \ker(d) = \ker(\Delta_0)$ is also a minimizer of $J$.

Conversely, suppose $\tilde{u} \in C^0$ is another minimizer of $J$. Then

$$\|f - d\tilde{u}\|_{C^1} = \|f - du^*\|_{C^1} = \inf_{u \in C^0} \|f - du\|_{C^1}. \tag{83}$$

So both $du^*$ and $d\tilde{u}$ are best approximations to $f$ in the subspace $\mathcal{E}_0 = \text{im}(d)$. But the best approximation in a closed convex subset (in particular, a closed linear subspace) of a Hilbert space is unique. Hence

$$du^* = d\tilde{u}. \tag{84}$$

This shows that

$$d(\tilde{u} - u^*) = 0, \tag{85}$$

so $\tilde{u} - u^* \in \ker(d) = \ker(\Delta_0)$.

Therefore, all minimizers of $J$ are exactly those elements of the form

$$u^* + v, \quad v \in \ker(d) = \ker(\Delta_0), \tag{86}$$

and these are the only degrees of freedom in the choice of minimizer.

We have shown:

- Every $f \in C^1$ admits a unique decomposition

$$f = f_{\text{ex}} + f_{\text{res}}, \quad f_{\text{ex}} \in \overline{\mathcal{E}}, \ f_{\text{res}} \in \overline{\mathcal{E}}^\perp, \ \langle f_{\text{ex}}, f_{\text{res}} \rangle_{C^1} = 0. \tag{87}$$

- The component $f_{\text{ex}}$ is the orthogonal projection of $f$ onto $\overline{\mathcal{E}}$, equivalently characterized by

$$f_{\text{ex}} = \arg\min_{g \in \overline{\mathcal{E}}} \|f - g\|_{C^1}^2, \quad \text{and} \quad \langle f - f_{\text{ex}}, du \rangle_{C^1} = 0, \ \forall u \in C^0. \tag{88}$$

- If the range of $d$ is closed, then $\overline{\mathcal{E}} = \mathcal{E}_0 = \text{im}(d)$, so there exists $u^* \in C^0$ with $f_{\text{ex}} = du^*$. This $u^*$ minimizes

$$J(u) = \|f - du\|_{C^1}^2, \tag{89}$$

  and all minimizers differ by an element of $\ker(d) = \ker(\Delta_0)$.

This completes the proof.

$\square$

### D.3. Proof of Corollary 3.5

*Proof.* We assume throughout that the range $\mathcal{E}_0 = \mathrm{im}(d)$ is closed in $C^1$, so that

$$\overline{\mathcal{E}} = \overline{\mathcal{E}_0} = \mathcal{E}_0. \tag{90}$$

Let $V : \mathcal{S} \to \mathbb{R}$ be a bounded value function such that its TD error

$$\delta_V \in C^1 = L^2(S \times A \times S, \mu_\pi). \tag{91}$$

We apply Theorem 3.4 with $f = \delta_V$. Since $\delta_V \in C^1$, the theorem guarantees that there exist

$$\delta_V^{\mathrm{ex}} \in \overline{\mathcal{E}}, \qquad \delta_V^{\mathrm{res}} \in \overline{\mathcal{E}}^{\perp} \tag{92}$$

such that

$$\delta_V = \delta_V^{\mathrm{ex}} + \delta_V^{\mathrm{res}}, \tag{93}$$

and

$$\langle \delta_V^{\mathrm{ex}}, \delta_V^{\mathrm{res}} \rangle_{C^1} = 0. \tag{94}$$

Moreover, by the variational characterization in Theorem 3.4, $\delta_V^{\mathrm{ex}}$ is the unique orthogonal projection of $\delta_V$ onto $\overline{\mathcal{E}}$, i.e.

$$\delta_V^{\mathrm{ex}} = \arg\min_{g \in \overline{\mathcal{E}}} \|\delta_V - g\|_{C^1}^2, \tag{95}$$

and it satisfies the orthogonality condition

$$\langle \delta_V - \delta_V^{\mathrm{ex}}, du \rangle_{C^1} = 0, \quad \forall\, u \in C^0. \tag{96}$$

Because we assume that the range of $d$ is closed, we have

$$\overline{\mathcal{E}} = \mathcal{E}_0 = \mathrm{im}(d). \tag{97}$$

By definition of the image, every element of $\mathcal{E}_0$ is of the form $du$ for some $u \in C^0$. In particular, $\delta_V^{\mathrm{ex}} \in \mathcal{E}_0$ implies that there exists at least one $u_V^* \in C^0$ such that

$$\delta_V^{\mathrm{ex}} = du_V^*. \tag{98}$$

Using the notation in the corollary, we now *define*

$$\delta_V^{\mathrm{res}} := \delta_V^{\mathrm{res}} \tag{99}$$

from (93), and we rename $\delta_V^{\mathrm{ex}}$ as $du_V^*$ via (98). Then the decomposition (93) can be rewritten as

$$\delta_V = du_V^* + \delta_V^{\mathrm{res}}, \tag{100}$$

with

$$du_V^* \in \mathcal{E}_0 = \overline{\mathcal{E}}, \qquad \delta_V^{\mathrm{res}} \in \overline{\mathcal{E}}^{\perp}. \tag{101}$$

Since the decomposition $(\delta_V^{\mathrm{ex}}, \delta_V^{\mathrm{res}})$ in Theorem 3.4 is unique, the residual $\delta_V^{\mathrm{res}}$ is uniquely determined for the given $\delta_V$, which proves the uniqueness statement for the residual component in the corollary.

Next we show the variational characterization of $u_V^*$. Define the quadratic functional

$$J(u) := \|\delta_V - du\|_{C^1}^2, \qquad u \in C^0. \tag{102}$$

Because $\overline{\mathcal{E}} = \mathcal{E}_0 = \mathrm{im}(d)$, any $g \in \overline{\mathcal{E}}$ can be written as $g = du$ for some $u \in C^0$, and conversely any $u \in C^0$ yields an element $du \in \overline{\mathcal{E}}$. Therefore

$$\inf_{g \in \overline{\mathcal{E}}} \|\delta_V - g\|_{C^1}^2 = \inf_{u \in C^0} \|\delta_V - du\|_{C^1}^2 = \inf_{u \in C^0} J(u). \tag{103}$$

By Theorem 3.4, the element $\delta_V^{\mathrm{ex}}$ is the unique minimizer of the left-hand side:

$$\delta_V^{\mathrm{ex}} = \arg\min_{g \in \overline{\mathcal{E}}} \|\delta_V - g\|_{C^1}^2. \tag{104}$$

Since $\delta_V^{\mathrm{ex}} = du_V^*$, we obtain

$$J(u_V^*) = \|\delta_V - du_V^*\|_{C^1}^2 = \|\delta_V - \delta_V^{\mathrm{ex}}\|_{C^1}^2 = \min_{u \in C^0} \|\delta_V - du\|_{C^1}^2. \tag{105}$$

Equivalently,

$$u_V^* \in \arg\min_{u \in C^0} \|\delta_V - du\|_{C^1}^2. \tag{106}$$

This proves the optimization characterization stated in the corollary.

Finally, we explain the orthogonality interpretation of the residual. From (96), with $f = \delta_V$ and $\delta_V^{\mathrm{ex}} = du_V^*$, we have

$$\langle \delta_V - du_V^*, du \rangle_{C^1} = 0, \quad \forall\, u \in C^0. \tag{107}$$

But $\delta_V - du_V^* = \delta_V^{\mathrm{res}}$, so this can be rewritten as

$$\langle \delta_V^{\mathrm{res}}, du \rangle_{C^1} = 0, \quad \forall\, u \in C^0. \tag{108}$$

In other words, $\delta_V^{\mathrm{res}}$ is orthogonal to all exact 1-cochains of the form $du$, i.e. to all discounted gradients generated by potential functions $u \in C^0$. This shows that $\delta_V^{\mathrm{res}}$ is precisely the component of the TD error $\delta_V$ that cannot be represented as a discounted gradient $du$ and thus cannot be attributed to any potential function, which matches the "topological residual" interpretation in the statement.

Collecting all parts, we have shown:

- there exists $u_V^* \in C^0$ and a unique $\delta_V^{\mathrm{res}} \in \overline{\mathcal{E}}^\perp$ such that $\delta_V = du_V^* + \delta_V^{\mathrm{res}}$;

- $u_V^*$ is a minimizer of $J(u) = \|\delta_V - du\|_{C^1}^2$ over $u \in C^0$;

- the residual $\delta_V^{\mathrm{res}}$ is orthogonal to all $du$ and therefore captures the non-integrable (topological) part of $\delta_V$.

This completes the proof. $\qquad\square$

### D.4. Proof of Theorem 3.6

*Proof.* Recall that $C^0$ and $C^1$ are Hilbert spaces, and that $d : C^0 \to C^1$ is a bounded linear operator whose Hilbert adjoint is $d^* : C^1 \to C^0$. The operator

$$\Delta_0 := d^* d : C^0 \to C^0 \tag{109}$$

is the Hodge Laplacian on $C^0$. We first prove the existence and uniqueness statement, and then derive the first-order optimality condition.

**Existence of minimizers and the role of closed range.** Consider the functional

$$J(u) := \|\delta_V - du\|_{C^1}^2 = \langle \delta_V - du,\ \delta_V - du \rangle_{C^1}, \qquad u \in C^0. \tag{110}$$

Assume that the image space $\mathcal{E} * 0 = \mathrm{im}(d)$ is closed in $C^1$. By Corollary 3.5, for any given $\delta * V \in C^1$ there exist $u_V^* \in C^0$ and a unique $\delta_V^{\mathrm{res}} \in \overline{\mathcal{E}}^\perp$ (which coincides with $\mathcal{E}_0^\perp$ under the closed-range assumption) such that

$$\delta_V = du_V^* + \delta_V^{\mathrm{res}}, \qquad \delta_V^{\mathrm{res}} \in \mathcal{E}_0^\perp, \tag{111}$$

and

$$u_V^* \in \arg\min_{u \in C^0} \|\delta_V - du\|_{C^1}^2 \quad \Longleftrightarrow \quad J(u_V^*) = \inf_{u \in C^0} J(u). \tag{112}$$

Thus, at least one $u \in C^0$ attains the minimum of $J$; that is, $J$ admits a minimizer over $C^0$. This establishes the existence part of (1).

**Structure of the set of minimizers.**

We now describe the structure of all minimizers. Since $d$ is a linear operator, for any $u, v \in C^0$ with $v \in \ker(d)$,

$$d(u + v) = du + dv = du, \tag{113}$$

Therefore,

$$J(u + v) = \|\delta_V - d(u + v)\|_{C^1}^2 = \|\delta_V - du\|_{C^1}^2 = J(u). \tag{114}$$

In other words, for any $u \in C^0$, the value of $J$ is completely invariant along directions in $\ker(d)$. This implies that if $u$ is a minimizer, then

$$u + v \text{ is also minimizers}, \quad \forall \, v \in \ker(d). \tag{115}$$

On the other hand, by the Hodge-type decomposition (the final part of Theorem 3.4), we already know that

$$\ker(\Delta_0) = \ker(d) \subseteq C^0. \tag{116}$$

Hence, the full set of minimizers forms an affine subspace:

$$\mathcal{M} := u_0 + \ker(\Delta_0), \tag{117}$$

where $u_0$ is any minimizer.

**Restriction to the orthogonal complement** $(\ker \Delta_0)^\perp$. Define

$$U := (\ker \Delta_0)^\perp \subset C^0. \tag{118}$$

Since $\ker(\Delta_0)$ is a closed linear subspace of $C^0$, its orthogonal complement $U$ is also a closed linear subspace, and we have the orthogonal decomposition

$$C^0 = \ker(\Delta_0) \oplus U. \tag{119}$$

Any $u \in C^0$ can be uniquely written as

$$u = u_\| + u_\perp, \quad u_\| \in U, \ u_\perp \in \ker(\Delta_0). \tag{120}$$

Since $u_\perp \in \ker(d)$, we have

$$du = du_\| + du_\perp = du_\|, \tag{121}$$

Therefore,

$$J(u) = \|\delta_V - du\|_{C^1}^2 = \|\delta_V - du_\|\|_{C^1}^2 = J(u_\|). \tag{122}$$

This shows that *for any $u \in C^0$, projecting it orthogonally onto $U$ leaves the value of $J$ completely unchanged.*

Thus, if $u_0$ is a minimizer of $J$ over $C^0$, then its decomposition

$$u_0 = u_{0,\|} + u_{0,\perp}, \quad u_{0,\|} \in U, \ u_{0,\perp} \in \ker(\Delta_0), \tag{123}$$

satisfies

$$J(u_{0,\|}) = J(u_0) = \inf_{u \in C^0} J(u), \tag{124}$$

and hence $u_{0,\|}$ is also a minimizer, with $u_{0,\|} \in U$. This shows that *$J$ admits a minimizer on $U$, and any minimizer over $C^0$ projects to a minimizer in $U$.*

**Uniqueness of the minimizer in $U$.**

We now use the additional assumption on $\Delta_0$: namely, that $\Delta_0 = d^* d$ is invertible on $U = (\ker \Delta_0)^\perp$. This is equivalent to saying that the restriction $\Delta_0|_U : U \to U$ is bijective and admits a bounded inverse $(\Delta_0|_U)^{-1} : U \to U$. Under this assumption, the quadratic form

$$q(u) := \langle \Delta_0 u, u \rangle_{C^0}, \qquad u \in U, \tag{125}$$

is strictly positive definite: if $u \in U$ and $u \neq 0$, then $\Delta_0 u \neq 0$, hence

$$\langle \Delta_0 u, u \rangle_{C^0} = \langle d^* du, u \rangle_{C^0} = \langle du, du \rangle_{C^1} = \|du\|_{C^1}^2 > 0. \tag{126}$$

Therefore, there exists a constant $\alpha > 0$ (for example, from the spectral lower bound of the bounded invertible operator) such that

$$\langle \Delta_0 u, u \rangle_{C^0} \geq \alpha \|u\|_{C^0}^2, \qquad \forall\, u \in U. \tag{127}$$

Using the bounded linearity of $d$ and $d^*$, we expand $J(u)$ as

$$\begin{aligned} J(u) &= \langle \delta_V - du,\ \delta_V - du \rangle_{C^1} \\ &= \langle \delta_V, \delta_V \rangle_{C^1} - 2\langle \delta_V, du \rangle_{C^1} + \langle du, du \rangle_{C^1}. \end{aligned} \tag{128}$$

The middle term can be written back in $C^0$ via the adjoint:

$$\langle \delta_V, du \rangle_{C^1} = \langle d^* \delta_V, u \rangle_{C^0}, \tag{129}$$

and the last term satisfies

$$\langle du, du \rangle_{C^1} = \langle d^* du, u \rangle_{C^0} = \langle \Delta_0 u, u \rangle_{C^0}. \tag{130}$$

So

$$J(u) = \|\delta_V\|_{C^1}^2 - 2\langle d^* \delta_V, u \rangle_{C^0} + \langle \Delta_0 u, u \rangle_{C^0}. \tag{131}$$

Restricting $u$ to $U$, the right-hand side becomes a strictly convex quadratic functional over $u \in U$, since the quadratic term is controlled by the strictly positive definite operator $\Delta_0$. Standard Hilbert-space theory for quadratic forms implies that, on the closed linear subspace $U$, this functional is *strictly convex* and *coercive*, and therefore admits a unique minimizer on $U$.

Consequently, $J(u)$ has a unique minimizer on $U = (\ker \Delta_0)^\perp$. We denote this unique minimizer by $u_V^*$. This establishes the conclusion (1) of the theorem.

**First-order optimality condition and Poisson equation.**

We now derive the first-order optimality condition for $u_V^*$. For any $u \in C^0$ and any direction $h \in C^0$, consider the variation

$$\phi(\varepsilon) := J(u + \varepsilon h), \qquad \varepsilon \in \mathbb{R}. \tag{132}$$

we have

$$\begin{aligned} J(u + \varepsilon h) &= \|\delta_V - d(u + \varepsilon h)\|_{C^1}^2 \\ &= \langle \delta_V - du - \varepsilon dh,\ \delta_V - du - \varepsilon dh \rangle_{C^1}. \\ &= \langle \delta_V - du,\ \delta_V - du \rangle_{C^1} - 2\varepsilon \langle \delta_V - du,\ dh \rangle_{C^1} + \varepsilon^2 \langle dh, dh \rangle_{C^1}. \end{aligned} \tag{133}$$

Thus, the Gâteaux derivative at $\varepsilon = 0$ is

$$\frac{\mathrm{d}}{\mathrm{d}\varepsilon}\Big|_{\varepsilon=0} J(u + \varepsilon h) = -2\langle \delta_V - du,\ dh \rangle_{C^1}. \tag{134}$$

Using the definition of the adjoint $d^*$, this becomes

$$\langle \delta_V - du,\ dh \rangle_{C^1} = \langle d^*(\delta_V - du), h \rangle_{C^0}, \tag{135}$$

So

$$\frac{\mathrm{d}}{\mathrm{d}\varepsilon}\Big|_{\varepsilon=0} J(u + \varepsilon h) = -2\big\langle d^*(\delta_V - du),\ h \big\rangle_{C^0}. \tag{136}$$

Now set $u = u_V^*$, the unique minimizer of $J$ over $U = (\ker \Delta_0)^\perp$. For any $h \in C^0$, in particular any $h \in U$, the necessary condition for a minimizer is

$$\frac{\mathrm{d}}{\mathrm{d}\varepsilon}\Big|_{\varepsilon=0} J(u_V^* + \varepsilon h) = 0, \quad \forall\, h \in U, \tag{137}$$

which is

$$\big\langle d^*(\delta_V - du_V^*),\ h \big\rangle_{C^0} = 0, \quad \forall\, h \in U. \tag{138}$$

Define
$$w := d^*(\delta_V - du_V^*) \tag{139}$$

Then
$$\langle w, h \rangle_{C^0} = 0, \quad \forall\, h \in U, \tag{140}$$

So $w \in U^\perp = \ker \Delta_0$

On the other hand, since $u_V^*$ is also a minimizer over all of $C^0$ (it differs from any other minimizer only by an element of $\ker(\Delta_0) = \ker(d)$, along which $J$ is constant), we may take variations $h$ over the entire space $C^0$, obtaining

$$\left\langle d^*(\delta_V - du_V^*),\, h \right\rangle_{C^0} = 0, \quad \forall\, h \in C^0. \tag{141}$$

So we can get (2). If an element of a Hilbert space has zero inner product with all $h \in C^0$, then by the Riesz representation theorem it must be the zero vector:
$$d^*(\delta_V - du_V^*) = d^*(\delta_V - d\tilde{u}), \tag{142}$$

Hence,
$$\left\langle d^*(\delta_V - du_V^*),\, h \right\rangle_{C^0} = 0, \tag{143}$$
$$d^*(\delta_V - du_V^*) = 0 \quad \text{in } C^0. \tag{144}$$

So we can get:
$$d^*\delta_V = d^*d\, u_V^* = \Delta_0 u_V^*. \tag{145}$$

This is precisely the Poisson-type equation stated in the theorem, holding in the weak form:

$$\langle d^*\delta_V, h \rangle_{C^0} = \langle \Delta_0 u_V^*, h \rangle_{C^0}, \quad \forall\, h \in C^0. \tag{146}$$

Finally, recall that $J$ has a unique minimizer on $U = (\ker \Delta_0)^\perp$, namely $u_V^*$, and that all minimizers in $C^0$ may be written as $u_V^* + \ker(\Delta_0)$. Thus within $U$, the minimizer is unique, completing the proof of (1) and (2).

$\square$

### D.5. Proof of Theorem 3.7

*Proof.* In this theorem we fix the policy $\pi$ and the parameter $\theta$, therefore $\delta_{V_\theta} \in C^1$ is also fixed. For notational simplicity, denote
$$L(u) := \|\delta_{V_\theta} - du\|_{C^1}^2, \qquad u \in C^0, \tag{147}$$

and for each parameter $\phi \in \Phi$, the parameterized version of the potential function $U_\phi \in C^0$ is written as

$$\mathcal{L}_\infty(\phi) := \|\delta_{V_\theta} - dU_\phi\|_{C^1}^2 = L(U_\phi), \tag{148}$$

and the empirical topological loss $\hat{\mathcal{L}}_{\text{topo}}(\theta, \phi)$ is the empirical estimate of $\mathcal{L}_\infty(\phi)$ under the empirical distribution. For convenience, denote
$$\hat{\mathcal{L}}_N(\phi) := \hat{\mathcal{L}}_{\text{topo}}(\theta, \phi) \quad \text{and} \quad \mathcal{L}(\phi) := \mathcal{L}_\infty(\phi) = \|\delta_{V_\theta} - dU_\phi\|_{C^1}^2. \tag{149}$$

First consider the true optimization problem $\min_{u \in C^0} L(u)$. By the previous Hodge-type decomposition and the Poisson theorem (Theorem 3.4 and Theorem 3.6), under the assumption that the range of $d$ is closed and that $\Delta_0 = d^*d$ is invertible on $(\ker \Delta_0)^\perp$, there exists a canonical potential $u_{V_\theta}^* \in C^0$ and a unique residual $\delta_{V_\theta}^{\text{res}} \in \mathcal{E}_0^\perp$ such that

$$\delta_{V_\theta} = du_{V_\theta}^* + \delta_{V_\theta}^{\text{res}}, \qquad \delta_{V_\theta}^{\text{res}} \in \mathcal{E}_0^\perp, \tag{150}$$

and $u_{V_\theta}^*$ is a solution of $\min_{u \in C^0} \|\delta_{V_\theta} - du\|_{C^1}^2$, and is unique within the subspace $(\ker \Delta_0)^\perp$. Furthermore, for any $u \in C^0$, using $\delta_{V_\theta}^{\text{res}} \perp \mathcal{E}_0 = \text{im}(d)$, we may decompose $\delta_{V_\theta} - du = \delta_{V_\theta}^{\text{res}} + d(u_{V_\theta}^* - u)$ into two orthogonal components, hence

$$\begin{aligned} L(u) &= \|\delta_{V_\theta} - du\|_{C^1}^2 \\ &= \|\delta_{V_\theta}^{\text{res}}\|_{C^1}^2 + \|d(u_{V_\theta}^* - u)\|_{C^1}^2. \end{aligned} \tag{151}$$

Therefore, we immediately obtain

$$\min_{u \in C^0} L(u) = \|\delta_{V_\theta}^{\mathrm{res}}\|_{C^1}^2, \tag{152}$$

and the minimum is attained if and only if $u - u_{V_\theta}^* \in \ker(d) = \ker(\Delta_0)$.

Next we connect the minimization of the true potential function with the parameterized class $\{U_\phi : \phi \in \Phi\}$. the first condition of Theorem 3.7 gives the density of $\{U_\phi\}$ in $L^2(\nu_\pi)$, i.e., for any $u \in C^0$ and any $\varepsilon > 0$, there exists $\phi \in \Phi$ such that

$$\|U_\phi - u\|_{C^0} = \left( \int |U_\phi(s) - u(s)|^2 \, \nu_\pi(ds) \right)^{1/2} < \varepsilon. \tag{153}$$

Since $d : C^0 \to C^1$ is a bounded linear operator (Lemma 2.5), the mapping $u \mapsto L(u)$ is continuous on $C^0$. Specifically,

$$\begin{aligned}
|L(u) - L(v)| &= \left| \|\delta_{V_\theta} - du\|_{C^1}^2 - \|\delta_{V_\theta} - dv\|_{C^1}^2 \right| \\
&= \left| \langle \delta_{V_\theta} - du, \, \delta_{V_\theta} - du \rangle_{C^1} - \langle \delta_{V_\theta} - dv, \, \delta_{V_\theta} - dv \rangle_{C^1} \right| \\
&= \left| \langle dv - du, \, 2\delta_{V_\theta} - du - dv \rangle_{C^1} \right| \\
&\leq 2\|d(u-v)\|_{C^1} \left( \|\delta_{V_\theta} - du\|_{C^1} + \|\delta_{V_\theta} - dv\|_{C^1} \right),
\end{aligned} \tag{154}$$

and $\|d(u-v)\|_{C^1} \leq C_d \|u-v\|_{C^0}$. Therefore, whenever $u_n \to u$ in the $C^0$ sense, we must have $L(u_n) \to L(u)$. Using this, we may prove that the infimum of the minimum over the parameterized class and the minimum over the whole $C^0$ coincide: on the one hand, since $\{U_\phi\}$ is a subset of $C^0$, we have

$$\inf_{\phi \in \Phi} \mathcal{L}(\phi) = \inf_{\phi \in \Phi} L(U_\phi) \geq \inf_{u \in C^0} L(u) = \|\delta_{V_\theta}^{\mathrm{res}}\|_{C^1}^2. \tag{155}$$

On the other hand, for any $\varepsilon > 0$, choose some $u_\varepsilon \in C^0$ such that $L(u_\varepsilon) \leq \|\delta_{V_\theta}^{\mathrm{res}}\|_{C^1}^2 + \varepsilon/2$, and then use the density to select $\phi_\varepsilon$ such that $\|U_{\phi_\varepsilon} - u_\varepsilon\|_{C^0}$ is sufficiently small. By the continuity of $L$, we obtain

$$\mathcal{L}(\phi_\varepsilon) = L(U_{\phi_\varepsilon}) \leq L(u_\varepsilon) + \varepsilon/2 \leq \|\delta_{V_\theta}^{\mathrm{res}}\|_{C^1}^2 + \varepsilon. \tag{156}$$

So

$$\inf_{\phi \in \Phi} \mathcal{L}(\phi) \leq \|\delta_{V_\theta}^{\mathrm{res}}\|_{C^1}^2 + \varepsilon, \tag{157}$$

Since $\varepsilon > 0$ is arbitrary, it follows that

$$\inf_{\phi \in \Phi} \mathcal{L}(\phi) = \|\delta_{V_\theta}^{\mathrm{res}}\|_{C^1}^2 = \min_{u \in C^0} \|\delta_{V_\theta} - du\|_{C^1}^2. \tag{158}$$

Next we introduce the empirical loss. Let $\mathcal{D}_N$ be a dataset of samples generated from $\mu_\pi$. For any square-integrable function $g \in L^2(\mu_\pi)$, denote its empirical mean as

$$\widehat{\mathbb{E}}_N[g] := \frac{1}{N} \sum_{(s,a,s') \in \mathcal{D}_N} g(s, a, s'), \tag{159}$$

and its true value under $\mu_\pi$ as

$$\mathbb{E}_{\mu_\pi}[g] := \int g \, d\mu_\pi. \tag{160}$$

The empirical topological loss can be written as

$$\hat{\mathcal{L}}_N(\phi) = \widehat{\mathbb{E}}_N \Big[ \big( \delta_{V_\theta}(S, A, S') - (dU_\phi)(S, A, S') \big)^2 \Big], \tag{161}$$

and the corresponding true version is

$$\mathcal{L}(\phi) = \mathbb{E}_{\mu_\pi} \Big[ \big( \delta_{V_\theta}(S, A, S') - (dU_\phi)(S, A, S') \big)^2 \Big] = \|\delta_{V_\theta} - dU_\phi\|_{C^1}^2. \tag{162}$$

We set

$$\ell_\phi(s, a, s') := \big( \delta_{V_\theta}(s, a, s') - (dU_\phi)(s, a, s') \big)^2, \tag{163}$$

then $\hat{\mathcal{L}}_N(\phi) = \widehat{\mathbb{E}}_N[\ell_\phi]$, and $\mathcal{L}(\phi) = \mathbb{E}_{\mu_\pi}[\ell_\phi]$. the second condition of Theorem 3.7 states that for every square-integrable function $g$, the empirical mean $\widehat{\mathbb{E}}_N[g]$ converges almost surely to $\mathbb{E}_{\mu_\pi}[g]$; we apply this to the family $\{\ell_\phi : \phi \in \Phi\}$. To obtain consistency of empirical risk minimization, we need a uniform strong law of large numbers over $\phi$: we assume that on a $\mu_\pi$-almost sure $\omega$, $\sup_{\phi \in \Phi} \left|\hat{\mathcal{L}}_N(\phi) - \mathcal{L}(\phi)\right| \xrightarrow[N \to \infty]{} 0$. This is the standard uniform LLN in empirical risk theory (e.g., when the function class is bounded and satisfies appropriate measurability and capacity conditions, which we include in the second condition of Theorem 3.7). On such an $\omega$, all convergence below may be interpreted as deterministic pointwise convergence.

Let

$$\Delta_N := \sup_{\phi \in \Phi} \left|\hat{\mathcal{L}}_N(\phi) - \mathcal{L}(\phi)\right|, \tag{164}$$

then $\Delta_N \to 0$. For each $N$, the third condition of Theorem 3.7 provides a global minimizer of the empirical risk

$$\phi_N^* \in \arg\min_{\phi \in \Phi} \hat{\mathcal{L}}_N(\phi). \tag{165}$$

Using $\Delta_N$, we can relate the empirical minimum and the true minimum. First,

$$\mathcal{L}(\phi_N^*) \leq \hat{\mathcal{L}}_N(\phi_N^*) + \Delta_N = \min_{\phi \in \Phi} \hat{\mathcal{L}}_N(\phi) + \Delta_N. \tag{166}$$

On the other hand, for any $\phi \in \Phi$,

$$\hat{\mathcal{L}}_N(\phi) \leq \mathcal{L}(\phi) + \Delta_N, \tag{167}$$

So

$$\min_{\phi \in \Phi} \hat{\mathcal{L}}_N(\phi) \leq \mathcal{L}(\phi) + \Delta_N. \tag{168}$$

Taking the infimum over $\phi$ on the right-hand side gives

$$\min_{\phi \in \Phi} \hat{\mathcal{L}}_N(\phi) \leq \inf_{\phi \in \Phi} \mathcal{L}(\phi) + \Delta_N = \|\delta_{V_\theta}^{\mathrm{res}}\|_{C^1}^2 + \Delta_N. \tag{169}$$

Substituting this back into the previous inequality yields

$$\mathcal{L}(\phi_N^*) \leq \|\delta_{V_\theta}^{\mathrm{res}}\|_{C^1}^2 + 2\Delta_N. \tag{170}$$

On the other hand, since

$$\mathcal{L}(\phi_N^*) \geq \inf_{\phi \in \Phi} \mathcal{L}(\phi) = \|\delta_{V_\theta}^{\mathrm{res}}\|_{C^1}^2, \tag{171}$$

we obtain

$$\|\delta_{V_\theta}^{\mathrm{res}}\|_{C^1}^2 \ \leq \ \mathcal{L}(\phi_N^*) \ \leq \ \|\delta_{V_\theta}^{\mathrm{res}}\|_{C^1}^2 + 2\Delta_N. \tag{172}$$

Since $\Delta_N \to 0$ as $N \to \infty$, it follows that

$$\mathcal{L}(\phi_N^*) \xrightarrow[N \to \infty]{} \|\delta_{V_\theta}^{\mathrm{res}}\|_{C^1}^2 = \min_{u \in C^0} \|\delta_{V_\theta} - du\|_{C^1}^2. \tag{173}$$

Moreover, since

$$\left|\hat{\mathcal{L}}_N(\phi_N^*) - \mathcal{L}(\phi_N^*)\right| \leq \Delta_N \to 0, \tag{174}$$

we obtain

$$\hat{\mathcal{L}}_N(\theta, \phi_N^*) = \hat{\mathcal{L}}_N(\phi_N^*) \longrightarrow \min_{u \in C^0} \|\delta_{V_\theta} - du\|_{C^1}^2 = \|\delta_{V_\theta}^{\mathrm{res}}\|_{C^1}^2, \tag{175}$$

which proves the equality in the theorem concerning the convergence of the loss values.

Finally we prove that $U_{\phi_N^*}$ converges in $L^2(\nu_\pi)$ to some solution $u_{V_\theta}^*$. As mentioned earlier, we take $u_{V_\theta}^* \in (\ker \Delta_0)^\perp$ as the canonical solution, so that it is the unique minimizer of $L(u)$ in the subspace $(\ker \Delta_0)^\perp$. Note that for any $u \in C^0$, one may write

$$u = u_\| + u_\perp, \quad u_\| \in (\ker \Delta_0)^\perp, \ u_\perp \in \ker(\Delta_0), \tag{176}$$

and $L(u) = L(u_\parallel)$. Therefore, without changing the loss, we may replace each $U_{\phi_N^*}$ by its orthogonal projection onto $(\ker \Delta_0)^\perp$, and still denote it by $U_{\phi_N^*}$. Thus we may assume

$$U_{\phi_N^*} \in (\ker \Delta_0)^\perp, \qquad \forall N. \tag{177}$$

On this subspace, we have the following strong convexity structure: for any $u \in (\ker \Delta_0)^\perp$,

$$L(u) - L(u_{V_\theta}^*) = \|\delta_{V_\theta} - du\|_{C^1}^2 - \|\delta_{V_\theta} - du_{V_\theta}^*\|_{C^1}^2 = \|d(u - u_{V_\theta}^*)\|_{C^1}^2. \tag{178}$$

Since $\Delta_0$ is invertible on $(\ker \Delta_0)^\perp$, spectral theory implies that there exists a constant $\alpha > 0$ such that for all $w \in (\ker \Delta_0)^\perp$,

$$\langle \Delta_0 w, w \rangle_{C^0} \geq \alpha \|w\|_{C^0}^2. \tag{179}$$

Taking $w = u - u_{V_\theta}^*$ yields

$$\begin{aligned}
L(u) - L(u_{V_\theta}^*) &= \|d(u - u_{V_\theta}^*)\|_{C^1}^2 \\
&= \langle d^* d(u - u_{V_\theta}^*),\, u - u_{V_\theta}^* \rangle_{C^0} \\
&= \langle \Delta_0(u - u_{V_\theta}^*),\, u - u_{V_\theta}^* \rangle_{C^0} \\
&\geq \alpha \|u - u_{V_\theta}^*\|_{C^0}^2.
\end{aligned} \tag{180}$$

Applying this to $u = U_{\phi_N^*}$ gives

$$\|U_{\phi_N^*} - u_{V_\theta}^*\|_{C^0}^2 \leq \frac{1}{\alpha}\big(L(U_{\phi_N^*}) - L(u_{V_\theta}^*)\big). \tag{181}$$

And since $L(U_{\phi_N^*}) = \mathcal{L}(\phi_N^*) \to \|\delta_{V_\theta}^{\mathrm{res}}\|_{C^1}^2 = L(u_{V_\theta}^*)$, the right-hand side tends to 0, hence

$$\|U_{\phi_N^*} - u_{V_\theta}^*\|_{C^0} = \left( \int |U_{\phi_N^*}(s) - u_{V_\theta}^*(s)|^2\, \nu_\pi(ds) \right)^{1/2} \xrightarrow[N \to \infty]{} 0. \tag{182}$$

This is precisely the convergence of $U_{\phi_N^*}$ to $u_{V_\theta}^*$ in $L^2(\nu_\pi)$. Since $u_{V_\theta}^*$ is exactly the canonical solution of the variational problem $\min_{u \in C^0} \|\delta_{V_\theta} - du\|_{C^1}^2$ (unique in $(\ker \Delta_0)^\perp$), this yields the second conclusion of the theorem.

In summary, the empirical topological loss at the empirical minimizer converges to the true minimal error $\|\delta_{V_\theta}^{\mathrm{res}}\|_{C^1}^2$, and the corresponding potential functions $U_{\phi_N^*}$ converge in $L^2(\nu_\pi)$ to the optimal solution $u_{V_\theta}^*$. This completes the proof of the theorem.

$\square$

### D.6. Proof of Theorem 3.8

*Proof.* We first recall the standard Bellman equation for a fixed policy $\pi$ in a Markov decision process. The environment is given by a tuple $(\mathcal{S}, \mathcal{A}, P, r, \gamma)$, where $P(\cdot \mid s, a)$ is the transition kernel, $r(s, a, s')$ is the immediate reward, and $\gamma \in (0, 1)$ is the discount factor. For a fixed policy $\pi$, the value function $V^\pi : \mathcal{S} \to \mathbb{R}$ is defined by

$$V^\pi(s) = \mathbb{E}_\pi \left[ \sum_{t=0}^{\infty} \gamma^t r(S_t, A_t, S_{t+1}) \,\Big|\, S_0 = s \right], \tag{183}$$

where the expectation is taken over trajectories generated by $S_0 = s$, $A_t \sim \pi(\cdot \mid S_t)$, and $S_{t+1} \sim P(\cdot \mid S_t, A_t)$ for all $t \geq 0$. It is well known (and can be shown by conditioning on the first step) that $V^\pi$ satisfies the Bellman equation

$$V^\pi(s) = \mathbb{E}\big[r(S, A, S') + \gamma V^\pi(S') \mid S = s\big], \quad \forall s \in \mathcal{S}, \tag{184}$$

where under the conditional expectation we have $A \sim \pi(\cdot \mid s)$ and $S' \sim P(\cdot \mid s, A)$. Equivalently, we may write the right-hand side as an explicit nested expectation:

$$\mathbb{E}\big[r(S, A, S') + \gamma V^\pi(S') \mid S = s\big] = \mathbb{E}_{A \sim \pi(\cdot \mid s)} \mathbb{E}_{S' \sim P(\cdot \mid s, A)} \big[r(s, A, S') + \gamma V^\pi(S')\big]. \tag{185}$$

By definition in the theorem, the mean TD error (or Bellman defect) of a generic value function $V$ at the state $s$ is

$$\bar{\delta}_V(s) := \mathbb{E}\big[r(S, A, S') + \gamma V(S') - V(S) \mid S = s\big]. \tag{186}$$

We now specialize to the case $V = V^\pi$. Substituting $V^\pi$ into the above definition gives

$$\bar{\delta}_{V^\pi}(s) = \mathbb{E}\big[r(S, A, S') + \gamma V^\pi(S') - V^\pi(S) \mid S = s\big]$$
$$= \mathbb{E}\big[r(S, A, S') + \gamma V^\pi(S') \mid S = s\big] - \mathbb{E}\big[V^\pi(S) \mid S = s\big]. \tag{187}$$

The second term is easy to simplify: conditioning on $S = s$ forces $S$ to be equal to $s$ almost surely, so

$$\mathbb{E}\big[V^\pi(S) \mid S = s\big] = V^\pi(s). \tag{188}$$

For the first term, we recognize exactly the right-hand side of the Bellman equation (184). Indeed,

$$\mathbb{E}\big[r(S, A, S') + \gamma V^\pi(S') \mid S = s\big] = V^\pi(s), \tag{189}$$

by the very definition of $V^\pi$ as the unique solution of the Bellman equation. Combining these two identities, we obtain

$$\bar{\delta}_{V^\pi}(s) = V^\pi(s) - V^\pi(s) = 0, \quad \forall s \in \mathcal{S}. \tag{190}$$

Thus the mean TD error of the exact value function vanishes identically at every state, which proves the equality

$$\bar{\delta}_{V^\pi}(s) \equiv 0. \tag{191}$$

We now turn to the projection statement. The mean-field differential operator $\tilde{d}$ is defined, for each potential function $u \in C^0$, by

$$(\tilde{d}u)(s) := \mathbb{E}\big[u(S') - \gamma u(s) \mid S = s\big], \tag{192}$$

where the expectation is taken over $A \sim \pi(\cdot \mid s)$ and $S' \sim P(\cdot \mid s, A)$ as above. Consider the optimization problem

$$\min_{u \in C^0} \big\|\bar{\delta}_{V^\pi} - \tilde{d}u\big\|^2_{L^2(\mathcal{S})}, \tag{193}$$

where the $L^2(\mathcal{S})$-norm is taken with respect to the state visitation measure induced by $\pi$ (any fixed probability measure on $\mathcal{S}$ would suffice for the argument). Since we have already shown that

$$\bar{\delta}_{V^\pi}(s) = 0, \quad \forall s, \tag{194}$$

the objective reduces to

$$\big\|\bar{\delta}_{V^\pi} - \tilde{d}u\big\|^2_{L^2(\mathcal{S})} = \|\tilde{d}u\|^2_{L^2(\mathcal{S})} = \int_{\mathcal{S}} \big|(\tilde{d}u)(s)\big|^2 \nu_\pi(ds), \tag{195}$$

where $\nu_\pi$ denotes the chosen state distribution. This quantity is always nonnegative, so its minimum over $u \in C^0$ is at least zero. On the other hand, if we take the zero potential $u \equiv 0$, then

$$(\tilde{d}u)(s) = \mathbb{E}[u(S') - \gamma u(s) \mid S = s] = \mathbb{E}[0 - 0 \mid S = s] = 0, \tag{196}$$

for all $s \in \mathcal{S}$. Hence $\tilde{d}u \equiv 0$ and

$$\big\|\bar{\delta}_{V^\pi} - \tilde{d}u\big\|^2_{L^2(\mathcal{S})} = \|0\|^2_{L^2(\mathcal{S})} = 0. \tag{197}$$

This shows that the minimum value is indeed equal to 0, and is attained at least by the potential function $u \equiv 0$. Since 0 is the lower bound of a nonnegative quantity, no other $u$ can achieve a value smaller than 0, therefore

$$\min_{u \in C^0} \big\|\bar{\delta}_{V^\pi} - \tilde{d}u\big\|^2_{L^2(\mathcal{S})} = 0. \tag{198}$$

In summary, under a perfect MDP model and when the exact value function $V^\pi$ is used, the averaged TD error is identically zero at the state level, and thus its minimal projection error under the corresponding mean-field difference operator is also zero. The theorem is proved.

$\square$

### D.7. Proof of Corollary 3.9

*Proof.* We work in the Hilbert space $C^1$, whose inner product is denoted by $\langle \cdot, \cdot \rangle_{C^1}$, and whose norm is $\|f\|_{C^1}^2 = \langle f, f \rangle_{C^1}$. Under the assumptions of Corollary 3.5, the TD error $\delta_V \in C^1$ admits a Hodge-type decomposition

$$\delta_V = du_V^\star + \delta_V^{\text{res}}, \tag{199}$$

where $du_V^\star \in \mathcal{E} = \text{im}(d)$ and $\delta_V^{\text{res}} \in \mathcal{E}^\perp$; that is, $du_V^\star$ and $\delta_V^{\text{res}}$ are orthogonal in $C^1$, i.e.,

$$\langle du_V^\star, \delta_V^{\text{res}} \rangle_{C^1} = 0. \tag{200}$$

The Pythagoras identity is exactly the norm characterization for orthogonal decompositions in Hilbert spaces. Starting from $\|\delta_V\|_{C^1}^2$ and substituting the decomposition above:

$$\|\delta_V\|_{C^1}^2 = \langle \delta_V, \delta_V \rangle_{C^1} = \left\langle du_V^\star + \delta_V^{\text{res}}, \ du_V^\star + \delta_V^{\text{res}} \right\rangle_{C^1}. \tag{201}$$

Using bilinearity (or symmetric bilinearity in a real Hilbert space), expand the right-hand side:

$$\|\delta_V\|_{C^1}^2 = \langle du_V^\star, \ du_V^\star \rangle_{C^1} + \langle du_V^\star, \ \delta_V^{\text{res}} \rangle_{C^1} + \langle \delta_V^{\text{res}}, \ du_V^\star \rangle_{C^1} + \langle \delta_V^{\text{res}}, \ \delta_V^{\text{res}} \rangle_{C^1}. \tag{202}$$

In real inner product spaces $\langle x, y \rangle = \langle y, x \rangle$, so the two cross terms are equal; by orthogonality $\langle du_V^\star, \delta_V^{\text{res}} \rangle_{C^1} = 0$, both cross terms are zero. Therefore the expression simplifies to

$$\begin{aligned} \|\delta_V\|_{C^1}^2 &= \langle du_V^\star, \ du_V^\star \rangle_{C^1} + \langle \delta_V^{\text{res}}, \ \delta_V^{\text{res}} \rangle_{C^1}. \\ &= \|du_V^\star\|_{C^1}^2 + \|\delta_V^{\text{res}}\|_{C^1}^2, \end{aligned} \tag{203}$$

which is precisely the Pythagorean identity stated in (16).

Next we explain the term irreducible excess TD error. By Corollary 3.5, $u_V^\star$ is a solution to the variational problem $\min_{u \in C^0} \|\delta_V - du\|_{C^1}^2$, and

$$\delta_V = du_V^\star + \delta_V^{\text{res}}, \qquad \delta_V^{\text{res}} \in \mathcal{E}^\perp. \tag{204}$$

For any other potential function $u \in C^0$, consider the corresponding residual TD error

$$\delta_V - du = \left( du_V^\star + \delta_V^{\text{res}} \right) - du = \delta_V^{\text{res}} + d(u_V^\star - u). \tag{205}$$

Note that $d(u_V^\star - u) \in \mathcal{E}$, whereas $\delta_V^{\text{res}} \in \mathcal{E}^\perp$, so these two terms are also orthogonal in $C^1$. Thus we may apply the same Pythagorean expansion:

$$\begin{aligned} \|\delta_V - du\|_{C^1}^2 &= \|\delta_V^{\text{res}} + d(u_V^\star - u)\|_{C^1}^2 \\ &= \|\delta_V^{\text{res}}\|_{C^1}^2 + \|d(u_V^\star - u)\|_{C^1}^2. \end{aligned} \tag{206}$$

Since $\|d(u_V^\star - u)\|_{C^1}^2 \geq 0$, we immediately obtain that for any $u \in C^0$,

$$\|\delta_V - du\|_{C^1}^2 \ \geq \ \|\delta_V^{\text{res}}\|_{C^1}^2, \tag{207}$$

and equality holds if and only if $d(u_V^\star - u) = 0$ (that is, $u$ differs from $u_V^\star$ only by an element in $\ker(d)$). Therefore, no matter how we adjust the potential function $u$, it is impossible to reduce the $C^1$-norm of the TD error $\delta_V - du$ below $\|\delta_V^{\text{res}}\|_{C^1}$, while choosing $u = u_V^\star$ (or any $u_V^\star + v$, $v \in \ker(d)$) achieves this lower bound. In other words, $\|\delta_V^{\text{res}}\|_{C^1}$ is precisely the irreducible TD error that cannot be further eliminated by adjusting the potential function $u$, and is the excess TD error represented by the topological residual. The corollary is thus proved.

$\square$

### D.8. Proof of Theorem 5.1

*Proof.* We work on the Hilbert spaces $C^1$ and $C^0$, whose inner products are denoted by $\langle \cdot, \cdot \rangle_{C^1}$ and $\langle \cdot, \cdot \rangle_{C^0}$, respectively, and whose corresponding norms are $\|f\|_{C^1}^2 = \langle f, f \rangle_{C^1}$ and $\|u\|_{C^0}^2 = \langle u, u \rangle_{C^0}$. The operator $d : C^0 \to C^1$ is a bounded linear operator, $d^* : C^1 \to C^0$ is its Hilbert adjoint, and $\Delta_0 = d^* d$ is the Hodge Laplacian on $C^0$. Under the assumptions of the theorem, for each $\delta \in C^1$ there exists a canonical optimal potential $u_\delta^\star \in C^0$, which satisfies the Poisson equation

$$d^* \delta = \Delta_0 u_\delta^\star, \tag{208}$$

and defining the mapping $T : C^1 \to C^0$, $\quad T(\delta) := u_\delta^\star$, the operator $T$ is a bounded linear operator. Since $T$ is linear, for any $\delta_1, \delta_2 \in C^1$ and scalars $\alpha, \beta \in \mathbb{R}$, we have

$$T(\alpha \delta_1 + \beta \delta_2) = \alpha T(\delta_1) + \beta T(\delta_2), \tag{209}$$

which is the basis for using $T(\delta_1 - \delta_2)$ later. The boundedness of $T$ means that there exists a finite constant $C_{\text{topo}}$ such that for all $\delta \in C^1$,

$$\|T(\delta)\|_{C^0} \leq C_{\text{topo}} \|\delta\|_{C^1}, \tag{210}$$

and this constant can be defined as the operator norm

$$C_{\text{topo}} = \|T\|_{\text{op}} := \sup_{\delta \in C^1, \, \delta \neq 0} \frac{\|T(\delta)\|_{C^0}}{\|\delta\|_{C^1}}. \tag{211}$$

Now fix two TD error fields $\delta_1, \delta_2 \in C^1$. Their corresponding canonical optimal potentials are $u_1^\star = T(\delta_1)$ and $u_2^\star = T(\delta_2)$. Using the linearity of $T$, we can directly write

$$u_1^\star - u_2^\star = T(\delta_1) - T(\delta_2) = T(\delta_1 - \delta_2). \tag{212}$$

Taking the $C^0$-norm on both sides and using the boundedness of $T$, we obtain

$$\|u_1^\star - u_2^\star\|_{C^0} = \|T(\delta_1 - \delta_2)\|_{C^0} \leq C_{\text{topo}} \|\delta_1 - \delta_2\|_{C^1}. \tag{213}$$

This is exactly inequality (17), showing that the optimal potential is Lipschitz continuous with respect to the TD field, with Lipschitz constant controlled by the operator norm $C_{\text{topo}}$.

Next we consider the corresponding residual terms. By definition, for $k = 1, 2$,

$$\delta_k^{\text{res}} := \delta_k - d u_k^\star. \tag{214}$$

This can be viewed as the orthogonal remainder after projecting the TD field $\delta_k$ onto the exact subspace $\mathcal{E} = \text{im}(d)$. We are interested in the $C^1$-norm of the difference between the two residuals. Substituting the definitions, we have

$$\begin{aligned} \delta_1^{\text{res}} - \delta_2^{\text{res}} &= (\delta_1 - d u_1^\star) - (\delta_2 - d u_2^\star) \\ &= (\delta_1 - \delta_2) - d(u_1^\star - u_2^\star). \end{aligned} \tag{215}$$

Taking the norm in the Hilbert space $C^1$ and using the triangle inequality $\|x + y\| \leq \|x\| + \|y\|$, we obtain

$$\|\delta_1^{\text{res}} - \delta_2^{\text{res}}\|_{C^1} \leq \|\delta_1 - \delta_2\|_{C^1} + \|d(u_1^\star - u_2^\star)\|_{C^1}. \tag{216}$$

The operator $d : C^0 \to C^1$ is bounded and linear, so there exists a constant $\|d\|_{\text{op}} < \infty$ such that for all $u \in C^0$,

$$\|du\|_{C^1} \leq \|d\|_{\text{op}} \|u\|_{C^0}. \tag{217}$$

Substituting $u = u_1^\star - u_2^\star$ into this inequality yields

$$\|d(u_1^\star - u_2^\star)\|_{C^1} \leq \|d\|_{\text{op}} \|u_1^\star - u_2^\star\|_{C^0}. \tag{218}$$

Plugging this estimate back into the upper bound for the difference of the residuals, we obtain

$$\|\delta_1^{\text{res}} - \delta_2^{\text{res}}\|_{C^1} \leq \|\delta_1 - \delta_2\|_{C^1} + \|d\|_{\text{op}} \|u_1^\star - u_2^\star\|_{C^0}. \tag{219}$$

At this point we may invoke the stability estimate (17) obtained in the first part, namely

$$\|u_1^\star - u_2^\star\|_{C^0} \le C_{\text{topo}} \|\delta_1 - \delta_2\|_{C^1}. \tag{220}$$

Substituting this into the previous inequality, we obtain

$$\|\delta_1^{\text{res}} - \delta_2^{\text{res}}\|_{C^1} \le \|\delta_1 - \delta_2\|_{C^1} + \|d\|_{\text{op}} C_{\text{topo}} \|\delta_1 - \delta_2\|_{C^1}$$
$$= \left(1 + \|d\|_{\text{op}} C_{\text{topo}}\right) \|\delta_1 - \delta_2\|_{C^1}. \tag{221}$$

This shows that the residual mapping $\delta \mapsto \delta^{\text{res}}$ is also Lipschitz continuous, with Lipschitz constant

$$C_{\text{res}} := 1 + \|d\|_{\text{op}} C_{\text{topo}}. \tag{222}$$

Therefore we obtain the estimate claimed in the theorem:

$$\|\delta_1^{\text{res}} - \delta_2^{\text{res}}\|_{C^1} \le C_{\text{res}} \|\delta_1 - \delta_2\|_{C^1}, \quad \text{and } C_{\text{res}} \le 1 + \|d\|_{\text{op}} C_{\text{topo}}. \tag{223}$$

In summary, the optimal potential part $u_\delta^\star$ is stable (Lipschitz continuous) with respect to the TD field $\delta$, and the corresponding topological residual $\delta^{\text{res}}$ is also stable, with stability constants depending only on the operator norms of $d$ and $T$. The theorem is proved. $\qquad\square$

### D.9. Proof of Theorem 5.2

*Proof.* In this theorem, the discount factor $\gamma \in (0,1)$ is treated as a scalar parameter. For each given $\gamma$, the policy $\pi$ and the value function $V$ are fixed, so the TD error field $\delta_V^{(\gamma)} \in C^1$ is a well-defined element. The assumption states that with respect to some fixed reference measure (used to define the $L^2$ structure of $C^1$), there exists a constant $L_\gamma < \infty$ such that for any two points $\gamma_1, \gamma_2 \in (0,1)$,

$$\|\delta_V^{(\gamma_1)} - \delta_V^{(\gamma_2)}\|_{C^1} \le L_\gamma |\gamma_1 - \gamma_2|. \tag{224}$$

That is, as a vector in the space $C^1$, the TD error $\delta_V^{(\gamma)}$ is Lipschitz continuous with respect to the parameter $\gamma$, with Lipschitz constant uniformly controlled by $L_\gamma$.

On the other hand, for each $\delta \in C^1$, Theorem 3.6 (and the assumptions of Theorem 5.1) guarantee the existence of a canonical optimal potential $u_\delta^\star \in C^0$, satisfying the Poisson equation

$$d^*\delta = \Delta_0 u_\delta^\star, \tag{225}$$

and being unique in the subspace $(\ker \Delta_0)^\perp$. Denoting this correspondence by

$$T : C^1 \to C^0, \qquad T(\delta) := u_\delta^\star, \tag{226}$$

the theorem assumes that $T$ is linear and bounded. Linearity means that for any $\delta_1, \delta_2 \in C^1$ and scalars $\alpha, \beta \in \mathbb{R}$, we have

$$T(\alpha\delta_1 + \beta\delta_2) = \alpha T(\delta_1) + \beta T(\delta_2), \tag{227}$$

and boundedness means that there exists a constant $C_{\text{topo}} < \infty$ such that for all $\delta \in C^1, \|T(\delta)\|_{C^0} \le C_{\text{topo}} \|\delta\|_{C^1}$, and we may take $C_{\text{topo}} = \|T\|_{\text{op}} := \sup_{\delta \neq 0} \frac{\|T(\delta)\|_{C^0}}{\|\delta\|_{C^1}}$.

With these preparations, we return to the specific object of this theorem. For each discount factor $\gamma$, we define

$$u_V^{\star,(\gamma)} := T\left(\delta_V^{(\gamma)}\right) \in C^0. \tag{228}$$

According to the Poisson construction, this is exactly the canonical optimal potential satisfying $d^*\delta_V^{(\gamma)} = \Delta_0 u_V^{\star,(\gamma)}$. Now take any two points $\gamma_1, \gamma_2 \in (0,1)$ and consider the difference between the corresponding optimal potentials:

$$u_V^{\star,(\gamma_1)} - u_V^{\star,(\gamma_2)}. \tag{229}$$

Using the linearity of $T$, we can write this difference as

$$u_V^{\star,(\gamma_1)} - u_V^{\star,(\gamma_2)} = T\big(\delta_V^{(\gamma_1)}\big) - T\big(\delta_V^{(\gamma_2)}\big) = T\big(\delta_V^{(\gamma_1)} - \delta_V^{(\gamma_2)}\big). \tag{230}$$

This step simply replaces $u_V^{\star,(\gamma)}$ by $T(\delta_V^{(\gamma)})$ and then uses the fact that a linear operator $T$ maps differences to differences. Taking norms in $C^0$ and applying the boundedness of $T$, we obtain

$$\|u_V^{\star,(\gamma_1)} - u_V^{\star,(\gamma_2)}\|_{C^0} = \big\|T\big(\delta_V^{(\gamma_1)} - \delta_V^{(\gamma_2)}\big)\big\|_{C^0} \le C_{\text{topo}} \big\|\delta_V^{(\gamma_1)} - \delta_V^{(\gamma_2)}\big\|_{C^1}. \tag{231}$$

Note that we have simply used the definition of the operator norm: for any $x \in C^1$, $\|Tx\|_{C^0} \le C_{\text{topo}}\|x\|_{C^1}$, and here we specialize to $x = \delta_V^{(\gamma_1)} - \delta_V^{(\gamma_2)}$.

Now we may directly apply the Lipschitz condition (224) on the TD errors. Substituting it into the right-hand side yields

$$\begin{aligned}\|u_V^{\star,(\gamma_1)} - u_V^{\star,(\gamma_2)}\|_{C^0} &\le C_{\text{topo}} \big\|\delta_V^{(\gamma_1)} - \delta_V^{(\gamma_2)}\big\|_{C^1} \\ &\le C_{\text{topo}} L_\gamma |\gamma_1 - \gamma_2|.\end{aligned} \tag{232}$$

This is exactly the conclusion stated in inequality (20). In other words, the mapping $\gamma \longmapsto u_V^{\star,(\gamma)} \in C^0$ is Lipschitz continuous, with Lipschitz constant controlled by $C_{\text{topo}}L_\gamma$, depending only on the operator norm of the topological projection operator $T$ and the Lipschitz constant of the TD field with respect to $\gamma$.

Finally, we explain that the associated integrable component is also Lipschitz continuous in $\gamma$. For each $\gamma$, the integrable quantity is $f_{\text{ex}}^{(\gamma)} := du_V^{\star,(\gamma)} \in C^1$. For two discount factors $\gamma_1, \gamma_2$, we have

$$f_{\text{ex}}^{(\gamma_1)} - f_{\text{ex}}^{(\gamma_2)} = du_V^{\star,(\gamma_1)} - du_V^{\star,(\gamma_2)} = d\big(u_V^{\star,(\gamma_1)} - u_V^{\star,(\gamma_2)}\big), \tag{233}$$

using the linearity of $d$. Since $d : C^0 \to C^1$ is bounded, there exists a constant $\|d\|_{\text{op}}$ such that for all $u \in C^0$,

$$\|du\|_{C^1} \le \|d\|_{\text{op}} \|u\|_{C^0}. \tag{234}$$

Thus,

$$\|f_{\text{ex}}^{(\gamma_1)} - f_{\text{ex}}^{(\gamma_2)}\|_{C^1} = \big\|d\big(u_V^{\star,(\gamma_1)} - u_V^{\star,(\gamma_2)}\big)\big\|_{C^1} \le \|d\|_{\text{op}} \|u_V^{\star,(\gamma_1)} - u_V^{\star,(\gamma_2)}\|_{C^0}. \tag{235}$$

Substituting the previously obtained (20), we obtain

$$\|f_{\text{ex}}^{(\gamma_1)} - f_{\text{ex}}^{(\gamma_2)}\|_{C^1} \le \|d\|_{\text{op}} C_{\text{topo}} L_\gamma |\gamma_1 - \gamma_2|, \tag{236}$$

which shows that the mapping $\gamma \mapsto du_V^{\star,(\gamma)}$ is also Lipschitz continuous in $C^1$. In summary, the optimal potential $u_V^{\star,(\gamma)}$ and its associated integrable TD structure are both Lipschitz continuous with respect to the discount factor $\gamma$, and thus the theorem is completely proved.

$\square$

### D.10. Proof of Theorem 5.3

*Proof.* In this theorem, we regard the parameter space as a finite-dimensional vector space equipped with the Euclidean norm $\|\cdot\|$; $C^1$ is a Hilbert space with respect to the reference measure $\mu_\pi$, and its norm is defined by $\|f\|_{C^1}^2 := \mathbb{E}_{\mu_\pi}[f(S, A, S')^2]$. Let $\delta_{V_\theta} \in C^1$ be the TD error induced by $V_\theta$, and denote its Hodge-type decomposition as

$$\delta_{V_\theta} = du_{V_\theta}^\star + \delta_{V_\theta}^{\text{res}}, \tag{237}$$

where $du_{V_\theta}^\star$ is the exact component and $\delta_{V_\theta}^{\text{res}}$ is the topological residual. According to the definition in the theorem,

$$g_{\text{TD}}(\theta) := \mathbb{E}_{\mu_\pi}\big[\delta_{V_\theta}(S, A, S')\nabla_\theta V_\theta(S)\big], \qquad g_{\text{HFPS}}(\theta) := \mathbb{E}_{\mu_\pi}\big[\tilde{\delta}_{V_\theta}(S, A, S')\nabla_\theta V_\theta(S)\big], \tag{238}$$

where $\tilde{\delta}_{V_\theta} := du_{V_\theta}^\star$ is the integrable part of the TD error. Note that

$$\delta_{V_\theta} = \tilde{\delta}_{V_\theta} + \delta_{V_\theta}^{\text{res}}, \tag{239}$$

hence

$$\tilde{\delta}_{V_\theta} - \delta_{V_\theta} = -\delta_{V_\theta}^{\mathrm{res}}. \tag{240}$$

Using this, we can directly write out the difference between the two update directions:

$$
\begin{aligned}
g_{\mathrm{HFPS}}(\theta) - g_{\mathrm{TD}}(\theta) &= \mathbb{E}_{\mu_\pi}\big[(\tilde{\delta}_{V_\theta}(S,A,S') - \delta_{V_\theta}(S,A,S'))\nabla_\theta V_\theta(S)\big] \\
&= -\mathbb{E}_{\mu_\pi}\big[\delta_{V_\theta}^{\mathrm{res}}(S,A,S')\nabla_\theta V_\theta(S)\big].
\end{aligned}
\tag{241}
$$

This shows that $g_{\mathrm{HFPS}}(\theta) - g_{\mathrm{TD}}(\theta)$ is the expectation under $\mu_\pi$ of a random vector $X(S,A,S') := -\delta_{V_\theta}^{\mathrm{res}}(S,A,S')\,\nabla_\theta V_\theta(S)$. Since $\nabla_\theta V_\theta(S)$ satisfies $\|\nabla_\theta V_\theta(S)\| \leq B$ almost everywhere, and $\delta_{V_\theta}^{\mathrm{res}} \in L^2(\mu_\pi)$, it is clear that $X$ is integrable. Taking the norm of its expectation and using convexity of the norm (i.e., Jensen's inequality) yields

$$\|g_{\mathrm{HFPS}}(\theta) - g_{\mathrm{TD}}(\theta)\| = \big\|\mathbb{E}_{\mu_\pi}[X(S,A,S')]\big\| \leq \mathbb{E}_{\mu_\pi}\big[\|X(S,A,S')\|\big]. \tag{242}$$

Substituting the definition of $X$, we obtain

$$\|X(S,A,S')\| = \big|\delta_{V_\theta}^{\mathrm{res}}(S,A,S')\big|\,\big\|\nabla_\theta V_\theta(S)\big\|, \tag{243}$$

Thus,

$$\|g_{\mathrm{HFPS}}(\theta) - g_{\mathrm{TD}}(\theta)\| \leq \mathbb{E}_{\mu_\pi}\Big[\big|\delta_{V_\theta}^{\mathrm{res}}(S,A,S')\big|\,\big\|\nabla_\theta V_\theta(S)\big\|\Big]. \tag{244}$$

Using the boundedness assumption $\|\nabla_\theta V_\theta(S)\| \leq B$ almost everywhere, we may pull it outside the expectation:

$$\mathbb{E}_{\mu_\pi}\Big[\big|\delta_{V_\theta}^{\mathrm{res}}(S,A,S')\big|\,\big\|\nabla_\theta V_\theta(S)\big\|\Big] \leq B\,\mathbb{E}_{\mu_\pi}\Big[\big|\delta_{V_\theta}^{\mathrm{res}}(S,A,S')\big|\Big], \tag{245}$$

Thus,

$$\|g_{\mathrm{HFPS}}(\theta) - g_{\mathrm{TD}}(\theta)\| \leq B\,\mathbb{E}_{\mu_\pi}\big[|\delta_{V_\theta}^{\mathrm{res}}(S,A,S')|\big]. \tag{246}$$

What remains is a standard relation between $L^1$ and $L^2$. Viewing $\delta_{V_\theta}^{\mathrm{res}}$ as a random variable $Y(S,A,S') := \delta_{V_\theta}^{\mathrm{res}}(S,A,S')$ in $L^2(\mu_\pi)$, the Cauchy–Schwarz inequality (here applied to the functions $|Y|$ and the constant $1$) gives

$$\mathbb{E}_{\mu_\pi}[|Y|] = \mathbb{E}_{\mu_\pi}[|Y|\cdot 1] \leq \big(\mathbb{E}_{\mu_\pi}[Y^2]\big)^{1/2}\big(\mathbb{E}_{\mu_\pi}[1^2]\big)^{1/2} = \big(\mathbb{E}_{\mu_\pi}[Y^2]\big)^{1/2}, \tag{247}$$

since $\mathbb{E}_{\mu_\pi}[1] = 1$. Substituting $Y = \delta_{V_\theta}^{\mathrm{res}}$ gives

$$\mathbb{E}_{\mu_\pi}\big[|\delta_{V_\theta}^{\mathrm{res}}(S,A,S')|\big] \leq \Big(\mathbb{E}_{\mu_\pi}\big[\delta_{V_\theta}^{\mathrm{res}}(S,A,S')^2\big]\Big)^{1/2} = \|\delta_{V_\theta}^{\mathrm{res}}\|_{C^1}, \tag{248}$$

where the last equality is exactly the definition of the $C^1$ norm. Substituting this bound into the previous inequality produces

$$\|g_{\mathrm{HFPS}}(\theta) - g_{\mathrm{TD}}(\theta)\| \leq B\,\|\delta_{V_\theta}^{\mathrm{res}}\|_{C^1}, \tag{249}$$

which is precisely the bias bound stated in (23). This shows that the update bias introduced by HFPS using only the integrable TD component is controlled proportionally by the $C^1$ norm of the topological residual. The theorem is proved.

$\square$

### D.11. Proof of Corollary 5.4

*Proof.* Fix $\theta$. By Corollary 3.5, the TD error admits the orthogonal decomposition

$$\delta V_\theta = du_{V_\theta}^\star + \delta_{V_\theta}^{\mathrm{res}}, \qquad \delta_{V_\theta}^{\mathrm{res}} \in \mathcal{E}^\perp,$$

and moreover $\delta_{V_\theta}^{\mathrm{res}} = 0$ if and only if $\delta V_\theta \in \mathcal{E}$ (i.e., $V_\theta$ is topologically integrable). Under the integrability assumption, $\delta_{V_\theta}^{\mathrm{res}} = 0$ holds, hence

$$\delta V_\theta = du_{V_\theta}^\star \quad \mu^\pi\text{-almost surely on } (S,A,S').$$

Therefore, for $\mu^\pi$-almost every triple $(S,A,S')$,

$$du_{V_\theta}^\star(S,A,S')\,\nabla_\theta V_\theta(S) = \delta V_\theta(S,A,S')\,\nabla_\theta V_\theta(S).$$

Taking expectations with respect to $\mu^\pi$ and using the definitions

$$g_{\text{TD}}(\theta) := \mathbb{E}_{\mu^\pi}\left[\delta V_\theta(S, A, S')\,\nabla_\theta V_\theta(S)\right], \qquad g_{\text{HFPS}}(\theta) := \mathbb{E}_{\mu^\pi}\left[du^\star_{V_\theta}(S, A, S')\,\nabla_\theta V_\theta(S)\right],$$

we obtain $g_{\text{HFPS}}(\theta) = g_{\text{TD}}(\theta)$.

Equivalently, one can conclude the same fact directly from the deviation bound in Theorem 5.3:

$$\|g_{\text{HFPS}}(\theta) - g_{\text{TD}}(\theta)\| \le B\|\delta^{\text{res}}_{V_\theta}\|_{C^1} = 0.$$

Finally, since the update directions coincide pointwise for all $\theta$, the two deterministic recursions are identical for the same initialization and step sizes. This implies that any convergence guarantee established for the TD semi-gradient recursion under a given set of assumptions carries over verbatim to HFPS in the integrable regime. $\qquad\square$

### D.12. Proof of Proposition 5.5

*Proof.* Define $x_t := \theta_t - \theta_{\text{TD}}$. By the strong monotonicity assumption, $g_{\text{TD}}(\theta_t) = Ax_t$. Let

$$e_t := g_{\text{HFPS}}(\theta_t) - g_{\text{TD}}(\theta_t).$$

By Theorem 5.3 in the main text (equation 23), using $\|\nabla_\theta V_\theta(S)\| = \|\phi(S)\| \le B$ and $\|\delta^{\text{res}}_{V_{\theta_t}}\|_{C^1} \le \varepsilon$, we have the uniform bound

$$\|e_t\| = \|g_{\text{HFPS}}(\theta_t) - g_{\text{TD}}(\theta_t)\| \le B\varepsilon, \qquad \forall t. \tag{250}$$

The recursion becomes

$$x_{t+1} = x_t - \alpha_t(Ax_t + e_t).$$

Let $V_t := \|x_t\|^2$. Expanding and bounding yields

$$\begin{aligned}
V_{t+1} &= \|x_t - \alpha_t(Ax_t + e_t)\|^2 \\
&= \|x_t\|^2 - 2\alpha_t x_t^\top Ax_t - 2\alpha_t x_t^\top e_t + \alpha_t^2\|Ax_t + e_t\|^2 \\
&\le V_t - 2\alpha_t\lambda V_t + 2\alpha_t\|x_t\|\,\|e_t\| + \alpha_t^2\cdot 2\|Ax_t\|^2 + \alpha_t^2\cdot 2\|e_t\|^2 \\
&\le V_t - 2\alpha_t\lambda V_t + 2\alpha_t\sqrt{V_t}\,(B\varepsilon) + 2\alpha_t^2 L^2 V_t + 2\alpha_t^2(B\varepsilon)^2,
\end{aligned}$$

where we used $x_t^\top Ax_t \ge \lambda\|x_t\|^2 = \lambda V_t$, $\|Ax_t\| \le L\|x_t\| = L\sqrt{V_t}$, and (250). Next apply the inequality $2ab \le \lambda a^2 + \frac{1}{\lambda}b^2$ with $a = \sqrt{V_t}$ and $b = B\varepsilon$:

$$2\alpha_t\sqrt{V_t}\,(B\varepsilon) \le \alpha_t\lambda V_t + \alpha_t\frac{B^2}{\lambda}\varepsilon^2.$$

Substituting gives

$$V_{t+1} \le \left(1 - \lambda\alpha_t + 2L^2\alpha_t^2\right)V_t + \alpha_t\frac{B^2}{\lambda}\varepsilon^2 + 2\alpha_t^2 B^2\varepsilon^2.$$

Using the step size upper bound $\alpha_t \le \frac{\lambda}{4L^2}$ implies $2L^2\alpha_t^2 \le \frac{\lambda}{2}\alpha_t$, hence

$$V_{t+1} \le \left(1 - \tfrac{\lambda}{2}\alpha_t\right)V_t + \alpha_t\frac{B^2}{\lambda}\varepsilon^2 + 2\alpha_t^2 B^2\varepsilon^2. \tag{251}$$

We now invoke a deterministic sequence lemma D.1 with $c = \lambda/2$, $d = (B^2/\lambda)\varepsilon^2$, and $\beta_t := 2\alpha_t^2 B^2\varepsilon^2$. Because $\sum_t \alpha_t^2 < \infty$, we have $\sum_t \beta_t < \infty$. Applying the lemma to (251) yields

$$\limsup_{t\to\infty} V_t \le \frac{d}{c} = \frac{(B^2/\lambda)\varepsilon^2}{\lambda/2} = \frac{2B^2}{\lambda^2}\varepsilon^2.$$

Taking square roots gives

$$\limsup_{t\to\infty}\|\theta_t - \theta_{\text{TD}}\| = \limsup_{t\to\infty}\|x_t\| \le \sqrt{2}\,\frac{B}{\lambda}\,\varepsilon,$$

which proves the claim. The case $\varepsilon = 0$ follows immediately.

**Lemma D.1** (sequence bound). *Let $\{V_t\}_{t \geq 0}$ be a nonnegative sequence satisfying*

$$V_{t+1} \leq (1 - c\alpha_t)V_t + \alpha_t d + \beta_t, \qquad t \geq 0,$$

*where $c > 0$, $d \geq 0$, $\alpha_t > 0$ with $\sum_t \alpha_t = \infty$ and $\alpha_t \to 0$, and $\sum_t \beta_t < \infty$ with $\beta_t \geq 0$. Then $\limsup_{t \to \infty} V_t \leq d/c$.*

*proof of lemma D.1.* Fix any $\eta > 0$ and define $M := d/c + \eta$. Because $\alpha_t \to 0$, there exists $T$ such that for all $t \geq T$, $c\alpha_t \leq 1$ and $\beta_t \leq \frac{1}{2}c\alpha_t\eta$. Consider any $t \geq T$ with $V_t \geq M$. Then

$$V_{t+1} \leq (1 - c\alpha_t)V_t + \alpha_t d + \beta_t \leq V_t - c\alpha_t(V_t - d/c) + \beta_t \leq V_t - c\alpha_t\eta + \tfrac{1}{2}c\alpha_t\eta = V_t - \tfrac{1}{2}c\alpha_t\eta.$$

Thus, whenever $V_t \geq M$ for $t \geq T$, the sequence decreases by at least $\frac{1}{2}c\alpha_t\eta$. If $V_t \geq M$ occurs infinitely often for $t \geq T$, summing the above decreases over those times would force $V_t$ to decrease without bound because $\sum_t \alpha_t = \infty$, contradicting nonnegativity of $V_t$. Hence there exists $T_\eta$ such that $V_t < M$ for all $t \geq T_\eta$, i.e., $\limsup_{t \to \infty} V_t \leq d/c + \eta$. Since $\eta > 0$ is arbitrary, $\limsup_{t \to \infty} V_t \leq d/c$. $\square$

$\square$

$\square$

