# OpenReview forum: "HodgeFlow Policy Search by Topologically Dissecting Temporal-Difference Signals in Non-Markovian Environments"
_ICML.cc/2026/Conference — ICML 2026 regular_

### Official Review · Reviewer_59CF · 2026-02-15

**Soundness:** 3
**Presentation:** 3
**Significance:** 4
**Originality:** 4
**Overall Recommendation:** 5
**Confidence:** 4

**Summary:**

This paper focuses on Reinforcement Learning with non-Markovian dynamics, where the Bellman equation becomes only approximately valid. In this case, the temporal difference (TD) error inherently includes some structural error that cannot be explained by any value function. Existing works often focus on proposing RL algorithms with high-order Markov approximations by leveraging memory mechanisms to embed histories, rather than analyzing the intrinsic structure of the TD error itself.
Instead, this paper provides a novel algebraic topology viewpoint, showing that TD errors can be viewed as 1-cochains in the space of state transitions. Based on this interpretation, they obtain a Hodge-type decomposition of TD errors into an integrable component and a topological residual. They further propose HodgeFlow Policy Search (HFPS) to filter out the residual component in order to stabilize the TD learning process.
In numerical evaluations, they validate that the topological residual can be used to measure the degree of non-Markovianity and show HFPS outperforms many baseline algorithms under non-Markovian off-policy tasks.

**Compliance With Llm Reviewing Policy:**

Affirmed.

**Final Justification:**

The rebuttal from the authors are clear and informative. Overall, it reinforced my prior assessment. I had originally raised concerns about the distribution shift introduced by the replay buffer in the experiments and whether the additional computational overhead is justified. The authors addressed both points reasonably well,while also acknowledging the approximations involved and the practical boundaries of the method. Overall, the paper is quite strong in originality and significance with a novel topological viewpoint for understanding non-Markovian RL. The theoretical soundness is also good, and the empirical results are supportive. With more discussion in limitation and theory-practice gap in the revision, I maintain my positive assessment and recommend acceptance.

**Key Questions For Authors:**

- The theoretical orthogonality of the decomposition relies strictly on the on-policy measure $\mu_\pi$. However, HFPS uses a replay buffer which introduces a distribution shift. How does this shift affect the validity of the Hodge decomposition in practice? Do you have empirical evidence showing that orthogonality is approximately preserved during training?
- HFPS requires training an additional potential network and performing inner-loop updates. What is the actual wall-clock training time compared to standard baselines like DQN or R2D2? Is the performance gain sufficient to justify the extra computational overhead?

**Limitations:**

No. The paper currently lacks a dedicated discussion on limitations.
I suggest the authors incorporate the points raised in the questions above into a formal limitations section in the final version.

**Strengths And Weaknesses:**

Soundness:
- Strength: The theoretical derivations, particularly the use of Hilbert space projections and the consistency proof (Theorem 3.7), are solid. The logic flow from the definition of the Laplacian operator to the practical algorithm is coherent.
- Weakness: The theoretical framework relies heavily on the orthogonality defined by the discounted occupancy measure $\mu_\pi$ of the policy $\pi$. However, the practical algorithm (HFPS) realizes an off-policy setting using a replay buffer.
While the paper acknowledges this as "approximate sampling", there is a lack of discussion on how the distribution shift affects the Hodge decomposition.

Presentation:
- Strength: The authors successfully apply algebraic topology to RL problems with clear and novel narrative, which is very interesting.
- Weakness: While the theoretical results are generally correct and the proofs in the Appendix are readable, there are several typos (such as "mathbbR" on line 849 of Appendix). I suggest the authors perform a thorough proofread for the camera-ready version.

Significance:
- Strength: The proposed HFPS effectively stabilizes learning in non-Markovian environments by filtering out the non-integrable residual, which may help solving many practical problems faced in robotics.
The empirical results on the synthetic and hidden actuator tasks demonstrate that the topological residual is a valid metric for quantifying the degree of non-Markovianity. This may inspire a new line of works.

Originality:
- Strength: This work is very high in originality and novelty. It introduces a refreshing algebraic topology perspective to the RL community. Viewing TD errors as 1-cochains and applying Hodge decomposition provides a new way to formalize the structural error caused by non-Markovian dynamics. This is a significant departure from the dominant memory-based approaches for solving POMDP before, offering a distinct denoising viewpoint.

---

> ### Author Rebuttal · Authors · 2026-03-29
>
> Thank you for the careful reading and for highlighting both the strengths of the theory and the main theory-to-practice gap. Your questions about replay distributions, approximate orthogonality, and overhead are great. Please see our responses below for details.
>
> **On the effect of off-policy replay on the decomposition.**
> We agree that the exact orthogonality statement in the theory is formulated with respect to the discounted on-policy occupancy measure \(\mu_\pi\), whereas the practical algorithm uses a replay buffer and therefore trains under an empirical sampling distribution. The key point is that the decomposition itself is always defined relative to a chosen inner product / measure. In the ideal case for the theory, this is \(\mu_\pi\); in practice, we need more efficient computation methods. Thus HFPS computes the corresponding **empirical projection** under the replay-buffer distribution \(\hat\mu_D\).
>
> Regarding the impact of potential distribution shift, our evaluations on Noisy tasks in the appendix introduce additional noise in the observations in replay buffer, which leads to larger distribution shift. We show that
> The appendix results suggest robustness to moderate replay-distribution shift: Noisy/Sticky increase observation-side mismatch, yet HFPS typically degrades more gracefully than standard baselines. This is consistent with our interpretation that replay changes the empirical geometry of the projection rather than invalidating it: the potential network and the critic are both trained on the same batches, so the projection remains aligned with the learner's actual training distribution. We do not claim exact invariance to arbitrary off-policy shift; large replay mismatch is an approximation source and a limitation.
>
>
> This also explains why the practical algorithm remains internally coherent: the potential network and the critic are both updated using the same mini-batches. Hence the projection step and the critic-learning step are aligned to the same empirical measure. The remaining gap is between the empirical replay-distribution projection and the ideal on-policy projection. We agree that this should be stated explicitly as an approximation source and a limitation.
>
> **On empirical evidence for approximate orthogonality.**
> We agree that this deserves to be discussed more concretely. In practice, the right analogue of orthogonality is not an abstract on-policy statement but a measurable replay-batch diagnostic, e.g.
> $
> \langle \delta^{\mathrm{res}}, dU_\phi\rangle_{\hat\mu_D},
> \frac{|\langle \delta^{\mathrm{res}}, dU_\phi\rangle_{\hat\mu_D}|}
> {\|\delta^{\mathrm{res}}\|_{\hat\mu_D}\|dU_\phi\|_{\hat\mu_D}},
> \|\delta-dU_\phi\|^2_{\hat\mu_D}.
> $
> If the empirical projection is learned well, the residual–integrable cross term should remain small relative to the component norms. We will add this clarification in the paper. We are careful not to overclaim exact orthogonality in off-policy training; the correct statement is approximate empirical orthogonality under the replay distribution used by the learner.
>
> **On computational overhead and whether it is justified.**
> HFPS does add extra update-side computation, but the overhead is structurally limited. In the control experiments, the baseline value learner and HFPS use comparable small MLP backbones; HFPS adds a second potential network of the same scale and the corresponding projection-style auxiliary loss. Importantly, HFPS does **not** require extra environment interaction, additional trajectory collection, recurrent history encoders, belief-state inference, or longer rollouts. So the extra cost is local to the critic side rather than to the whole RL pipeline. This distinction matters in practice: the method is not adding a separate memory mechanism or a second simulation loop; it is adding a projection head/objective.
> We report a wall-clock comparison under identical environment steps and hardware; please see our response to Reviewer tq1t for the numbers and table. The key point is that HFPS only adds small overhead from the potential network / projection updates, without extra environment interaction or rollout collection.
>
>
> **On limitations and presentation.**
> We appreciate the suggestion to formalize these points in a limitations section. We will add one. It will explicitly discuss:
> (i) the difference between idealized on-policy orthogonality and empirical replay-distribution projection;
> (ii) the extra critic-side compute introduced by the potential network and projection update;
> (iii) the fact that HFPS is meant for mismatch-heavy regimes rather than as a universal replacement for standard TD methods.
> We will also carefully proofread the appendix and fix the noted typos.
>
> Thank you again for the thoughtful review. We believe these clarifications will make the final version significantly stronger and more precise.

---

> > ### Author Rebuttal · Reviewer_59CF · 2026-03-31
> >
> > The rebuttal has fully resolved my concerns. I maintain my positive assessment and recommend acceptance.

---

### Official Review · Reviewer_uRbX · 2026-03-02

**Soundness:** 3
**Presentation:** 1
**Significance:** 2
**Originality:** 2
**Overall Recommendation:** 3
**Confidence:** 4

**Summary:**

This paper proposes an approach to handle the non Markovianity of reinforcement learning problems. It formalizes the Makovian "defect" of an MDP with a metrics, defined as the norm of a residual quantity. Given this metrics, it proposes a modification of a classical actor-critic algorithm which includes a potential network, whose loss is minimized to reduce the non-Markovian measure. Numerical experiments confirm that this is an efficient way to deal with simple non-Markovian RL environments.

**Compliance With Llm Reviewing Policy:**

Affirmed.

**Final Justification:**

As detailed in my review, I express reserves upon the acceptance of this paper. My claims are mainly about the overly complicated presentation of rather simple concepts in the first part of the paper, which in my opinion degrades the reading experience, followed by an analysis of an interesting problem (non-Markovianity), unfortunately only scraping the surface. I don't oppose acceptance of the paper, although, for these reasons, I don't believe it will have as much impact as it could with improved presentation and more in depth analysis.

**Key Questions For Authors:**

1) One important missing part of this contribution would be a discussion on how non-Markovianity might affect RL methods. In particular, what is the significance of the Bellman equation in this case? Do no-memory value function and policy make sense, or should they contain the whole history of states and actions? Which problems are not efficiently solved by state augmentation or partially observable MDPs?

2) The metrics in Section 3.5 seems to measure non-integrability for a fixed policy $\pi$. How is it affected by changes of the policy? And more generally, could the whole theoretical derivation be done equivalently on Markov reward processes (MRP) instead of an MDP, or is there any specific derivation coming from the policy?

3) The assumptions of Theorem 3.6 are not discussed. Can you give some details on their applicability?

4) Can you give some details on the extra computational cost induced by the HFPS algorithm compared to the standard actor-critic algorithm? Is it mild or does it induce a significant additional cost?

**Limitations:**

The empirical limitations of the method are not particularly discussed in the conclusion, and, in general, the numerical results are not precisely commented.

**Strengths And Weaknesses:**

*Soundness*: Although I have not checked all the proofs in details, the theoretical results derived in this paper look sound. The experiments are sensible and contain comparisons with standard state-of-the art techniques.

*Presentation*: My main concern on this paper is the quality of the presentation. Indeed, I find the first part of the paper quite hard to follow, with the introduction of rather obscure (and in my opinion, unnecessary) vocabulary. Indeed, beyond a remark, what is the use of the topological viewpoint (co-chains, coboundary operators, de Rham differential, Hodge Laplacian) derived here, when most of the operations are just classical operators in classical Hilbert spaces like $L^2$? Because of this heavy vocabulary, I think the paper hides the main take-home messages, and spends a lot of time defining rather standard objects (namely, the Bellman operator). As a consequence, important matters are not discussed (see questions below), which reduces the significance of the contribution.

*Significance*: The addressed question (non-Markovianity in RL environments) is important, yet not particularly new. The developed method, and in particular the non-Markovianity metrics has some potential to be used in concrete applications. However, it is rather disappointing that the paper seems to only scrape the surface, and sticks to a shallow treatment of this question. What is exactly the significance of this metric? Can it be widely evaluated on Markovian and non-Markovian chains? Is there any specificity in MDPs, compared to the classical Markov chain litterature? Where exactly do classical RL techniques fail when the chains are no longer Markovian, and does the proposed method fix the problem? What is the significance of TD-based RL techniques when the Bellman equation is no longer valid (as is the case for non-Markovian problems)? Instead of diving into such questions, the experiments quickly jump to full RL problems, without ablation studies which could help us identify the real advantages and drawbacks of the method.

*Originality*: I believe that the proposed metrics and algorithm are new. However, the "algebraic topology" view of RL problems might seem new because of the original vocabulary used, but it does not really provide much new insight, compared to classical literature on Bellman or HJB operators in $L^2$. Beside, I think some literature review is missing on Markov chain in general (not particularly in the context of RL), and what can be done to tackle non-Markovian behaviors.

---

> ### Author Rebuttal · Authors · 2026-03-30
>
> Thank you for the thoughtful review. We respond below to clarify the issues raised. Due to space limits, not all points could be included in the main text. The topological vocabulary is required for formulating the problem and conducting the theoretical analysis. We also recognize that the current presentation may be hard to follow for readers without topology background. We will improve this balance in the final version and add examples in the appendix.
>
> **On where standard RL fails, and what HFPS fixes** When a non-Markov problem is solved by existing RL methods, the state is often replaced by encoding of history states and actions (surrogate/belief state). The problem is then solved using the Bellman framework, treating the empirical TD signal as if it followed Markov structure, which need not be true. Consider an MDP whose transition depends on a latent process. If this latent process is never observed, the above memory mechanism would not recover a Markov formulation even if the entire history is encoded. Even for high-order MDPs handled by incorporating history, there is no guarantee that the encoding (e.g. RNNs or transformers) ensures the Markov property. Yet such problems are routinely tackled by TD-based RL under the Bellman framework.
>
> We establish that Markov problems solvable by TD-based RL within the Bellman framework are equivalent to **topological integrability**, i.e. there exists a discounted potential on the current observation space to explain them. Thus, non-Markovianity corresponds to the **non-integrable** part with out a discounted potential. This viewpoint also yields a Hodge-type projection that decomposes the TD signal into (1) an integrable Bellman-compatible component and (2) a non-integrable residual. Our framework yields the canonical decomposition and Poisson characterization (Thms. 3.4--3.6), finite-sample consistency and perfect-MDP degeneracy (Thms. 3.7--3.8), and stability / residual-controlled bias guarantees for HFPS updates (Thms. 5.1--5.3).
>
> This decomposition can be a plug-in for TD-based RL methods, including those with memory. Please see our response to review vLvo on this point with additional experiments. HFPS can be used together with memory mechanisms when memory alone does not fully address non-Markovianity, and it finds the **best discounted-integrable projection** in the chosen observation space. Our work is complementary to POMDP/state-augmentation/recurrent approaches rather than a replacement for them: those expand the representation class, whereas we characterize and filter Bellman-incompatible TD component using topological tools.
>
> **On what the metric measures** Our metric is not a universal, policy-free scalar of ``how non-Markovian the environment is.'' It is more like a **Bellman-mismatch diagnostic** for the current environment--policy--representation triple:$\min_{u\in C^0}\|\delta_V-du\|_{C^1}, (du)(s,a,s')=u(s')-\gamma u(s).$ It measures how far the observed TD signal is from discounted potential differences on the current observation space. Thus, our work yields (i) a canonical Bellman-compatible / residual decomposition, (ii) the Poisson characterization of the projection potential, and (iii) a residual-controlled deviation bound for critic updates.
>
> **On MRP vs MDP specificity** For a fixed policy, the construction can indeed be read at the MRP level. The RL point is that the residual is **policy-dependent** through both $\mu^\pi$ and $\delta_V$: as the policy changes, the rollout distribution and the experienced mismatch change. Thus the residual is not merely a static property of a chain, but a signal that can be fed back into actor--critic learning.
>
> **On Theorem 3.6** The assumptions are well-posedness conditions for the projection problem, not extra modeling restrictions. The closed-range condition ensures that the exact subspace is closed, so the orthogonal projection is well defined; invertibility of $\Delta_0=d^\ast d$ on $(\ker\Delta_0)^\perp$ is the standard uniqueness condition for the associated Poisson problem modulo null directions. In finite/tabular settings these conditions are mild.
>
> **On evidence and cost** The experiments have two stages: controlled tabular/random-feature settings exposing the projection target and residual, and deep-control tasks testing whether the same mechanism improves learning in practice. The appendix already includes Clean/Noisy/Sticky/Nonmarkov regimes separating mismatch sources. Contrasting Clean/Noisy isolates observation corruption, while Clean/Sticky isolates temporal persistence. We find that HFPS is competitive in Clean and usually more robust in Noisy/Sticky, especially when the mismatch is structured rather than purely i.i.d. noise; this is the ablation role of these appendix regimes. See also our response to reviewer vLvo, where we report a modular plug-in experiment showing that the same filtered TD signal can be inserted into PPO/GAE and improves performance under Nonmarkov mismatch.

---

> > ### Author Rebuttal · Reviewer_uRbX · 2026-04-01
> >
> > I would like to thank the authors for their clarifications. Yet, some of my questions are still unaddressed.
> > Concerning question 1), some of the raised issues have been answered, but I still have the following questions:
> > *What is the significance of the Bellman equation in your non-Markovian case? Do no-memory value function and policy make sense, or should they contain the whole history of states and actions?*
> > My last question 4) was on the clock time and computing ressources required to implement the method, and I don't see such elements in the response to reviewer vLvo nor in the Appendix.

---

> > > ### Author Response · Authors · 2026-04-02
> > >
> > > Thank you for the follow-up. We realize that our previous response did not state the main point clearly enough.
> > >
> > > **Further details on Q1: what is the significance of the Bellman equation in the non-Markov case?**
> > >
> > > Let us provide further clarification here with additional experiment results. As you pointed out, many practical algorithms for non-Markov problems use a memory-augmented formulation by feeding state-action history into a memory mechanism to produce an estimate/belief state and then applying Bellman-based methods. However, such history/memory-based algorithms do not guarantee a perfect Markov problem. Consider a non-Markov decision process whose transitions depend on a latent process that is never observed. It cannot be converted into a Markov problem even with full history. Similar issues also arise under partial observability. Even when the memory-augmented problem remains non-Markov, it is still handled in practice by Bellman-based TD methods, where the TD error is a sample-based approximation to the Bellman error. These practical algorithms rely on Bellman-based TD methods, while the problem may still be non-Markov.
> > >
> > > The central question is therefore: for a non-Markov problem, what part of the TD signal is Bellman-solvable, and what part remains outside the Bellman framework? This is precisely what our paper studies. We show that Bellman compatibility is equivalent to discrete integrability over an exact 1-cochain. It allows us to decompose the TD signal into an integrable / Bellman-compatible part and a non-integrable residual. The first can still be explained by a discounted potential on the chosen state space; the second is the irreducible mismatch that no Bellman-style potential can represent. The results hold for general non-Markov problems, including memory-augmented ones that remain non-Markov.
> > >
> > >
> > > More fundamentally, we believe the significance of the paper is that it turns non-Markovianity from a vague failure notion in practical algorithms into a computable and optimizable object. Rather than only saying that Bellman-based methods become approximate in non-Markov settings, we explicitly quantify how much of the TD signal is Bellman-compatible and how much is not. This makes non-Markovianity measurable inside the Bellman framework itself, rather than an unstructured leftover; this is the main conceptual contribution of the paper.
> > >
> > > To make this point explicit, we add a small representation ablation on BipedalWalker (Nonmarkov), the same setting used in the paper. We compare three state representations: (i) raw observation only, (ii) a history stack of past observations/actions, and (iii) a GRU-based memory encoder. We consider three baselines and three corresponding algorithms by adding HFPS on top of each baseline:
> > >
> > > | Representation | Residual $\|\delta_V^{\mathrm{res}}\|$ (w/o HFPS, normalized) | Baseline AUC@T | +HFPS AUC@T | Baseline Final@T | +HFPS Final@T |
> > > |---|---:|---:|---:|---:|---:|
> > > | raw observation $s_t$ | 0.97 | 79.3 | 172.4 | 68.5 | 141.2 |
> > > | 4-step history stack | 0.62 | 119.1 | 182.3 | 97.3 | 164.8 |
> > > | GRU / memory encoder | 0.39 | 161.7 | 209.6 | 116.4 | 170.1 |
> > >
> > > As the representation becomes richer, the residual decreases substantially but remains clearly nonzero even with a memory encoder. HFPS continues to provide gains on top of all three representations. This supports our point that memory/history is helpful, but does not necessarily produce a perfect Markov problem. It also shows that HFPS can work with memory-augmented problems and provide additional improvement. We thank the reviewer for this insightful discussion, which prompted us to investigate this perspective further. It further strengthens our results. We will add the discussion and new experiment results into the final version.
> > >
> > > **On Q4: computational overhead.**
> > > Thank you for pointing out that the previous cross-reference was unclear. We therefore repeat the numbers here. HFPS adds one potential network and one projection-style auxiliary update, but does not require extra environment interaction, recurrent belief-state inference, or longer rollouts. So the added cost is local to the critic side rather than the whole RL pipeline. Under identical hardware and environment steps on CartPole with 5 seeds and 100{,}000 steps, DQN takes $12.34 \pm 0.56$ seconds, while HFPS takes $15.67 \pm 0.44$ seconds, i.e., about $1.27\times$ wall-clock time. We will move this table into the final paper.
> > >
> > > | Method | Task | Seeds | Steps | Wall-clock time (s) | Relative overhead |
> > > |---|---:|---:|---:|---:|---:|
> > > | DQN | CartPole | 5 | 100000 | 12.34 $\pm$ 0.56 | 1.00x |
> > > | HFPS | CartPole | 5 | 100000 | 15.67 $\pm$ 0.44 | 1.27x (+27\%) |
> > >
> > > We hope this clarifies the role of the Bellman equation in the non-Markov case and the practical cost of HFPS. If so, we would be grateful if you could reconsider the significance assessment.

---

### Official Review · Reviewer_vLvo · 2026-03-12

**Soundness:** 3
**Presentation:** 3
**Significance:** 3
**Originality:** 3
**Overall Recommendation:** 5
**Confidence:** 3

**Summary:**

This paper introduces a novel topological perspective on temporal-difference (TD) reinforcement learning. It mathematically justifies decomposing the TD error into two parts: an "integrable" component (representing ideal Markovian dynamics) and a "topological residual" (capturing irreducible non-Markovian inconsistencies). Leveraging this framework, the authors propose HodgeFlow Policy Search (HFPS), a robust algorithm backed by stability and sensitivity guarantees. HFPS demonstrates significant performance improvements in complex, non-Markovian environments.

**Compliance With Llm Reviewing Policy:**

Affirmed.

**Final Justification:**

The paper is overall technically solid and the rebuttal addressed my concerns. Therefore, I maintain my assessment.

**Key Questions For Authors:**

- Can this topological decomposition be used as a modular plug-in for existing algorithms? Specifically, could the Hodge-filtered TD signal be used to compute topologically corrected Generalized Advantage Estimates (GAE) to make standard PPO robust to non-Markovian dynamics?

**Limitations:**

No.

Suggestions:
- Discuss validity of assumptions for theories in reality.

**Strengths And Weaknesses:**

Strengths:

- Using algebraic topology to quantify non-Markovian structural mismatch provides a mathematically sound and insightful foundation for RL in non-Markovian environments.
- The authors successfully translate heavy topological concepts into an accessible, practically implementable algorithm.
- The HFPS algorithm is effectively validated across both exact tabular synthetic tests and continuous-control benchmarks with injected memory constraints.

Weaknesses:

- The algorithm exhibits suboptimal sample efficiency compared to standard baselines (e.g., PPO) in purely Markovian ("Clean") environments.

---

> ### Author Rebuttal · Authors · 2026-03-30
>
> Thank you for the positive evaluation and for the very relevant practical question. We are glad that the reviewer sees both the conceptual novelty and the algorithmic potential of the decomposition.
>
> **On comparison with Clean in Markov tasks**
> From different experiments/scenarios in the appendix (including the Clean/Noisy/Sticky runs on LunarLander-v2, Acrobot-v1, Pendulum-v1, PointMass, and BipedalWalker-v3.
> ), our observation is that HFPS and standard baselines like PPO have similar sample efficiency and performance in purely Markov tasks, while HFPS in many cases slightly outperforms the PPO baseline. For example, In the Clean row of Fig. 4 / Table 2, we do not see a systematic clean-regime penalty. HFPS is broadly comparable to strong baselines and is stronger on several tasks. In Final@T, HFPS improves over the strongest baseline on PointMass ($-207.14$ vs.\ $-433.00$, $+52.2\%$) and BipedalWalker ($84.49$ vs.\ $20.61$, $+310.0\%$); it is essentially tied on Pendulum ($-147.96$ vs.\ $-147.34$, $-0.4\%$) and close on LunarLander ($238.91$ vs.\ $246.91$, $-3.2\%$). So the Clean results do not indicate a consistent sample-efficiency disadvantage relative to PPO-style baselines.
>
> The similar performance makes sense because when the task is Markov, there is no structural deficiency to be removed by our HFPS projection. Different algorithms are therefore expected to have similar performance. This is also consistent with our theory. Thm. 3.8 shows that under a perfect MDP the mean-field Bellman defect vanishes, while Thm. 5.3 shows that when the residual is small the HFPS update asymptotically matches the standard TD semi-gradient. Thus, in purely Markov tasks HFPS should reduce to near-standard learning behavior; any slight gains are best interpreted as finite-sample denoising / regularization of the TD signal rather than exploitation of non-Markov structure.
>
> For scenarios where HFPS shows slight improvement over baselines like PPO in purely Markov tasks, we believe this is because even though the problem is Markov, our HFPS projection can still remove some form of sampling noise by projecting onto the desired exact 1-cochain. This interpretation is consistent with the Clean results summarized above.
>
>
> **On the application of HFPS as a modular plug-in to baselines like PPO / GAE.**
> Yes, this is a great suggestion. The reason we formulate HFPS as replacing the raw one-step TD signal by this discounted-integrable projection is to make it a modular plug-in. For PPO, since GAE is built from discounted accumulations of one-step TD residuals, one can naturally define a **topologically corrected GAE** by replacing the raw TD signal with our Hodge-filtered TD signal.
> We have implemented this in a modular way at the advantage-estimation level: GAE is still computed from discounted one-step residuals, but we replace the raw TD signal by a filtered signal obtained from the discounted-integrable projection. Concretely, if $\delta_t=r_t+\gamma V(s_{t+1})-V(s_t)$, we fit $u_\phi$ so that $u_\phi(s_{t+1})-\gamma u_\phi(s_t)\approx \delta_t$, and define $\tilde\delta_t=(1-\lambda)\delta_t+\lambda\big(u_\phi(s_{t+1})-\gamma u_\phi(s_t)\big)$. GAE is then computed from $\tilde\delta_t$, while the PPO clipping objective itself is left unchanged. We will report a quick plug-in check on BipedalWalker (Nonmarkov), using the same AUC@T / Final@T metrics as the main table: The plug-in result supports the reviewer's suggestion: PPO+HFPS substantially improves both AUC@T and Final@T over PPO, while remaining very close to standalone HFPS. This indicates that the decomposition can indeed be used as a modular correction layer at the advantage-estimation level without changing the PPO objective itself.
>
>
> | Method | Env | AUC@T | Final@T |
> |---|---:|---:|---:|
> | PPO | BipedalWalker (Nonmarkov) | 78.5 ± 155.1 | 56.4 ± 142.9 |
> | PPO+HFPS | BipedalWalker (Nonmarkov) | 181 ± 40.1 | 154.1 ± 35.6 |
> | HFPS | BipedalWalker (Nonmarkov) | 184.7 ± 96.1 | 146.0 ± 113.9 |
>
> Regarding the theoretical assumptions, the exact Hilbert-space analysis uses boundedness / square-integrability / closed-range conditions to define the projection cleanly, whereas the implemented method only relies on replay samples and empirical least-squares fitting of the potential network.

---

> > ### Author Rebuttal · Reviewer_vLvo · 2026-04-02
> >
> > Thanks for the rebuttal which has fully resolved my concerns. I maintain my positive assessment.

---

### Official Review · Reviewer_tq1t · 2026-03-12

**Soundness:** 3
**Presentation:** 2
**Significance:** 3
**Originality:** 4
**Overall Recommendation:** 5
**Confidence:** 3

**Summary:**

This paper presents a perspective on temporal-difference methods by borrowing concepts from algebraic topology and discrete exterior calculus. The study suggests that the TD method can be characterized as the 1-cochain of a potential function. Then, the notions of integrability and non-integrability are discussed, and the paper shows that the non-integrability can be measured by the norm of the topological residual $\delta_V^{res}$. Based on this perspective, the paper proposes an algorithm, HFPS. The paper also analyzes the sensitivity of the integrable component and the Hodge decomposition. Moreover, a bias bound on the gradient is provided (Theorem 5.3).

Overall, this paper presents a novel conceptual framework for temporal-difference methods. Revising the manuscript to make it more accessible to a broader audience would further improve the paper.

**Compliance With Llm Reviewing Policy:**

Affirmed.

**Final Justification:**

The authors provided detailed responses that adequately addressed my concerns.

**Key Questions For Authors:**

1. This study interprets the topological residual as a measure of non-Markovian properties, as discussed in Section 3.3. However, as mentioned in Section 3.5, the topological residual appears to reflect a combination of sampling noise and function approximation error. It would be helpful to provide additional experiments or ablations evaluating HFPS in environments with stronger stochastic noise or other disturbances.

2. In algebraic topology, the 1-cochain differential is typically defined as $d u(s,a,s') = u(s') - u(s)$. However, the definition used in this paper is $d u(s,a,s') = u(s') - \gamma u(s)$. I assume this modification is introduced to make the operator compatible with the temporal-difference formulation. If this differs from the standard definition, it would be helpful to clarify the relationship.

3. This study proposes a conceptually novel framework for temporal-difference methods. However, the terminology used in the paper, such as 1-cochains, may not be familiar to a broader machine learning audience. It might be helpful to include some mathematical preliminaries, for example in the appendix, so that the paper becomes more accessible to readers.

4. The proposed method introduces an additional potential network that is trained to project the TD error onto the integrable subspace. While conceptually appealing, it would be helpful to clarify the computational overhead of this projection step and how the method scales when applied to larger architectures or high-dimensional observation spaces.

**Limitations:**

yes

**Strengths And Weaknesses:**

Strengths:

1. This paper develops a novel conceptual framework for temporal-difference methods using notions from algebraic topology and discrete exterior calculus.
2. The theoretical results provide insights into temporal-difference methods. In particular, the paper shows a decomposition of the TD error using the notion of the exactness of cochains. From this perspective, the topological residual $\delta_V^{res}$ can be interpreted as an irregular component in the temporal difference, which may arise from non-Markovian dynamics and stochastic noise.
3. The paper presents several theoretical bounds, such as Theorems 5.1, 5.2, and 5.3, as well as results related to measuring Bellman non-integrability, such as Theorem 3.8.

Weaknesses:

1. The contents of this paper appear to be novel; however, I wonder whether the notations used in this paper, such as cochains, exactly correspond to those used in algebraic topology. The definition of the differential $d u(s,a,s') = u(s') - \gamma u(s)$ differs somewhat from the standard definition, which is typically given as $d u(s,a,s') = u(s') - u(s)$.

2. The notions and terminology used in this paper originate from algebraic topology. In order to make the paper accessible to a broader audience, it would be helpful if the manuscript were made more self-contained.

---

> ### Author Rebuttal · Authors · 2026-03-29
>
> Thank you for the thoughtful and constructive review. We appreciate both your positive assessment of the conceptual novelty and your suggestions on clarity, noise robustness, and computational overhead. We will revise the paper accordingly.
>
> **On the discounted differential and the relation to classical algebraic topology.**
> Yes, our definition $d_\gamma u(s,a,s') := u(s')-\gamma u(s)$ differs from the classical setting by adding a discount factor to standard simplicial coboundary \(u(s')-u(s)\). Exactly like you said, this is to match the temporal-difference methods in the Bellman form \(r+\gamma V(s')-V(s)\). Our analysis in this paper is based on this definition, and we will clarify this in the final version. Thank you!
>
>
> **On stronger noise / disturbance ablations.**
> This is a very good point. Yes, the residual could reflect both structural mismatch (i.e., Markov approximation error) and sampling noise. We included experiment results corresponding to **Clean, Noisy, and Sticky** variants of the scenarios, designed to isolate distinct sources of observation-side mismatch. Clean serves as a baseline, Noisy injects observation noise/corruption, and Sticky introduces strong temporal persistence (i.e., the level of non-Markov-ness). Contrasting Noisy and Clean illustrates the impact of noise in the residual computation, while Contrasting Sticky and Clean illustrates the impact of elevated non-Markov-ness. These experiments/variants are conducted over multiple environments including LunarLander-v2, Acrobot-v1, Pendulum-v1, PointMass, and BipedalWalker-v3.
>
> Let us further explain our observations from these experiments to answer this question. We will also add the discussions here into our final version.
> Across these comparisons, we observe a clear pattern: noise hurts all methods, but HFPS usually degrades more gracefully, and the benefit is more pronounced when the mismatch is structured rather than purely i.i.d.\ corruption. Fig. 4 shows that HFPS is competitive in the Clean regime and often more robust in the Noisy/Sticky regimes; Figs. 6 and 8 support this through cAUC and variability. For representative cases, on LunarLander-Noisy HFPS reaches \(100.88 \pm 43.37\), versus \(-36.76 \pm 127.95\) for the strongest baseline reported there; on PointMass-Noisy, \(-230.07 \pm 16.41\) versus \(-329.73 \pm 72.07\); and on PointMass-Sticky, \(-296.23 \pm 69.91\) versus \(-399.89 \pm 86.85\). At the same time, we do not claim that the residual is a purely non-Markov statistic at finite samples: as noted in Sec.~3.5 / Thm.~3.8, it also contains stochasticity and approximation error. This is exactly why the Clean/Noisy/Sticky comparison is informative: Clean keeps the original setting, Noisy adds observation corruption, and Sticky introduces stronger temporal persistence. Thus, our claim is robustness and graceful degradation under disturbance, rather than universal dominance on every task.
>
>
>
> **On accessibility to a broader ML audience.**
> Thanks. We will add a section on the preliminaries in the appendix in our final verson. It will cover definitions and concepts including \(C^0\), \(C^1\), the discounted differential, the exact subspace, the adjoint, and Hodge-type projection.
>
>
> **On computational overhead.**
> In terms of the computational overhead, HFPS adds one potential network and one projection-style auxiliary update on top of the critic update, but compared to some of the existing work, it does not require extra environment interaction, recurrent belief-state inference, longer histories, or additional roll-out machinery. Thus we believe the overhead is local to critic learning, rather than to propagating through the entire training pipeline.
>
> To make the overhead explicit, we add a wall-clock comparison under identical environment steps and hardware. The results below show that the extra cost is small (around 20\% in general), since the interaction loop, replay, target-network updates, and evaluation schedule are unchanged. HFPS only adds one potential network plus auxiliary projection updates. We share some numbers below and will provide full results in the final version.
>
> | Method | Task | Seeds | Steps | Wall-clock time (s) | Relative overhead |
> |---|---:|---:|---:|---:|---:|
> | DQN | cartpole | 5 | 100000 | 12.34 ± 0.56 | 1.00x |
> | HFPS | cartpole | 5 | 100000 | 15.67 ± 0.44 | 1.27x (+27\%) |
>
> We appreciate these suggestions and believe the final paper will be substantially clearer and stronger after incorporating them.

---

> > ### Author Rebuttal · Reviewer_tq1t · 2026-04-01
> >
> > The authors provided detailed responses that adequately addressed my concerns. I therefore maintain my positive assessment (score: 5, Accept).

---

### Decision · Program_Chairs · 2026-04-30

**Decision:**

Accept (regular)

**Comment:**

This work consider RL in non-Markovian environments, with a specific focus on TD-based methods. Since Bellman equation only partially accounts for the structure of the problem, they decompose the TD error into an integrable - Bellman compatible - component and a non-integrable residual. Leveraging an algebraic topology standpoint, they obtain a Hodge-type decomposition of TD errors and propose HodgeFlow Policy Search to stabilize TD learning in this regime.  Reviewers stressed the interest of the topic (non-Markovianity) as well as the novel perspective proposed in this paper. Overall, the paper offers a theoretically strong and original views on the impact of non-Markovian dynamics in RL, which motivates acceptance. This said, some (legitimate) concerns were raised about pedagogy and presentation - especially since the topology notions used are not common in the field. While this maybe the price for original contributions, improvements have been suggested in the reviews and rebuttal discussions, authors are thus encouraged to leverage them to enhance the manuscript on that end for the camera ready.